# Gut microbiomes of wild great apes fluctuate seasonally in response to diet

Allison L. Hicks[1], Kerry Jo Lee[1], Mara Couto-Rodriguez[1], Juber Patel[1], Rohini Sinha[1], Cheng Guo[1], Sarah H. Olson[2], Anton Seimon[3], Tracie A. Seimon[1,4], Alain U. Ondzie[2], William B. Karesh[5,6], Patricia Reed[2], Kenneth N. Cameron[2], W. Ian Lipkin[1,7,8,9] & Brent L. Williams [1,7,8]

The microbiome is essential for extraction of energy and nutrition from plant-based diets and may have facilitated primate adaptation to new dietary niches in response to rapid environmental shifts. Here we use 16S rRNA sequencing to characterize the microbiota of wild western lowland gorillas and sympatric central chimpanzees and demonstrate compositional divergence between the microbiotas of gorillas, chimpanzees, Old World monkeys, and modern humans. We show that gorilla and chimpanzee microbiomes fluctuate with seasonal rainfall patterns and frugivory. Metagenomic sequencing of gorilla microbiomes demonstrates distinctions in functional metabolic pathways, archaea, and dietary plants among enterotypes, suggesting that dietary seasonality dictates shifts in the microbiome and its capacity for microbial plant fiber digestion versus growth on mucus glycans. These data indicate that great ape microbiomes are malleable in response to dietary shifts, suggesting a role for microbiome plasticity in driving dietary flexibility, which may provide fundamental insights into the mechanisms by which diet has driven the evolution of human gut microbiomes.

[1] Center for Infection and Immunity, Columbia University, New York, NY 10032, USA. [2] Wildlife Conservation Society, Wildlife Health Program, Bronx, NY 10460, USA. [3] Department of Geography and Planning, Appalachian State University, Boone, NC 28608, USA. [4] Wildlife Conservation Society, Zoological Health Program, Bronx, NY 10460, USA. [5] Wildlife Conservation Society, Global Health Program, Bronx, NY 10460, USA. [6] EcoHealth Alliance, New York, NY 10001, USA. [7] Department of Epidemiology, Mailman School of Public Health, Columbia University, New York, NY 10032, USA. [8] Department of Pathology and Cell Biology, College of Physicians and Surgeons, Columbia University, New York, NY 10032, USA. [9] Department of Neurology, College of Physicians and Surgeons, Columbia University, New York, NY 10032, USA. Correspondence and requests for materials should be addressed to B.L.W. (email: bw2101@cumc.columbia.edu)

The gastrointestinal microbiome impacts states of health and disease through various mechanisms relating to metabolism, immunity, and development. Evidence is accumulating in humans and animals to suggest that microbiota composition is not a static state defining an individual but fluctuates in response to changes in environmental and lifestyle factors. A recent study has demonstrated seasonal reconfiguration of the microbiome in response to dietary fluctuation in the Hadza hunter-gatherers of Tanzania[1]. However, such seasonal fluctuations in other primate microbiomes remain under-characterized.

Unlike chimpanzees, which are ripe fruit specialists, seeking out ripe fruit throughout the year, western lowland gorillas (WLGs, *Gorilla gorilla gorilla*) are seasonal frugivores, selectively shifting their dietary habits throughout each year to accommodate seasonal resource availability[2]. During periods of high-fruit availability, up to 70% of WLG feeding time is devoted to succulent fruit, while, during periods of fruit scarcity, feeding time is almost exclusively devoted to leaves, bark, herbs, and fibrous fruits[2]. Ripe fruit availability is reduced, though still regulated seasonally, in environments inhabited by mountain gorillas (*G. beringei beringei*)[3, 4]. WLG and mountain gorilla microbiomes converge during periods of fruit scarcity[3], suggesting that seasonal resource availability may have dramatic impacts on gorilla microbiomes. However, more complete sampling across months and seasons is needed in order to draw conclusions about the full spectrum of seasonal variation in microbiomes of wild gorillas. Geographical range of WLGs may also influence microbiome composition and metabolomic profiles[5], but extreme seasonal variation in WLG frugivory were not addressed. Dramatic seasonal shifts in WLG diets present a unique opportunity to examine short-term effects of recurrent dietary fluctuations on the composition of the microbiome.

Human studies from various populations have suggested that the relative abundance of core bacterial taxa in intestinal microbiomes of individuals varies widely and that individuals cluster within two to three stratified variants or "enterotypes" defined by *Bacteroides*, *Prevotella*, or *Ruminococcus*[6–10]. These enterotypes, or stratified clusters within a population defined by microbiota composition, are linked to long-term dietary patterns[8]. *Bacteroides* enterotypes in humans are associated with animal-based diets, while *Prevotella* enterotypes are associated with plant-based, carbohydrate-rich diets[8], and these enterotypes are functionally distinguished based on their saccharolytic, proteolytic, and lipolytic profiles[11, 12]. However, the relative abundance of core taxa defining enterotypes can change, at least within some individual humans and animals, leading to enterotype switching. Previous studies have demonstrated enterotype switching in non-human primates (NHPs) on the scale of months and in wild-caught mice within one week[13–16]. While enterotype switching has also been reported in human populations[7, 17, 18], factors governing shifts in microbial taxa that define the majority of microbiota variance in a population, and thus enterotypes, of humans and other animals remain unclear. From an evolutionary perspective, environmental instability may be conceptually important, as functional variations between compositionally distinct gut microbial communities may have facilitated adaptation of early primates to changing nutritional resource availability[19].

In this study, we investigated fecal microbiota of 87 individual unhabituated wild WLGs and 18 sympatric central chimpanzees (*Pan troglodytes troglodytes*) from the Republic of the Congo, with samples collected over a three-year period. To place our results in the context of humans and other NHPs, we compared fecal microbiota from these WLGs and chimpanzees to those of US and Mongolian humans[9] and Old World monkeys[15, 20]. We further investigated temporal dynamics of WLG and chimpanzee

microbiomes in order to understand how seasonal variation in resource availability may drive fluctuation in the composition of the microbiome. Using metagenomic approaches, we compared microbial functional metabolic pathways, archaeal communities, and dietary plant content among representative WLG fecal samples. This study presents evidence that the microbiomes of our closest living relatives fluctuate seasonally in response to diet, and provides insight into how symbiotic bacteria may have shaped the trajectory of primate evolution.

## Results

**Composition of WLG and sympatric chimpanzee gut microbiota.** To characterize microbiotas of wild African great apes, we sequenced the V1–V3 region of the bacterial 16S rRNA gene in fecal samples collected from 87 individual WLGs (Fig. 1a) and 18 sympatric chimpanzees (Supplementary Data 1-3) from the Sangha region of the Republic of the Congo (Fig. 1b, Supplementary Note 1) over 2008–2010. These data were compared with V1–V3 sequences from studies of Old World monkeys (baboons, black-and-white colobus, red colobus, and red-tailed guenon)[15, 20], as well as from human populations from the US (Human Microbiome Project) and Mongolia[9].

Firmicutes dominated the WLG microbiota (Fig. 1c), as was the case for other NHPs (Supplementary Fig. 1a–e). In contrast, US and Mongolian humans are dominated by the Bacteroidetes phylum (Supplementary Fig. 1f, g), a distinction especially evident in the US population. The most striking difference in WLGs compared to humans and other NHPs is the high-relative abundance of the phylum Chloroflexi. Further, Spirochaetes is the fourth most abundant phylum in both WLGs and sympatric chimpanzees, while it is either absent or present in very low-relative abundance in Old World monkeys and humans. We found no evidence for gender-specific differences in WLG microbiota. For additional information on phylum- and genus-level composition of WLG and sympatric chimpanzee microbiota compared to humans and Old World monkeys, see Supplementary Data 4, Supplementary Fig. 2 and Supplementary Note 2.

Genus- and phylum-level alpha diversity metrics (observed taxa and Chao1) revealed significantly higher diversity and richness in NHPs (with the exception of baboons in genus-level analyses) compared to US and Mongolian humans (Fig. 1d and Supplementary Fig. 3a, 3c, and 3d). Consistent with previous findings, US humans had the lowest diversity and richness at these taxonomic levels[21]. This previous study also indicated that WLGs have greater genus-level diversity compared to allopatric chimpanzee populations[21]. Our findings comparing cross-seasonally collected samples from sympatric WLGs and chimpanzees indicates that genus-level diversity and richness are higher in chimpanzees than WLGs. Genus-level Shannon index was higher in all NHPs compared to humans, with US humans displaying the lowest index (Supplementary Fig. 3b). Phylum-level Shannon index was highest in African great apes (Supplementary Fig. 3e).

Inter-individual genus-level Bray–Curtis (BC) dissimilarity was higher in WLGs compared to chimpanzees and both human populations (Supplementary Fig. 4a). Inter-individual dissimilarity was highest in baboons. Both WLGs and baboons have pronounced seasonal variation in their diets, which may have a greater impact on inter-individual dissimilarity in these NHP populations[15]. Chimpanzees may have less dissimilarity in their microbial composition as they seek out ripe fruits year-round. Thus, seasonal variation in succulent fruit availability, while relevant, may have less impact on chimpanzee microbiota (see analyses below).

Principle coordinate analysis (PCoA) based on genus-level BC dissimilarity revealed distinct clustering of NHPs that separated from both US and Mongolian humans (Fig. 1e). US humans separated from both NHPs and most Mongolian humans along PC1 and PC2. In contrast, NHPs and Mongolian humans separated further along PC2. This likely reflects the high-relative abundance of *Bacteroides* in the US population, which is not found in NHPs and is less common in Mongolians (Supplementary Fig. 1 and Fig. 1c). Partitioning around medoids (PAM) clustering of all primate groups, inter-group BC dissimilarity, and unweighted UniFrac suggested that, in general, African great apes and Old World monkeys were more similar to each other than to either human population (Supplementary Figs. 4b–d, 5a, b and Supplementary Note 3), consistent with previous findings showing that distantly related NHPs and non-primates with largely folivorous diets form a distinct cluster based on unweighted UniFrac distance[19].

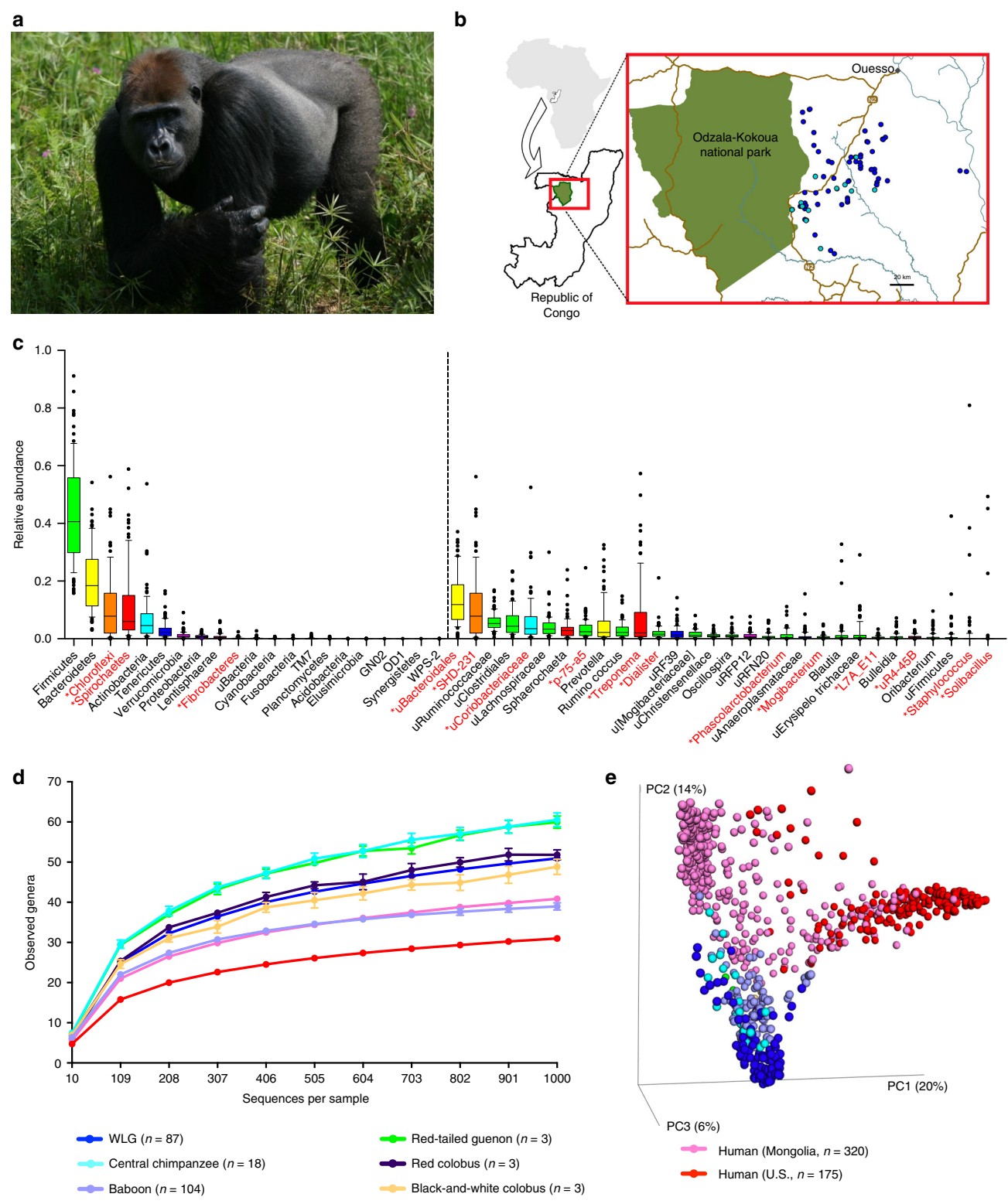

**Unique composition of WLG and sympatric chimpanzee microbiota**. In humans, inter-individual differences in gut microbiota appear to derive largely from continuous gradients of bacterial taxa[22, 23]. In contrast, some studies suggest that human microbiotas may be stratified into clusters or enterotypes[7, 24]. However, evaluation of variation in microbiotas of wild animal populations is sparse. Therefore, we have investigated inter-species and intra-species differences in the microbiota using methods that can identify differences arising from continuous variation of raw taxon-relative abundance, as well as stratified clusters.

Linear discriminant analysis effect size (LEfSe)[25] analysis revealed that unclassified Bacteroidales, SHD-231, and *Treponema* were the top genus-level biomarkers distinguishing WLGs from all other host groups (Fig. 2a and Supplementary Data 5). Sympatric chimpanzees were distinguished from all other groups by the relative abundance of *Erysipelotrichaceae* taxa, unclassified *Prevotella*, RFN20, and *Sphaerochaeta*. *Bacteroides* was the top genus-level biomarker of US humans, while *Prevotella* relative abundance was significantly higher in Mongolian humans. Baboons were unique in their high-relative abundance of *Bifidobacterium*, while other Old World monkeys were distinguished by their high-relative abundance of Firmicutes and Mollicutes at various taxonomic levels.

Moeller et al.[13], reported that eastern chimpanzees harbor gut communities that cluster into enterotypes analogous to human enterotypes. More recent studies by Moeller et al.[13], report that human microbiomes have substantially diverged from those of African apes at an accelerated rate[21], but that humans and gorillas share a *Prevotella* enterotype[14]. These studies do not address effects of seasonal factors on African great ape enterotypes and their convergence with or divergence from those of humans. We therefore evaluated the presence of enterotypes in microbiotas of WLGs and sympatric chimpanzees, as well as baboons and US and Mongolian humans, as described by Arumugam et al.[6] (Fig. 2b and Supplementary Fig. 5). These analyses identified four WLG enterotypes defined by the Chloroflexi genus SHD-231 (enterotype 1), *Treponema* (enterotype 2), *Prevotella* (enterotype 3) and *Solibacillus* and *Staphylococcus* (enterotype 4). Sequences from bacterial taxa whose relative abundance define unique WLG enterotypes (*Treponema*, SHD-231, and *Solibacillus/Staphylococcus*) were all most closely related to ruminant-derived strains (Supplementary Fig. 6). WLG enterotype 1 fell between enterotypes 2 and 3 along PC1, but separated along PC2 (Fig. 2b). While overlap was apparent between WLG enterotypes 3 and 4 along PC1 and PC2, enterotype 4 separated widely along PC3 from the other three enterotypes (Fig. 2b and Supplementary Fig. 7a). In sympatric chimpanzees, we identified only two enterotypes defined by *Sphaerochaeta* and *Prevotella*. Baboons also have two enterotypes: one defined by several genera, including *Faecalibacterium* and *Clostridium*, and another defined by *Bifidobacterium*. Both human populations had two enterotypes

defined by *Prevotella* and *Bacteroides*. The only enterotype shared among humans, chimpanzees, and WLGs was the *Prevotella* enterotype. For additional information on validation and geographic distribution of these enterotypes, see Supplementary Figs. 5, 7, and 8a, b and Supplementary Note 4.

**Seasonal variation in WLG and sympatric chimpanzee microbiota**. Monthly variation in WLG microbiota was evident (Fig. 3a–c, Supplementary Note 5, and Supplementary Fig. 9a). However, temporal relative abundance distribution of monthly bacterial biomarkers may be indicative of broader seasonal shifts in bacterial composition. Thus, seasonal variation could serve as a better proxy for changes in vegetation that impact WLG feeding behavior. Seasonality in most of the Congo Basin is characterized by a bimodal distribution of rainfall with alternating wet and dry seasons that correlate with a bimodal vegetation profile[26]. We analyzed Tropical Rainfall Measuring Mission (TRMM) rainfall data in the region of sampling from 2001 to 2010 to confirm the bimodal seasonality of rainfall in the sampling area (Fig. 4a, Supplementary Fig. 9b). Although seasonal trends supporting two wet and two dry seasons are apparent, anomalies during the period of collection (2008–2010) could have impacted vegetation and thus WLG feeding behavior and, ultimately, the microbiota. Total regional monthly rainfall for each of the years of sample collection indicated that some months did not follow long-term seasonal trends (Fig. 4b). Furthermore, responses of vegetation to rainfall in tropical forests of Central Africa have a phase lag of up to a month[26]. Examination of enhanced vegetation index for the sampling region from 2008 to 2010 indicated that while there was little inter-annual variation in vegetation levels, for some seasons (wet season 1 and dry season 2), changes in vegetation gradually occurred in response to rainfall, while in other seasons (wet season 2 and dry season 1), vegetation levels responded rapidly to changes in rainfall (Fig. 4c, Supplementary Fig. 9c). Therefore, we have evaluated compositional differences in the microbiota in WLGs in relation to both long-term seasonal trends and average daily rainfall over the 30 days prior to the date of collection for each sample, used to adjust for the lag phase for vegetation response and any yearly anomalies. LEfSe was used to examine variations in WLG fecal microbiota over the two wet seasons and two dry seasons defined by decadal rainfall patterns (Fig. 4a), as well as more discrete wet and dry periods, determined by average daily rainfall over the 30 days prior to sample collection (defined as wet and dry months, Supplementary Fig. 9d). By both measures, *Treponema* relative abundance was significantly associated with wet periods (seasons or months) (Supplementary Fig. 10a, b). Furthermore, the relative abundance of *Prevotella* and unclassified *Clostridiaceae* were associated with individual dry seasons (Supplementary Fig. 10a). These differences were not attributable to year of sample collection, collection site latitude or longitude, or gender.

**Fig. 1** Sample collection and fecal microbiota composition of WLGs and chimpanzees compared to humans and other non-human primates. **a** WLG (*G. g. gorilla*)-photo by T. Breuer/WCS. **b** Collection sites of the 87 WLG and 18 chimpanzee fecal samples (dark blue and turquoise dots, respectively). All collection coordinates were east of Odzala-Kokoua National Park in the Sangha region of the Republic of the Congo. Raw spatial files were sourced from public or open source databases, including Map Library (www.maplibrary.org; Africa and Republic of Congo), The World Bank Data Catalog (https://datacatalog.worldbank.org; roads), Humanitarian OpenStreetMap Team (https://data.humdata.org; rivers) and the World Resource Institute Congo Basin Forest Atlases (www.wri.org/our-work/project/congo-basin-forest-atlases; Odzala). Maps were created and modified with R version 3.4.0, Inkscape and Gimp software. **c** Box-and-whiskers plots showing the distributions of all bacterial phyla (left of dotted line), as well as the 30 most abundant bacterial genera (right of dotted line) in feces from the 87 WLGs in this study. Phyla and genera labels in red with asterisks indicate bacterial taxa identified by LEfSe as being elevated in WLGs compared to other host types (see Fig. 2a). Phyla and genera labels preceded by "u" represent taxonomic groups that could only be classified at higher taxonomic levels. **d** Alpha diversity rarefaction plot based on the observed genera metric. Mean observed genera for each host type are shown with the standard error of the mean. **e** PCoA plot based on the BC dissimilarity among bacterial genus-level abundance distributions for each host type. Color designations for each host type shown in **d**, **e** are indicated at the bottom

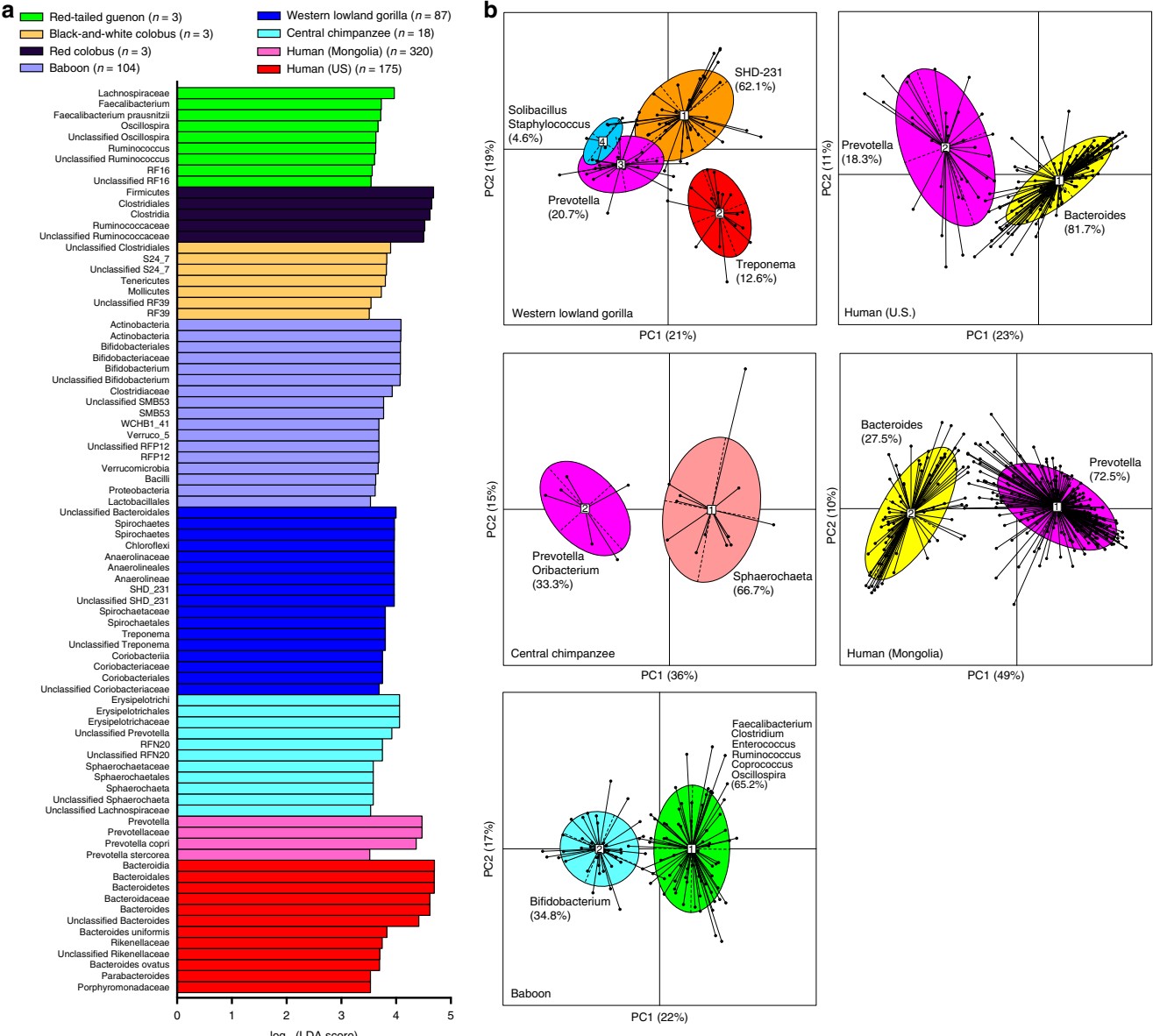

**Fig. 2** The relative abundance of fecal bacterial taxa and enterotype clusters distinguish WLGs, sympatric chimpanzees, Old World monkeys, and two human populations. **a** Bar chart showing the log-transformed LDA scores of bacterial taxa of each NHP or human host type identified by LEfSe analysis. Note that while a log-transformed LDA score of 2 was used as a threshold for identification of significant taxa, LEfSe analysis of this dataset yielded 377 taxa. Therefore, only bacterial taxa with LDA score ≥3.5 are shown (see Supplementary Data 5 for all associated taxa). **b** Principal coordinate analysis plots based on rJSD among genus-level relative abundance distributions with the four WLG, two chimpanzee, two baboon, two Mongolian human, and two US human enterotypes identified by PAM analysis indicated. Numbered white rectangles indicate the centroid of each cluster, while solid black lines indicate the distance of each sample from the centroid of the cluster. Defining genera of each enterotype identified by BCA and LEfSe analysis (see Supplementary Fig. 5n–r) are indicated along with the percentage of samples from the dataset that was assigned to each enterotype

The relative abundance of two genera that defined WLG enterotypes, *Treponema* and *Prevotella*, varied with seasonal rainfall patterns; thus, we investigated differences in average daily rainfall over the 30 days prior to sample collection among samples stratified by enterotype (Fig. 5a). Samples from enterotype 2, defined by *Treponema*, were collected during periods with higher rainfall than samples from each of the other three enterotypes. Furthermore, the proportions of samples assigned to each enterotype were different between wet and dry months, as well as wet and dry seasons (Fig. 5b and Supplementary Fig. 10c, d), and proportions of both enterotype 2 and enterotype 3 differed significantly between wet and dry months (Fig. 5b). Collectively, these data support a

seasonally defined distribution of enterotypes in this WLG population.

While we had fewer sympatric chimpanzees, precluding evaluation of monthly variation in their microbiota, we have evaluated stratification by wet and dry months. Chimpanzees are ripe fruit specialists, but also adjust their dietary patterns in response to seasonal availability of fruit. *Prevotella* was a significant biomarker of chimpanzee samples collected in dry months (Supplementary Fig. 10e), while unclassified Deltaproteobacteria and unclassified Spirochaetes were the top taxa associated with wet months. Similar to what we found for WLGs, chimpanzee samples from enterotype 1, defined by *Sphaerochaeta* relative abundance, were collected during periods of significantly

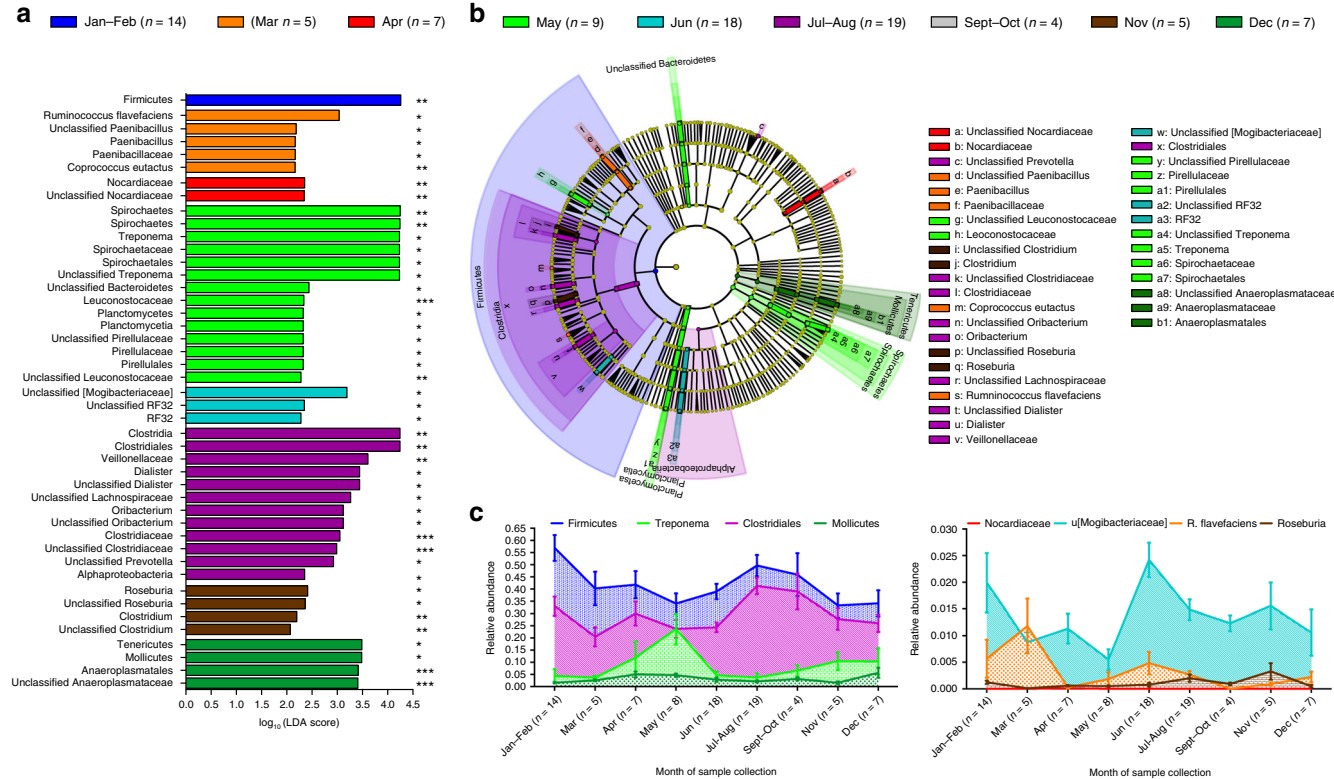

**Fig. 3** The fecal microbiota composition of WLGs varies by month of sample collection. **a** Bar chart showing the log-transformed LDA scores and **b** cladogram showing the phylogenetic relationships of bacterial taxa found to be significantly associated with month of sample collection by LEfSe. Asterisks in **a** indicate that Box-Cox-transformed relative abundance was significantly predicted by month of sample collection in ANCOVA analyses, adjusting for the effects of potentially confounding variables (*$P < 0.05$, **$P < 0.01$, ***$P < 0.001$). **c** Area charts showing the mean monthly relative abundance of the top bacterial taxa associated with each month. Error bars indicate the standard error of the mean

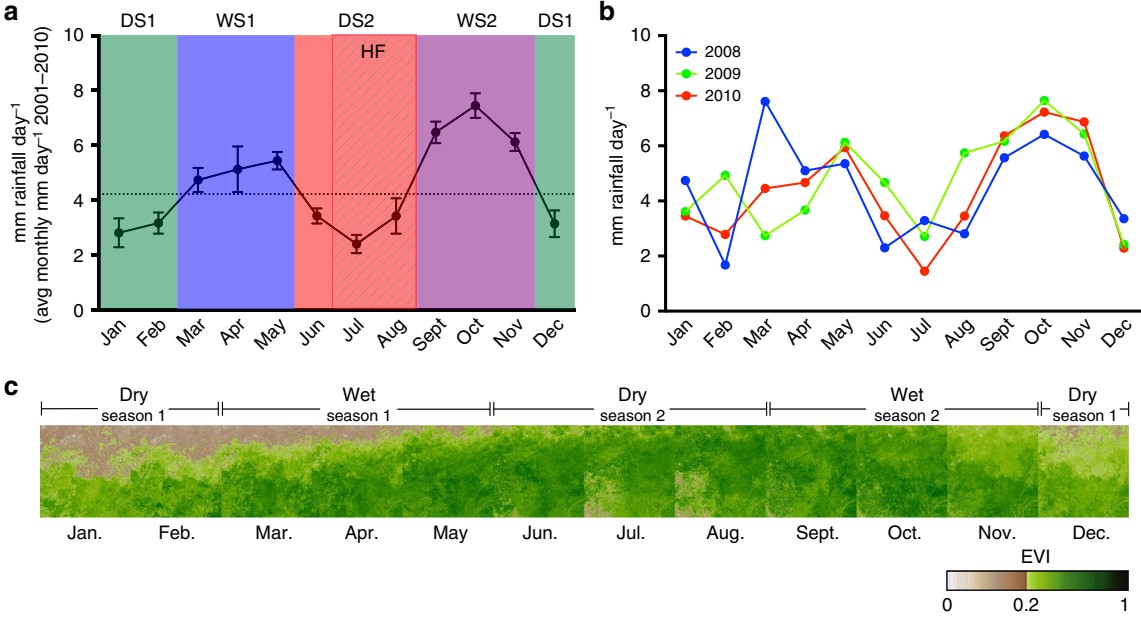

**Fig. 4** Seasonal rainfall patterns for Sangha region. **a** Line graph indicating the rainfall rate (average rainfall per day over each month from 2001 to 2010 with the standard error of the mean indicated) based on TRMM satellite data for the Sangha region. The two wet seasons (WS1 and WS2) and the two dry seasons (DS1 and DS2) are indicated by colored shading. The dashed line indicates the cutoff for average per day monthly rainfall between wet and dry months (4.23 mm day$^{-1}$). The season with the highest fruit (HF) availability (Jul–Aug) is also indicated. **b** Line graph indicating the total monthly rainfall for the individual years in which samples were collected. **c** Per-month enhanced vegetation indices based on Terra/MODIS satellite data for the region of sample collection from 2010

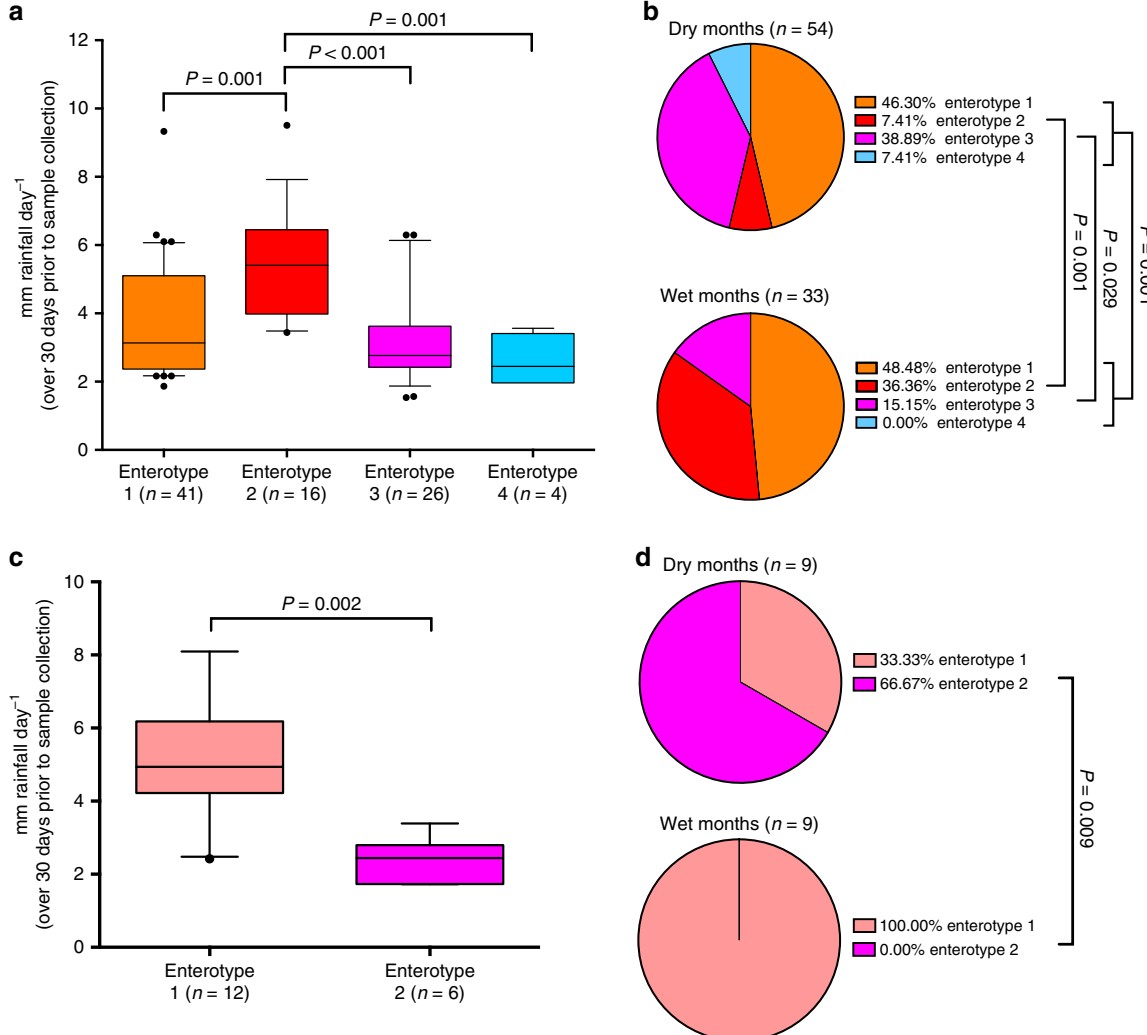

**Fig. 5** The fecal microbiota compositions and enterotype distribution of WLGs and sympatric chimpanzees vary with seasonal patterns of rainfall. **a** Box-and-whiskers plots showing the distribution of average rainfall per day over the 30 days prior to sample collection for each WLG enterotype. **b** Pie charts showing the proportion of samples from each WLG enterotype in wet (>4.23 mm day$^{-1}$) and dry months (<4.23 mm day$^{-1}$). **c** Box-and-whiskers plots showing the distribution of average rainfall per day over the 30 days prior to sample collection for each chimpanzee enterotype. **d** Pie charts showing the proportion of samples from each chimpanzee enterotype in wet (>4.23 mm day$^{-1}$) and dry months (<4.23 mm day$^{-1}$). Two-tailed $P$-values from Mann–Whitney tests are indicated in **a** and **c**. Two-tailed $P$-values from a $\chi^2$-test (all WLG enterotypes in dry months vs. wet months) and Fisher's exact tests (WLG enterotype 2 in dry months vs. wet months, WLG enterotype 3 in dry months vs. wet months, chimpanzee enterotypes 1 and 2 in dry months vs. wet months) are indicated in **b**, **d**

higher rainfall than samples from enterotype 2, defined by *Prevotella* relative abundance (Fig. 5c). Furthermore, proportions of chimpanzee samples assigned to enterotype 1 and enterotype 2 differed significantly between wet and dry months (Fig. 5d). These results suggest that seasonality in these sympatric WLG and chimpanzee populations impacts both the relative abundance of taxa and defines community enterotype distribution.

**Frugivory-associated changes in WLG and chimpanzee microbiota.** Studies of WLG and other NHP feeding patterns in the Republic of the Congo and surrounding areas indicate that even in the case of a bimodal rainfall distribution, succulent fruit availability and consumption are uniquely elevated in dry season 2, predominantly in July and August[2, 27–30]. In contrast, during low frugivory (LF) periods (October–May, with June and September excluded as transitional months), leaves constitute the major fallback food. This distinct seasonality of diet provides an

opportunity to investigate impacts of pronounced dietary shifts on the microbiotas of great ape populations.

We identified bacterial taxa associated with the high frugivory (HF) season of July–August compared to periods of LF, even after adjusting for year of collection, collection site latitude or longitude, or gender (Fig. 6a). HF-associated bacterial taxa were dominated by Clostridia and *Prevotellaceae* members, which rose dramatically in relative abundance in July–August and declined over a period of months (September–December) thereafter (Fig. 6b). The slow decline in HF bacteria over an extended period may be indicative of a slow decline in residual succulent fruit consumption as availability wanes. Alternatively, HF bacteria, once established in WLG intestines, may resist replacement by other bacteria, despite dietary shifts. Interestingly, *Treponema* was the only genus associated with LF (Fig. 6a). *Treponema* relative abundance rose dramatically from March to May, declining rapidly in June as HF bacteria spike (Fig. 6c). Then, as HF bacteria declined over the next several months,

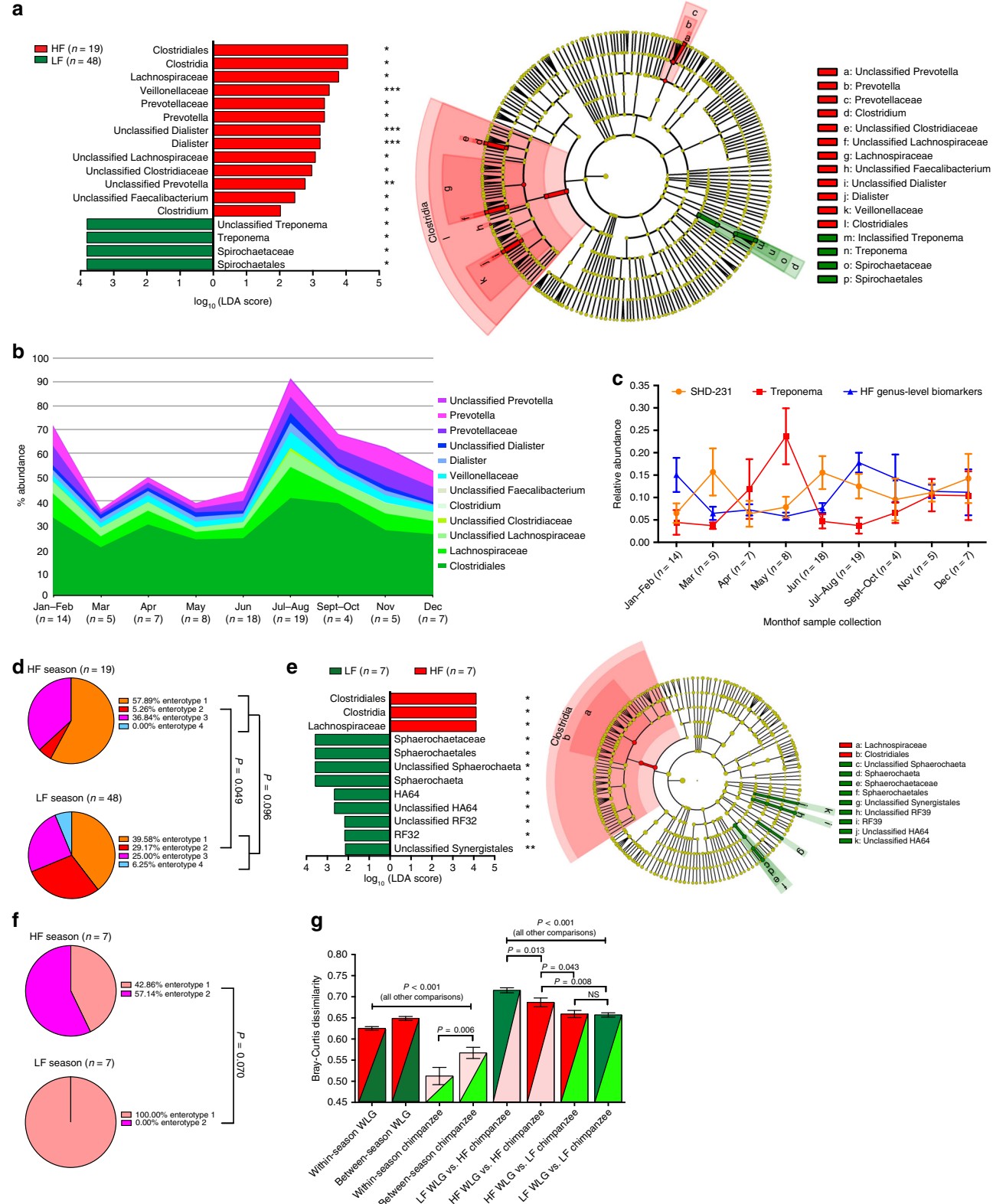

*Treponema* abundance rose steadily. We investigated differences in the distributions of samples assigned to each enterotype between HF and LF seasons and found that, similar to seasons defined by rainfall, prevalence of enterotype 2 varied significantly between HF and LF seasons (Fig. 6d). This is consistent with PCoA analysis based on BC dissimilarity showing separation of *Treponema* and *Prevotella* enterotypes along PC1, as well as a

clear gradient along PC1 based on the ratio of *Treponema* to *Prevotella* relative abundance (Supplementary Fig. 11a, b). Accordingly, PCoA analysis further demonstrates that the density of HF samples changes along PC1, consistent with changing relative abundance ratios of *Treponema* to *Prevotella* along PC1 and the near complete exclusion of samples collected during the HF period falling within *Treponema* enterotype 2

(Supplementary Fig. 11c). Furthermore, though not associated with HF or LF periods, *SHD-231* relative abundance was elevated in March and June when both *Treponema* and HF bacterial abundance were diminished (Fig. 6c). The relative abundance of *SHD-231* was the primary factor driving separation along PC2, and the ratio of the relative abundance of *SHD-231* to the cumulative relative abundance of *Treponema* and *Prevotella* revealed a clear gradient along PC2 (Supplementary Fig. 11d). This raises the possibility that *SHD-231* defines a transitional enterotype-like state where changes in dietary patterns (i.e., switch between high-fiber fallback foods and frugivory) have occurred, but the microbiome has not yet responded to a degree that results in switching between enterotypes 2 and 3. Indeed, within *SHD-231* enterotype 1 samples, the ratio of *Treponema* to *Prevotella* relative abundance differs significantly based on whether enterotype 1 samples were collected in the HF or LF season. The majority of HF-associated bacterial taxa, including *Prevotella*, are significantly elevated in enterotype 1 samples collected during the HF season, while *Treponema* is elevated in those collected during the LF season (Supplementary Fig. 11e, f). We found no evidence for an association between geographic distance between samples and pairwise microbiota BC dissimilarity within either the HF or LF season (Supplementary Fig. 8c) or between pairwise changes in *Treponema* or *Prevotella* relative abundance (Supplementary Fig. 8d, e).

While chimpanzees are more frugivorous than WLGs throughout the year, they also respond to seasonal changes in succulent fruit availability by increasing time spent feeding on fruit[31]. *Sphaerochaeta*, defining chimpanzee enterotype 1, was the top-ranked genus-level biomarker of the LF season (Fig. 6e), mirroring findings in WLGs where the Spirochaete *Treponema* was the top biomarker of the LF period (Fig. 6a). Similarly, Clostridia members, including *Lachnospiraceae*, were the top taxa associated with the HF season for both WLGs and chimpanzees (Fig. 6a, e, respectively). While *Prevotella* relative abundance was not a biomarker for HF in chimpanzees, *Prevotella* enterotype was associated with total rainfall and dry months in both WLGs and chimpanzees (Fig. 5a–d). We further observed a trend suggesting that the prevalence of chimpanzee enterotypes 1 and 2 varied between HF and LF (Fig. 6f).

These results suggest that resource availability may have a similar impact on microbiotas of sympatric African great ape species. However, impacts of frugivory on WLGs may be more extreme as they rely more heavily on fallback foods during periods of fruit scarcity. Accordingly, BC dissimilarity between WLGs in the HF season compared to WLGs in LF season was significantly higher than the dissimilarity between chimpanzees in the HF season compared to chimpanzees in the LF season (Fig. 6g). Niche separation among sympatric species may be most distinct during seasons of resource scarcity[32]. In both WLGs and

chimpanzees, BC dissimilarity was significantly lower in within-season (HF vs. HF and LF vs. LF) compared to between season (HF vs. LF) comparisons (Fig. 6g). Between sympatric African great apes, we find that BC dissimilarity is greatest between WLGs in the LF season and chimpanzees in the HF season (Fig. 6g). This comparison represents the most extreme dietary divergence between these apes, based on heavy reliance of WLGs on fallback foods during periods of fruit scarcity and accelerated frugivory in chimpanzees during the HF season. In contrast, when WLGs are in the HF season, their microbiotas become more similar to chimpanzees in both the HF and LF seasons (Fig. 6g). Surprisingly, WLGs and chimpanzees in the LF season had low dissimilarity (Fig. 6g). Collectively, these data suggest that composition of WLG and chimpanzee microbiota, including bacterial taxa that define enterotypes, is dynamic and responds seasonally to shifting resource availability and dietary preferences.

**Comparison of functional pathways between enterotype groups.** In order to investigate differences in functional capacity associated with WLG enterotypes, we performed metagenomic sequencing on representative WLG samples (*SHD-231*-abundant enterotype 1, $n = 5$; *Treponema*-abundant enterotype 2, $n = 5$; *Prevotella*-abundant enterotype 3, $n = 5$; and *Solibacillus/Staphylococcus*-abundant enterotype 4; $n = 4$).

WLG *SHD-231*-abundant enterotype 1 samples were enriched in superpathways for nucleotide sugar biosynthesis, purine metabolism, and carbohydrate synthesis compared to the other three enterotypes (Fig. 7). Additionally, enterotype 1 samples were enriched in superpathways for cell wall biogenesis, bacterial outer membrane biogenesis, capsule biogenesis, and spore coat biogenesis (Supplementary Data 6). See Fig. 7, Supplementary Note 6, Supplementary Data 6, and Supplementary Fig. 12 for additional information on metabolic functions associated with WLG enterotype 1.

WLG *Treponema*-abundant enterotype 2 samples, which were associated with the LF period when WLGs rely on plant parts higher in fiber and protein, were enriched in superpathways for pyrimidine metabolism, fermentation, glycan metabolism, and glucan metabolism (Fig. 7 and Supplementary Data 6). Enterotype 2 samples were enriched in various pathways for degradation of plant fibers, including beta-D-glucan, cellulose, xyloglucan, and starch degradation (Fig. 7 and Supplementary Data 6). Further analyses at the gene level demonstrated enrichment in cellulolytic (endoglucanase and cellodextrinase), hemicellulolytic (alpha-glucuronidase), lignocellulolytic (ligninase), and plant xyloglucan polymer degrading (xyloglucanase) enzymes[33] (Fig. 8a–e). Enrichment in plant-fiber degrading pathways and genes, enriched capacity for fermentation, and association of *Treponema* relative abundance with the LF period

**Fig. 6** The fecal microbiota compositions and enterotype distributions of WLGs and sympatric chimpanzees varies with fruit availability. **a** Bar chart showing the log-transformed LDA scores and cladogram showing the phylogenetic relationships of bacterial taxa in WLGs found to be significantly associated with seasons of high or low-fruit availability (HF or LF, respectively) by LEfSe. **b** Stacked area charts showing the mean monthly relative abundance for bacterial taxa associated with the HF season for WLGs. **c** Line chart showing the average monthly relative abundance of *Treponema* and SHD-231 in WLGs, as well as the average monthly cumulative relative abundance of genus-level HF biomarkers identified by LEfSe analysis. **d** Pie charts showing the proportion of samples from each WLG enterotype in the HF season (Jul–Aug) and LF season (Jan–May, Oct–Dec). **e** Bar chart showing the log-transformed LDA scores and cladogram showing the phylogenetic relationships of bacterial taxa in chimpanzees found to be significantly associated with seasons of high or low-fruit availability (HF or LF, respectively) by LEfSe. **f** Pie charts showing the proportion of samples from each chimpanzee enterotype in the HF season (Jul–Aug) and LF season (Jan–May, Oct–Dec). **g** Bar chart showing the mean intra- and inter-season (HF and LF) BC dissimilarities within and between WLG and chimpanzee samples based on genus-level relative abundance distributions. Asterisks in **a**, **e** (left panels) indicate that Box-Cox-transformed relative abundance was significantly predicted by HF or LF season in ANCOVA analyses, adjusting for the effects of potentially confounding variables (*$P < 0.05$, **$P < 0.01$, ***$P < 0.001$). Error bars in **c** and **g** indicate the standard error of the mean. Two-tailed $P$-values from a $\chi^2$-test (all enterotypes in the HF season vs. the LF season) and Fisher's exact tests (WLG enterotype 2 in the HF season vs. the LF season, WLG enterotype 3 in the HF season vs. the LF season, chimpanzee enterotypes 1 and 2 in the HF season vs. the LF season) are indicated in **d** and **f**. Two-tailed $P$-values from Mann–Whitney tests are indicated in **g**. NS not significant at $\alpha = 0.05$

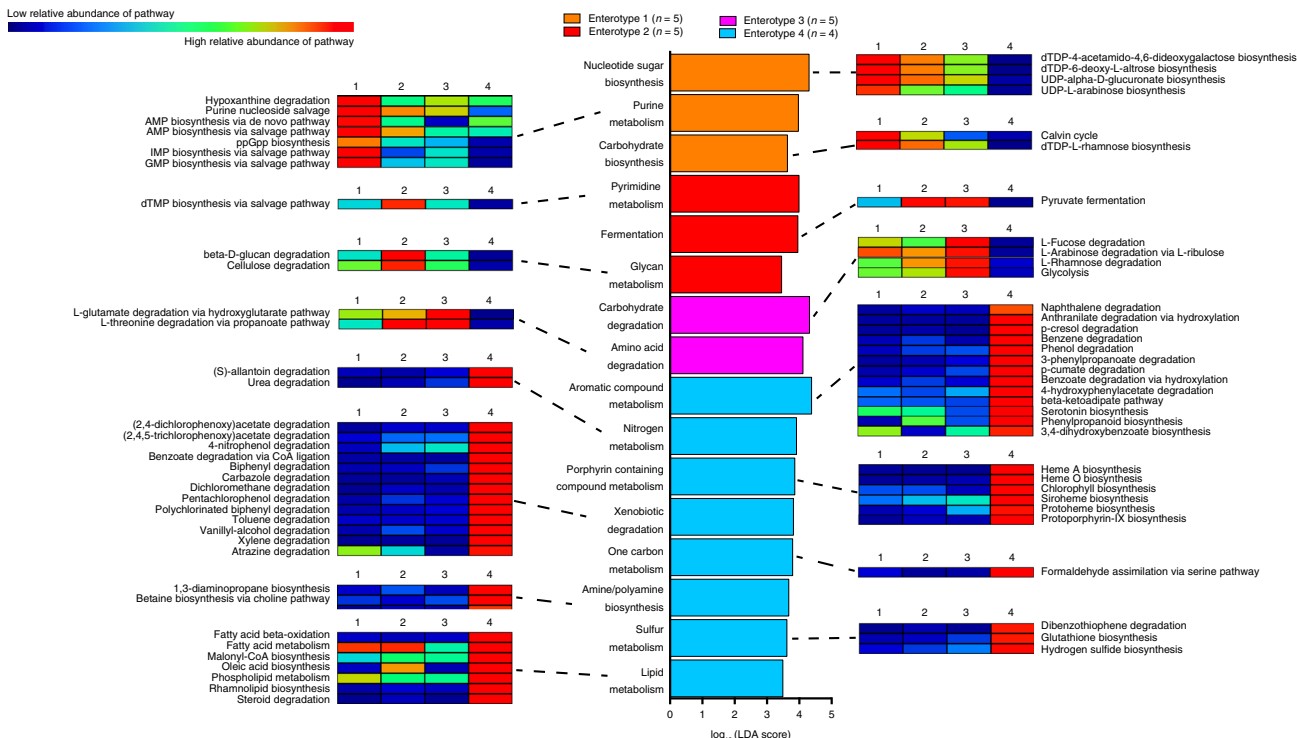

**Fig. 7** Shotgun metagenomic sequencing reveals differences in functional microbial pathways. Bar chart showing the log-transformed LDA scores of MetaCyc functional superpathways (identified by HUMAnN2 analysis) identified by LEfSe analysis as associated with each WLG enterotype. Heatmaps show the average relative abundance of functional pathways within each superpathway that were also determined by LEfSe to be associated with each WLG enterotype

suggest that high fiber fallback/staple foods may be important determinants of *Treponema* relative abundance and enterotype 2 membership.

In contrast, the top superpathways enriched in *Prevotella*-abundant WLG enterotype 3, which was associated with dry months and the HF period, were carbohydrate degradation and amino acid degradation (Fig. 7 and Supplementary Data 6). The strongest pathway biomarker for carbohydrate degradation was L-fucose degradation. Fucose is not a common constituent of plant polysaccharides, but is an important component of mucus glycans[34]. Therefore, we compared gene content along the entire pathway of fucose degradation. Genes for fucosidase, L-fucose:H + symporter permease (fucP), L-fucose isomerase (fucI), and L-fuculose-phosphate aldolase (fucA) were all enriched in metagenomes of enterotype 3 samples compared to samples from each of the other enterotypes, while L-fuculokinase (fucK) was also enriched in enterotype 1 samples (Fig. 8f). Thus, WLG *Prevotella*-abundant enterotype 3 may be associated with enrichment in bacteria suited for growth on intestinal mucus glycans. It has been shown in mice that fiber deprivation promotes growth and activity of colonic mucus-degrading bacteria[35]. In addition to fucose, O-linked and hybrid N-linked mucus glycans are made up of N-acetylglucosamine, N-acetylgalactosamine, N-acetylneuraminic acid (sialic acid), galactose, and mannose[34]. Enterotype 3 was enriched in a variety of important genes involved in mucus degradation, including alpha- and beta-N-acetylglucosaminidases, alpha-N-acetylgalactosaminidase, beta-galactosidase, and beta-glucuronidase (Supplementary Fig. 13a–g and Supplementary Note 6).

The top functional superpathways distinguishing the rare WLG *Solibacillus/Staphylococcus*-abundant enterotype 4 samples from the other three WLG enterotypes were aromatic compound, nitrogen, and porphyrin containing compound metabolism and xenobiotic degradation (Fig. 7). The majority of enterotype 4 enriched pathways were associated with superpathways of aromatic compound metabolism and xenobiotic degradation (Fig. 7 and Supplementary Data 6). See Fig. 7, Supplementary Note 6, and Supplementary Data 6 for additional information on metabolic functions associated with WLG enterotype 4.

As the relative abundance of *Treponema* and *Prevotella* were found to be associated with seasonal fluctuations in diet, we also compared metabolic pathways that most distinguished fecal samples dominated by these two taxa. As was the case when *Prevotella*-abundant enterotype 3 samples were compared against all other enterotypes, L-fucose degradation remained one of the top pathways discriminating enterotype 3 samples from *Treponema*-abundant enterotype 2 samples (Supplementary Fig. 14a and Supplementary Data 7). However, in the two-way comparison of seasonally fluctuant enterotype 2 and enterotype 3 samples, the top superpathway enriched in *Treponema*-abundant enterotype 2 samples was one carbon metabolism (Supplementary Data 7), consisting of four significantly enriched methanogenic pathways that are typically associated with archaea (Supplementary Fig. 14a and Supplementary Data 7). The methanogenic pathway for coenzyme B-coenzyme M hetero-disulfide reduction was also enriched in enterotype 2 samples, as was the biosynthetic pathway for L-pyrrolysine, which is found almost exclusively in methyltransferases from methanogens utilizing pathways of mono-, di- and tri-methylamine[36] (Supplementary Data 7).

**Association of methanogenic archaea with WLG enterotype 2.** Based on metagenomic sequencing from WLG fecal samples, the overwhelming majority of organismal relative abundance was bacterial (98.8% on average [range: 96.9–99.8%]). The remainder

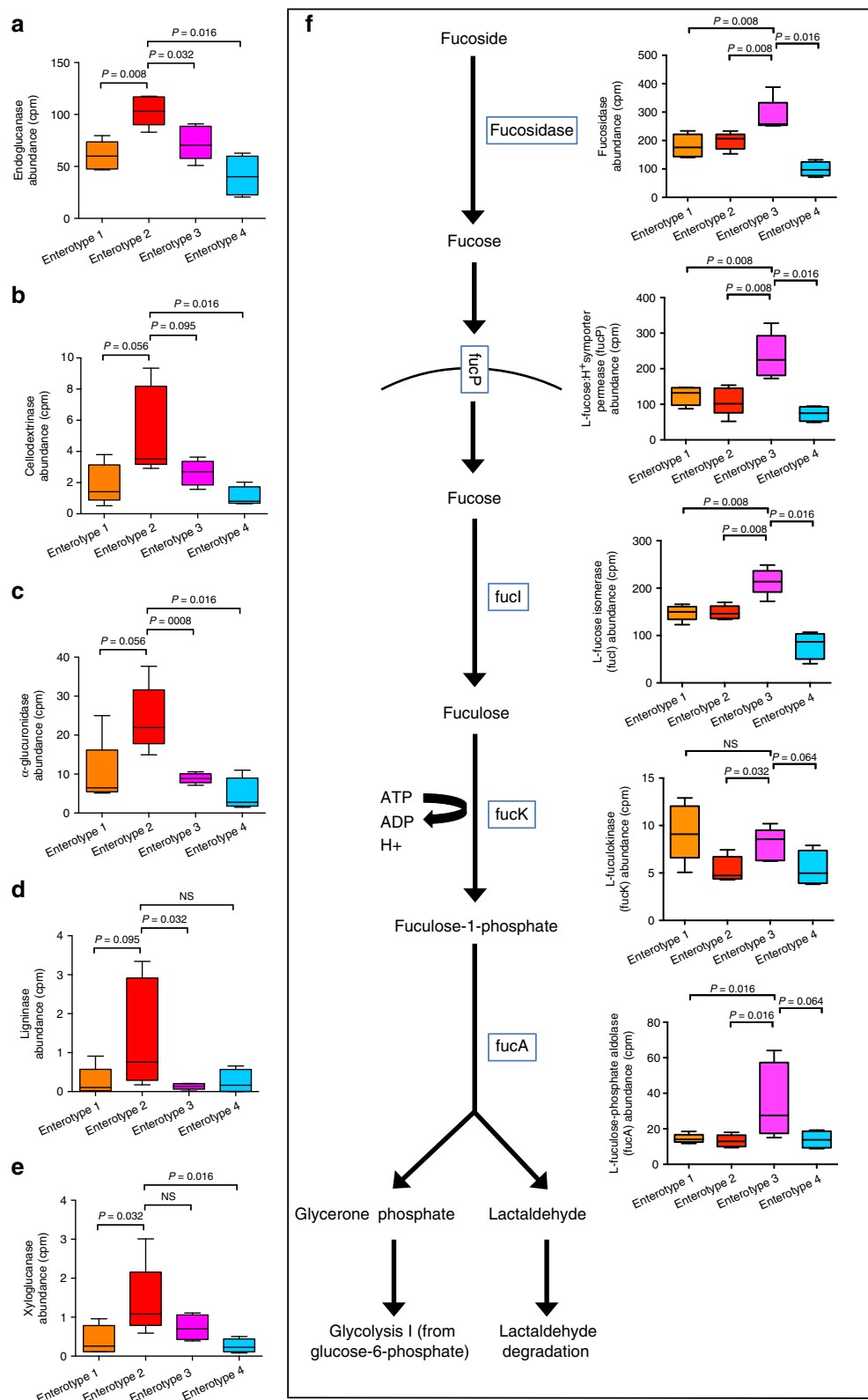

**Fig. 8** Genes coding for plant fiber degrading enzymes are enriched in the *Treponema*-abundant, LF-associated WLG enterotype 2, while genes encoding enzymes involved in fucose degradation are elevated in the *Prevotella*-abundant enterotype 3. Box-and-whiskers plots showing the relative abundance (in counts per million sequences, cpm) of **a** endoglucanase, **b** cellodextrinase, **c** alpha-glucuronidase, **d** ligninase, and **e** xyloglucanase (identified by HUMAnN2 analysis) in WLG enterotypes 1 ($n = 5$), 2 ($n = 5$), 3 ($n = 5$) and 4 ($n = 4$). **f** Schematic of the pathway for fucose degradation, along with box-and-whiskers plots showing the relative abundance (in counts per million sequences, cpm) of genes (identified by HUMAnN2 analysis) encoding the enzymes or transporter in samples from WLG enterotypes 1 ($n = 5$), 2 ($n = 5$), 3 ($n = 5$) and 4 ($n = 4$). Two-tailed *P*-values from Mann–Whitney tests are indicated. NS not significant at $\alpha = 0.05$

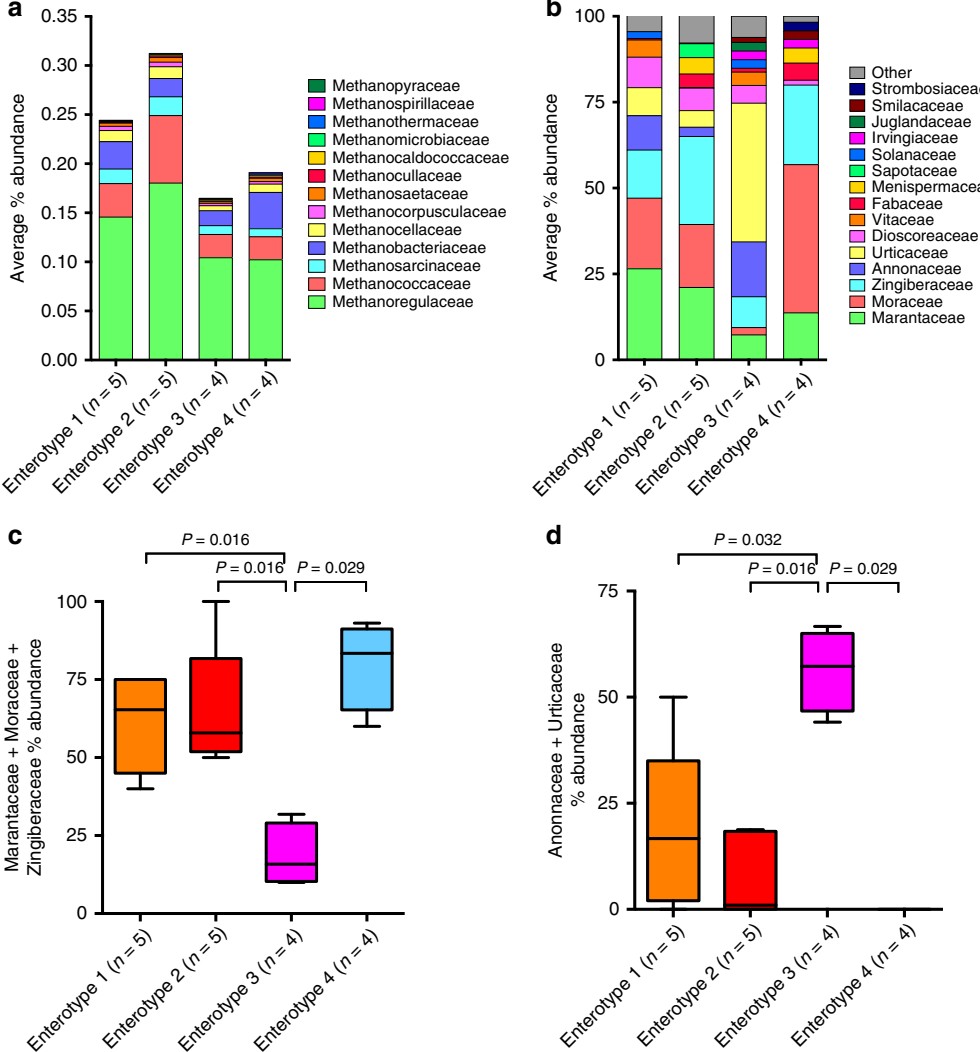

**Fig. 9** Shotgun metagenomic sequencing reveals differences in methanogenic archaea and fecal dietary plant taxa among WLG enterotypes. Stacked bar charts showing the average percent abundance of **a** methanogenic archaeal families (identified by MetaPhlAn2) and **b** plant families (identified by a plant marker gene approach) in each WLG enterotype. **c**, **d** Box-and-whiskers plots showing the distribution of cumulative plant family percent abundance in each of the WLG enterotypes. Two-tailed *P*-values from Mann–Whitney tests are indicated in **c**, **d**

was archaea (1.2% on average [range: 0.2–3.1%]). A recent 16S rRNA gene-based survey of fecal archaea found a high diversity of archaea in great apes, including gorillas. OTUs from only two orders of methanogenic archaea were detected, along with fewer OTUs associated with the phylum Thaumarchaeota[37]. Our unbiased metagenomic approach with marker gene analysis (MetaPhlAn2) revealed representatives from 15 orders of archaea in WLGs (Supplementary Data 8). Methanogenic archaea were highly abundant and diverse, represented by 13 archaeal families (Fig. 9a). The largest difference in relative abundance of methanogenic archaea was apparent between the two seasonally fluctuating enterotypes. In fact, enterotype 2 samples had higher relative abundance of total methanogenic archaea, as well as individual families of methanogenic archaea (Supplementary Fig. 14b–g). The predominant human gut-associated methanogenic archaeal species from the order Methanobacteriales, *Methanobrevibacter smithii*, and *Methanosphaera stadtmanae*, were present at high prevalence but not at high-relative abundance in WLGs. Our findings demonstrating enrichment in methanogenic metabolic pathways and higher relative abundance of methanogenic archaea in enterotype 2 samples compared to enterotype 3 samples suggest that methanogenesis carried out

by archaea may be a prominent feature associated with high *Treponema* relative abundance. Enterotype-specific *Methanobrevibacter* enrichment in humans has been associated with diets high in carbohydrates[6, 38].

**Evaluation of WLG diets from fecal metagenomics.** As our study examines free ranging wild gorilla populations, we have no direct observation information on diet. Therefore, we evaluated fecal metagenomic plant content from representative samples in each WLG enterotype. Using a plant marker gene approach[39], we identified 32 families of plants across samples. Plants from the families *Marantaceae*, *Moraceae*, and *Zingiberaceae*, which are staple or fallback foods of WLGs at nearly every study site for which WLG diet has been examined across West-Central Africa[40], accounted for nearly 60% of all identified sequences (Supplementary Data 9). Most *Marantaceae* marker sequences classified at genus- and species-levels were classified as *Haumania* sp. and *Megaphrynium macrostachyum*; *Moraceae* marker sequences were classified as *Ficus*, *Milicia* and *Morus*; and *Zingiberaceae* sequences were classified as *Aframomum* spp. These plant taxa are prominent components of WLG diets[40], suggesting

that this metagenomic-based analysis captured many dominant plant taxa in WLG diets.

Interestingly, frugivory-associated *Prevotella*-abundant enterotype 3 samples had significantly lower relative abundance of these staple plants than WLGs from the other enterotypes (Fig. 9b–c). Instead, *Annonaceae* and *Urticaceae* were the most abundant plant families detected in enterotype 3 samples, and their cumulative relative abundance was higher in enterotype 3 samples compared to the other enterotypes (Fig. 9b, d). *Duguetia* spp., *Duguetia confinis* and *Duguetia staudtii*, were the most abundant *Annonaceae* detected (Supplementary Data 9). Fruits and flowers of *Annonaceae* are consumed seasonally by WLGs[40]. *Duguetia confinis* and *Duguetia staudtii* are native to West Africa, including the Republic of the Congo, where they typically develop fruit from July–September, consistent with the frugivory period of WLGs in this area[41]. At the La Begique research site in southeast Cameroon, WLGs have been found to play an important role in fruit seed dispersal of *Duguetia staudtii*[42]. *Solibacillus/Staphylococcus*-abundant enterotype 4 samples had the highest relative abundance of *Moraceae* sequences, predominately from the genus *Ficus*. These results support the notion that differences in diet are the driving factor behind enterotypes, especially for samples that have a high-relative abundance of *Prevotella* from which staple/fallback plants represent a lower proportion of the diet (for additional discussion, see Supplementary Note 7).

## Discussion

Gorillas and chimpanzees are endangered species and our closest living relatives. Here we have investigated sympatric WLG and chimpanzee gut microbiota in comparison to other NHPs and extant humans. Our findings show that microbiotas of humans differ from those of wild African great apes. We further demonstrate temporal shifts in composition of WLG and chimpanzee microbiota across months and seasons defined by rainfall and fruit availability. These findings are consistent with a model wherein dietary plasticity in WLGs and likely other primates appears to be facilitated, at least in part, by a highly adaptable microbiota that reorganizes its composition in response to a shifting environment and seasonal feeding behaviors.

Our findings demonstrate that enterotype groupings can be highly dependent on shifting dietary patterns associated with seasonal availability of resources. This is consistent with previous studies that have demonstrated enterotype switching in individual habituated chimpanzees and, more recently, in baboons and gorillas[13–15]. In humans, it has been suggested that enterotypes are representative of "long-term dietary patterns"[8]. Though many humans enjoy a highly diverse and fluctuating diet, whether and how frequently dietary changes can lead to enterotype switching is unclear.

The relative abundances of bacterial taxa that define enterotype clusters are not only fluctuant but also appear to exist as relatively smooth gradients across a population rather than as bi- or multimodal distributions. Further, though these taxa have both high and variable relative abundances across a population, variations in their relative abundances that drive cluster separation are not necessarily reflective of variations in underlying microbial communities[43]. While there has been some controversy surrounding enterotyping, our findings are in agreement with the position that despite limitations of enterotyping, it can provide useful information on some aspects of community variation across a population and underlying factors driving such variation (here seasonality), particularly when combined with other methods[10]. However, our findings also suggest that caution should be applied when interpreting enterotyping results as enterotype assignment

within an individual or enterotype prevalence within a population may be dependent on factors such as seasonality and diet.

Only one of the four WLG enterotypes, the *Prevotella* enterotype, was shared among WLGs, chimpanzees, and humans, although deeper sampling of chimpanzees might reveal additional enterotypes. Humans from industrialized cultures, such as the US, have largely lost this enterotype in favor of the *Bacteroides* enterotype, while *Prevotella* remains dominant in rural communities around the world[44, 45]. *Prevotella* enterotypes are associated with plant-based diets high in carbohydrates[8]. Their scarcity in humans in industrialized cultures is likely a reflection of westernized diets high in protein and animal fat that emerged from our domestication and husbandry of animals ~10,000 years ago[46]. Indeed, diets consumed by industrialized nations today are widely disparate from ancestral plant-based diets to which anthropoids have adapted over millions of years[47]. *Prevotella* enterotype prevalence in WLGs was significantly elevated in dry months, and higher *Prevotella* relative abundance was associated with dry season 1 and HF, with similar results found in sympatric chimpanzees. This suggests that short-term dietary changes can influence *Prevotella* relative abundance and hence enterotype clustering at least in WLGs and chimpanzees. A recent study by Smits et al.[1], provides the first evidence for seasonal changes in microbiomes of human Hadza hunter-gatherers from Tanzania, wherein *Prevotellaceae* and *Spirochaetaceae* were found to be two of the most seasonally variable taxa, and *Prevotellaceae* declined in the wet season[1]. Concordance of these findings with ours provides evidence for a relationship between seasonal fluctuations and gut microbiome fluctuations that predates the emergence of Homininae.

*Prevotella* spp. are able to degrade hemicellulose, pectin, and simple carbohydrates, such as those found in fruits[48]. Based on bacterial metagenomic pathway analysis, WLG *Prevotella* enterotype was associated with carbohydrate degradation. However, ripe fruits are typically rich in soluble sugars that are readily digested and absorbed in the small intestine, and lower in fibers that reach the large intestine where the majority of bacterial fermentation occurs. We found enrichment of plant markers for fruit-bearing *Annonaceae*, as well as *Urticaceae*, in WLG *Prevotella*-abundant enterotype 3 samples and deficiency in plant markers for high-fiber staple and fallback foods. Thus, the frugivory period in WLGs may represent a period of relative fiber deficiency, during which the microbiome may alter its composition to favor a mucolytic phenotype. Indeed, our findings suggest enrichment for pathways and genes associated with bacterial growth on mucus glycans in the WLG *Prevotella* enterotype. The dichotomy between a microbiome with enhanced mucin degrading capacity and a microbiome enriched in plant fiber degrading capacity is evident in comparisons of human microbiomes from industrialized countries and hunter-gatherers[1]. However, our finding in WLGs is in opposition to our current understanding of *Prevotella* variance observed in humans, where high-relative abundance is associated with plant-based diets, presumably high in plant fibers, suggesting that our understanding of *Prevotella* variance in humans is incomplete and/or the metabolic functions of *Prevotella* spp. in WLGs are widely disparate from human-associated *Prevotella* spp. (see Supplementary Discussion).

*Treponema* relative abundance fluctuates seasonally and defines a distinct, seasonally responsive enterotype in WLGs. While *Treponema* prevalence is high in chimpanzees, relative abundance is low compared to WLGs, possibly reflecting the persistent ripe fruit specialization of chimpanzees[27]. Chimpanzees instead have a different Spirochaete-associated enterotype, defined by *Sphaerochaeta*, which, similar to WLGs, appears to fluctuate seasonally. Thus, microbiomes of primates, including

WLGs, may have some inherent specificity, allowing each primate to carefully balance dietary activity budgets with energy acquisition[20]. While prevalence and mean *Treponema* relative abundance in the US and Mongolian humans evaluated in this study are very low, there are a few known human populations that harbor substantial *Treponema*, including BaAka hunter-gatherers from the Central African Republic[49], Matses people from the Peruvian Amazon[50], Tunapuco people from the Andean highlands[50], and Hadza hunter-gathers from Tanzania[44], whose diets are high in fiber-rich plant foods. Thus, a greater understanding is needed to fully appreciate factors that have contributed to and consequences of the absence or loss of Spirochaetes from some human populations around the world, including westernized societies.

In support of a role for *Treponema* in plant fiber digestion in WLGs, *Treponema* abundance and the *Treponema* enterotype were significantly associated with wet periods, as well as the LF season, when WLGs rely heavily on fiber-rich fallback foods. Metagenomes of WLG *Treponema* enterotype samples were enriched in pathways and genes involved in fermentation and the degradation of plant fibers, including cellulose, hemicellulose, lignin, and xyloglucans. However, bacterial fermentation produces large amounts of hydrogen gas ($H_2$), which inhibits fermentative processes. Thus, for efficient fermentation to occur, a balance must be achieved between $H_2$-producing fermentative bacteria (hydrogenogenic) and $H_2$-consuming (hydrogenotrophic) microbes[34]. In WLGs, bacterial fermentation of plant fibers may be optimized via syntrophic relationships with $H_2$-scavenging methanogenic archaea, which were highly diverse in WLGs and reached the highest relative abundance in *Treponema* enterotype samples. Gut microbiomes of humans harbor relatively low abundance and diversity of hydrogenotrophic archaea compared to great apes[37], which could limit our capacity to efficiently ferment plant fibers. Together, these results suggest that seasonal fluctuation in *Treponema* and *Prevotella* relative abundance and enterotype distribution in WLGs may reflect oscillations in colonic capacity for plant fiber fermentation, with *Treponema* and fermentation dominating when fiber intake is maximized (i.e., the LF period) and *Prevotella* and colonic mucolysis dominating when fiber intake is minimized (i.e., the HF period)[40].

The remaining two enterotypes were defined by undercharacterized bacterial genera (Chloroflexi genus *SHD-231* and *Solibacillus/Staphylococcus*). *SHD-231* enterotype samples fell between seasonally fluctuant *Treponema* and *Prevotella* enterotypes along PC1, based on PCoA analysis, and *SHD-231* enterotype samples collected during the HF season were enriched in HF-associated bacterial taxa (including *Prevotella*), while *SHD-231* enterotype samples collected during the LF period were enriched in *Treponema*. Together these findings suggest that the *SHD-231* enterotype may represent the gradient between extremes of seasonal fluctuation in *Prevotella* and *Treponema* relative abundance, similar to gradients reported in humans between extremes of *Prevotella* and *Bacteroides*[22]. On the other hand, WLGs harboring the *Solibacillus-Staphylococcus* enterotype may reflect atypical feeding behavior in a small number of individuals (see Supplementary Discussion).

Early primates evolved in an arboreal environment much like those now inhabited by extant apes, where their survival was dependent on adaptation to dietary challenges. Such dietary pressures likely had large impacts on evolutionary trajectories of primates[47, 51]. Major events in hominin evolution have been ascribed to seasonality, including periods of moist-dry variability[51]. In this respect, seasonality of the tropics and feeding behaviors that extant NHPs exploit to adapt to seasonal shifts in resources may serve to expand our understanding of environmental and behavioral factors that drove human evolution[52]. While seasonality in diet is apparent in many extant primates, including isolated human populations, WLG may represent an extreme model for seasonal dietary shifts. As NHPs favor a plant-based diet, it may seem surprising that apes and humans do not encode enzymes required for efficient degradation of complex carbohydrates and plant fiber. However, these enzymes are encoded by intestinal bacterial symbionts with which primates co-evolved. The microbiome also plays a vital role in mammalian physiology, and thus, the genetic information encoded by symbiotic microbes is a requisite for host fitness and the foundation of the hologenomic theory of evolution[53, 54]. When, in the course of human evolution, rapid environmental changes occurred, genomic changes to hominins may not have been sufficiently rapid to maintain fitness and survival. However, changes in symbiotic microbes in response to external stimuli, including diet, can occur over days or months rather than over generations[8]. In this sense, the fitness of a species may be viewed in terms of an inherent level of adaptability that is reliant on the composition and malleability of the microbiome and its capacity to provide animals with dietary derived energy and nutrients.

Much can be gained from studies of wild animal microbiomes. From a biocentric perspective, and perhaps of greatest urgency, health monitoring in endangered wild primate populations is typically accomplished indirectly by population surveys or with invasive sampling. As microbiome studies have changed our collective thinking about the role of microbes in human health, defining microbial signatures and their inextricable relationship to the environment in wild animal populations provides a veritable microbial snapshot in time that can inform future microbial studies aimed at monitoring the health of threatened populations in response to potentially deleterious forces, such as human encroachment, habitat destruction, disease, and climate change. Indeed, impacts of these forces on microbiomes, health, and environments of NHPs is already evident[18, 55–57].

From an anthropocentric viewpoint, the microbiomes of wild primates may hold keys to both our evolutionary past, as well as our future health. As we continue to eliminate diversity of our contemporary microbiome with the use of antimicrobials and westernized diets, an era may arrive when we must search for probiotics outside of our current human-derived repertoire. The undercharacterized, highly diverse group of microbes inhabiting the intestinal tracts of our closest living relatives may possess an untapped potential to cross the species barrier into *Homo sapiens* and reset the ancestral fermentative, immunostimulatory, and metabolic capacity that modern humans may no longer harbor.

## Methods

**Study population and data collection.** Fecal samples ($n = 166$) from free-ranging wild western lowland gorillas (*Gorilla gorilla gorilla*) and central chimpanzees (*Pan troglodytes troglodytes*) were collected during 35 reconnaissance surveys conducted by the Wildlife Conservation Society (WCS) from 2006 to 2010 across a region spanning over 5500 km$^2$, located to the east of Odzala-Kokoua National Park in the Republic of the Congo. Non-invasive fecal sampling was undertaken with permission of the Congolese Ministry of Scientific Research (permit Nos. 003/MRS/DGRST/DMAST and 014/MRS/DGRST/DMAST) and in compliance with the American Society of Primatologists' Principles for the Ethical Treatment of NHPs. The species origin of fecal source was provisionally characterized by evaluating fecal morphology, which includes odor, color, and amount of fiber, as previously described[58] and later confirmed with genetic analyses (see below). Freshness was determined by physical appearance using previously described methods[59]. GPS coordinates were recorded for each fecal sample upon collection. Samples were immediately preserved in RNAlater® and were subsequently shipped to the Bronx Zoo, New York and the Center for Infection and Immunity, Columbia University, New York, where samples were maintained at −80 °C until processing.

**Nucleic acid extraction.** DNA was extracted from all 166 fecal samples collected by WCS using a modified protocol of the QIAmp DNA Stool Mini Kit (Qiagen Inc; Valencia CA, USA). The ease with which samples can be contaminated with

bacterial DNA is well recognized. Thus, we have applied every precaution to prevent and monitor any introduction of bacterial DNA from nucleic acid extractions to downstream PCR reactions. For each set of DNA extractions, two empty 2 mL tubes were taken through every step of the extraction to serve as negative controls for downstream 16S rDNA PCR. Such extraction reagent controls are essential to confirm that contaminating bacterial 16S rDNA has not been introduced from extraction reagents. A total of 22 extraction controls were prepared from 12 sets of extractions. Prior to extraction, all tubes, columns, 0.1 mm glass beads and 0.5 mm glass beads (MoBio Laboratories) were UV irradiated twice at a distance of 1 inch from UV bulbs and at a setting of $3000 \times 100 \ \mu J/cm^2$ in a SpectroLinker XL-1500 UV crosslinker (Spectronics Corporation). All kit extraction reagents (liquid) were aliquoted into 2 mL tubes in a UV hood at a volume not exceeding 1 ml and were UV irradiated as above. For each fecal sample, 220 mg of feces was centrifuged at 6000 r.p.m. for 5 min to pellet fecal material and bacteria, RNAlater was removed, and fecal pellets were resuspended in Buffer ASL (QIAgen, 1.4 mL) and transferred to UV irradiated 2 mL Safe-Lock tubes (Eppendorf) containing UV irradiated beads: one 5 mm steel bead (QIAgen) and 0.1 mm and 0.5 mm glass beads (MoBio Laboratories). In order to optimally lyse Gram-positive bacteria, samples were further disrupted with bead beating in a TissueLyser (QIAgen) for 5 min at 30 Hz and incubated for 5 min at 95 °C. The remaining steps for extraction followed the manufacturer's protocol and were performed in a UV hood. DNA concentration and purity were determined using a NanoDrop ND-100 spectrophotometer (NanoDrop Technnologies, Wilmington, DE) and stored at −80 °C.

**Genotyping, microsatellite analysis, and sex determination.** Genomic DNA from fecal samples was genotyped by amplifying the nearly 500 bp D-loop region of mitochondrial DNA to confirm species origin as described previously[60]. A subset of samples failed to yield efficient amplification of the D-loop region. For these samples, a 386 bp region of the mitochondrial 12S gene was amplified to confirm species origin as previously described[61]. The PCR amplification products obtained by both methods were subjected to Sanger sequencing, and sequences were evaluated using the Geneious v5.6.3 software (Biomatters; Auckland, New Zealand).

In order to avoid oversampling from the same individuals, all samples were subjected to microsatellite analyses via capillary electrophoresis by targeting 8 microsatellite loci: D18S536, D4S243, D10S676, D9S922, D2S1326, D2S1333, D4S1627, and D9S905[60, 62]. All microsatellite loci were analyzed twice, and those with apparent homozygosity were repeated seven times to exclude allelic dropout. Microsatellite loci were analyzed with Peak Scanner v1.0 (Applied Biosystems). Where more than one fecal sample was determined to have originated from a single individual, only one sample was included for downstream analysis. Collectively, these analyses conclusively identified 87 individual western lowland gorillas (WLGs) and 18 sympatric central chimpanzees (unbiased probability of identity «0.001, calculated using GIMLET[63]) (see Supplementary Data 1). All 87 WLG and 18 chimpanzee samples were subjected to 454 pyrosequencing of bacterial 16S rDNA.

For gender determination, a region of the amelogenin gene was amplified by PCR using primers AMEL-F212 (5′-ACCTCATCCTGGGCACCCTGG-3′) and a fluorescent NED labeled AMEL-R212 (NED- 5′-AGGCTTGAGGCCAACCATCAG-3′) that generate a 212 bp fragment from the X chromosome and a 218 bp fragment from the Y chromosome[62]. Size discrimination for amelogenin-based gender typing was determined via capillary electrophoresis. Samples for which gender could not be successfully determined based on the amelogenin gene were further evaluated by size discrimination on agarose gels of the ubiquitously transcribed tetratricopeptide repeat protein gene (UTX/UTY) as previously described[64]. The amelogenin and UTX/UTY assays allowed for gender determination for 82 of the 87 (94%) WLG fecal samples and 13/18 (72%) chimpanzee fecal samples. We do not have age information for any of the animals as these samples were collected from wild, non-habituated WLGs, and chimpanzees.

**Barcoded pyrosequencing of bacterial 16S rDNA.** Amplification of the V1–V3 region of bacterial 16S rDNA for pyrosequencing was performed on all 87 WLG and 18 chimpanzee samples (representing 87 individual gorillas and 18 chimpanzees, respectively, as determined by mitochondrial and microsatellite analysis) using 16S rDNA composite primers, consisting of FLX Titanium adapters, a sample barcode, and bacterial 16S rDNA-specific V1 (27 F) and V3 (534R) primer sequences as previously described[65]. Plate caps, 96-well PCR plates, 2 mL microcentrifuge tubes and Ultra Clean water (MOBIO) were UV irradiated in a SpectroLinker XL-1500 UV crosslinker ($3000 \times 100 \ \mu j/cm^2$) prior to PCR setup in a UV hood. Each 20 μL PCR reaction consisted of 1× Accuprime Buffer II, 0.75 units of Accuprime Taq DNA Polymerase High Fidelity (Life Technologies), 2.5 units of Sau3AI restriction enzyme, 200 nM of each primer and 100 ng of sample DNA. Before addition of primers, the PCR master mix was UV-irradiated to reduce downstream amplification of any potential contaminant DNA in reagents. Subsequently, the master mix was aliquoted into 96-well plates, and barcoded primers were added to each reaction. The 96-well plate was then incubated at 37 °C for 30 min to facilitate Sau3AI digestion of any remaining double-stranded DNA contaminants. After digestion, the plate was incubated on ice for 5 min, and sample DNA was then added to each reaction and immediately placed in the thermal

cycler. Extraction reagent controls and PCR reagent controls were included to control for any bacterial DNA contamination. The PCR cycling conditions were as follows: 95 °C for 5 min, 35 cycles of 95 °C for 20 s, 56 °C for 30 s and 72 °C for 5 min. All PCR products were run on 1% agarose gels stained with ethidium bromide, and products were purified using the QIAgen Gel Extraction kit. PCR products were further purified using Ampure magnetic purification beads (Beckman Coulter Genomics). Ampure purified products were quantified with the Quanti-iT PicoGreen dsDNA Assay Kit (Invitrogen). Equimolar ratios of each sample were combined to create DNA pools of barcoded libraries.

**Shotgun metagenomics.** Shotgun metagenomic sequencing was performed on 19 representative WLG samples. Samples were chosen from each enterotype by their separation from samples of other WLG enterotypes by PCoA based on BC, as well as their high-relative abundance of SHD-231 (n = 5; SHD-231 relative abundance avg. = 44.2%, std. dev. = 8.1%), Treponema (n = 5; Treponema relative abundance avg. = 43.5%, std. dev. = 9.8%), Prevotella (n = 5; Prevotella relative abundance avg. = 27.3%, std. dev. = 5.0%), or Solibacillus/Staphylococcus (n = 4; Solibacillus + Staphylococcus relative abundance avg. = 61.2%, std. dev. = 14.2%). For Illumina library preparation, genomic DNA was sheared to a total of 200-bp average fragment length using a Covaris E210 focused ultrasonicator. Sheared DNA was purified and used for Illumina library construction using the KAPA Hyper Prep kit (KK8504, Kapa Biosystems). Sequencing libraries were quantified using an Agilent Bioanalyzer 2100. Sequencing was carried out on the Illumina HiSeq 2500 platform (Illumina, San Diego, CA, USA), yielding ≈530 million single-end reads of 100 nucleotides in length (Average reads per sample = 28 million; range = 12–42 million). Raw shotgun metagenomic sequences were deposited in the Sequence Read Archive (SRA) under PRJNA382701.

**Bioinformatic analysis.** 16S rRNA gene sequences were analyzed using the open source software package QIIME (Quantitative Insights Into Microbial Ecology: http://qiime.org/) v1.7 and v1.8. Sequences from each run were de-multiplexed and quality filtered based on the following criteria: (1) length outside bounds of 200 and 1000 nucleotides, (2) number of ambiguous bases exceeds limit of six, (3) missing quality score, (4) mean quality score below a minimum of 25, (5) maximum homopolymer run exceeds a limit of six, (6) number of mismatches in the primer exceeds the limit of zero, (7) include sequences without a discernible reverse primer. Sequencing data were further denoised using flowgram clustering with QIIME's built in Denoiser. Reverse primers were removed using fine-tuned BLASTing, as well as positional criteria. In order to eliminate 454 inter-plate bias, quality of sequences was evaluated using the QIIME quality_scores_plot.py script. Differences in sequence quality were apparent at the 3′ end of sequences that exceeded 400 bp in length between different 454 runs. As this type of bias can inflate the number of OTUs and prevent downstream comparison between different 454 runs, we trimmed only sequences with length exceeding 400 bp to exactly 400 bp from each plate run. No plate effect was apparent based on alpha and beta diversity metrics after applying this sequence trimming criteria. Chimeric sequences were identified using usearch61, which uses a combination of de novo and reference-based chimera detection algorithms[66]. The 13.5 release of the Greengenes dataset was used as the reference dataset. Identified chimeric sequences were removed. De-multiplexed, trimmed, and quality-filtered sequences were deposited in the MG-RAST database (http://metagenomics.anl.gov) under the project ID 10912.

Open-reference OTU picking was carried out with usearch61_ref, using Greengenes as the reference dataset at a similarity threshold of 97% (roughly corresponding to species-level OTUs). Representative sequences from the OTUs were aligned to a pre-aligned database of sequences (the Greengenes core set) using PyNAST with quality thresholds set with a minimum sequence length of 150 nucleotides and a minimum percent identity of 75%. PyNAST alignment failures were investigated by blasting all sequences that failed to align. Taxonomies were assigned to OTUs using the RDP Classifier trained on the Greengenes 13.5 dataset. The 87 WLG and 18 chimpanzees were distributed across the sequential years 2008 (33 WLG and 6 chimpanzee individuals), 2009 (39 WLG and 7 chimpanzee individuals), and 2010 (15 WLG and 5 chimpanzee individuals). Alpha diversity metrics (including chao1, observed species, and Shannon) were calculated and rarefaction plotted to investigate differences between groups for diversity within samples based on the abundance of various taxa within a community. No differences in any of the alpha diversity metrics were observed among years of sampling (2008 vs. 2009 vs. 2010) for WLG and chimpanzee groups, suggesting that length of time for storage of older samples did not adversely impact diversity in our samples (Supplementary Fig. 15).

US human sequences from the Human Microbiome Project QIIME community profiling (HMPQCP) dataset were downloaded. Only stool samples sequenced over the V1–V3 region of bacterial 16S rDNA were selected for analysis (175 individual human stool samples). Bacterial 16S sequences from the Mongolian humans were obtained from MG-RAST (project no. 8437), while bacterial 16S sequences from the Old World monkeys were obtained from personal correspondence with Suleyman Yildirim (red-tailed guenons, black-and-white colobi, and red colobi[20]) and Martin Wu (baboons[15]). For the baboon dataset, samples from infant baboons were excluded. Sequences from all great ape, human, and Old World monkey datasets were combined, and QIIME was used to pick OTUs with usearch61 and

taxonomy was assigned using the RDP classifier trained on Greengenes 13.5 dataset as described above. Alpha diversity was calculated between these samples using chao1, observed species, and Shannon metrics. In order to compare bacterial communities based on their composition between individuals and groups, root Jensen-Shannon divergence, Bray–Curtis dissimilarity, and unweighted UniFrac were assessed and visualized with unsupervised principal coordinate analysis (PCoA) and distance histograms. The HMP, Mongolian, baboon, black-and-white colobus, red colobus, and red-tailed guenon datasets were generated using the same V1–V3 region primers as were used in our study of the WLG and sympatric chimpanzee microbiota, and all samples were sequenced on the same platform (Roche/454) to limit the impact of technical variation when comparing across studies. Different extraction methods were employed between studies, which may introduce minor artifacts. However, others have demonstrated that in comparisons among groups with large effect sizes (i.e., geographically distinct human groups or humans and other species), bacterial compositional differences outweigh technical variation[21, 67]. Furthermore, inter-individual subject variability is shown to outweigh the small proportion of variation associated with DNA extraction methods used to generate each of the datasets in our study, and the different extraction methods have been shown to have little impact on diversity metrics[68, 69].

For shotgun metagenomic sequences, host subtraction using Bowtie 2[70] was performed against the WLG, human, PhiX and mouse genomes to remove host and any contaminant sequences. The de-multiplexed reads were trimmed to remove primers and adaptors and filtered to exclude short reads and those with low complexity regions (Prinseq v0.20.2[71], minimum quality mean = 30, max Ns = 20). A total of 512 million reads remained in the dataset after host subtraction and quality filtering (Average reads per sample = 27 million; range = 12–40 million). HUMAnN2 v0.5.0 (http://huttenhower.sph.harvard.edu/humann2) was used with the UniPathway database for functional metagenomic profiling of filtered, host-subtracted metagenomic reads. Gene families were normalized to counts per million (cpm) prior to superpathway- and pathway-level annotation using the UniPathway database. LEfSe[25] was used to assess differences in superpathway and pathway functions (based on HUMAnN2) among samples from the four different WLG enterotypes and the two seasonally fluctuant enterotypes. MetaPhlAn2[72] was run on the host-subtracted, quality-filtered reads to assess archaeal taxonomic composition in each sample.

The identification of plant content in fecal samples was performed on shotgun metagenomic sequences using a previously described approach[39]. Briefly, following quality filtering and host subtraction, reads from WLG samples were separately aligned to three pre-built plant reference barcode databases (rbcL, matK, and trnL-F). MEGABLAST was used to select reads with a minimum of 98% identity and a minimum of 50 bp overlap with plant sequences in these databases. Alignments with incomplete overlaps were further excluded. Readsidentifier (v 1.0)[73] was then used to classify the remaining aligned reads to the lowest identifiable taxonomic levels in the databases. All sequence reads that met these alignment criteria in any of the three plant marker gene databases were included for downstream analyses.

**Satellite rainfall and vegetation analysis**. High-temporal resolution precipitation measurements of the satellite-borne Tropical Rainfall Measuring Mission (TRMM: http://trmm.gsfc.nasa.gov/) covering the period 1 January 1998–31 October 2013 were downloaded for the project from NASA (http://disc2.nascom.nasa.gov/daac-bin/OTF/HTTP_services.cgi?SERVICE=TRMM_ASCII&BBOX=-6.25,29.75,-6.00,30.00&TIME=1998-01-01T12:00:00%2F2013-10-31T12:00:00&FLAGS=3B42_V7_Daily,Time&-SHORTNAME=3B42_V7_Daily&VARIABLES=Rain). The data coverage extend from latitude 0.25–1.50 deg. N and longitude 15.25–16.50 deg. E. Three-hourly rainfall data from the 25 individual $0.25 \times 0.25$ degree latitude-longitude grid cells (~$27 \times 27$ km) enclosed within the $1.25 \times 1.25$ degree box were averaged and used as the basis for computing daily rainfall means and other quantities using Microsoft Excel software.

Images generated by monthly Terra/MODIS satellite data based on the enhanced vegetation indices for the region of sample collection (0.42–1.50 deg. N and 15.33–16.52 deg. E) from 2008 to 2010 were downloaded from NASA's Earth Observing System Data and Information System (http://reverb.echo.nasa.gov).

**Statistical analysis**. Box-and-whiskers plots and Mann–Whitney U-test: All box-and-whiskers plots shown in the main and supplementary figures represent the interquartile ranges (25th through 75th percentiles, boxes), medians (50th per-centiles, bars within the boxes), the 10th and 90th percentiles (whiskers above and below the boxes), and outliers beyond the whiskers (closed circles). All statistics based on data presented in box-and-whiskers plots are two-tailed P-values derived from Mann–Whitney U-tests.

LEfSe analysis: Differences in the relative abundance of bacteria at all taxonomic levels, as well as microbial functional metagenomic superpathways and pathways, were determined with linear discriminant analysis effect size (LEfSe), which couples tests of statistical significance with measures of effect size to rank the relevance of differentially abundant taxa[25]. Thus, the Kruskal–Wallis (or Mann–Whitney U-test) identifies taxa that are significantly different in relative abundance among different classes, and the linear discriminant analysis (LDA) identifies the effect size with which these taxa differentiate the classes. For each LEfSe analysis, an alpha value of 0.05 for the Kruskal–Wallis test and a

log-transformed LDA score of 2.0 were used as thresholds for significance. LEfSe analyses were used to evaluate differences among the fecal microbiota of the 87 WLG, 18 chimpanzee, 104 baboon, 3 red-tailed guenon, 3 black-and-white colobus, 3 red colobus, 320 Mongolian human, and 175 US human samples; differences within the fecal microbiota of the gorillas, chimpanzees, baboons, and humans among different enterotypes (see below); differences in the fecal microbiota of the gorillas among different months; differences in the fecal microbiota of the gorillas and chimpanzees among different seasons; and differences in the fecal microbiota of male and female gorillas. For the LEfSe analysis on the monthly variations in the gorilla fecal microbiota, samples collected in January or February, July or August, and September or October were consolidated into three classes (Jan–Feb, Jul–Aug, and Sept–Oct) to accommodate the small sample sizes in February, July, September, and October. Monthly LEfSe analysis was not performed on the 18 chimpanzee samples, as there were too few samples per month. LEfSe analysis on seasonal variations in the gorilla fecal microbiota was conducted by comparing bacterial relative abundance among the two wet seasons and two dry seasons (determined by total rainfall in the month of sample collection from over a 10-year period [2001–2010]), as well as between more discrete wet and dry periods (determined by the average daily rainfall over the 30 days prior to sample collection, >4.23 mm day$^{-1}$ mm for wet months and <4.23 mm day$^{-1}$ for dry months) and between seasons of high-fruit availability (HF) and low-fruit availability (LF) (July–August vs. other months, respectively). Chimpanzee samples were also analyzed for seasonal variations in the microbiota using the 30-day average daily rainfall wet/dry and HF/LF classifications. As with the monthly analysis, there were too few chimpanzee samples per class to examine differences among the two wet and two dry seasons determined by decadal rainfall patterns. The 4.23 mm day$^{-1}$ cutoff (equivalent to 127 mm total rainfall over a 30 day period) for wet and dry months was selected based on the highest upper limit of the standard error of the mean total rainfall for a month in either of the two dry seasons (August, 126 mm total or 4.07 mm day$^{-1}$), which did not overlap with the standard error intervals for total rainfall in any of the months in either wet season (see Fig. 4a). By dividing samples into wet and dry months using this 4.23 mm day$^{-1}$ cutoff, for both WLG and chimpanzee samples, there were 0.77 mm day$^{-1}$ and 0.76 mm day$^{-1}$ differences, respectively, in average daily rainfall over the 30 days prior to sample collection between the sample with the lowest average daily rainfall in the wet months group and the sample with the highest average daily rainfall in the dry months group (see Supplementary Fig. 9d).

ANCOVA analysis: ANCOVA analyses were conducted to confirm associations identified by LEfSe between the relative abundance of bacterial taxa and month and/or season of sample collection, adjusting for the effects of potentially confounding variables. As LEfSe has been shown to effectively reduce the false discovery rate, it serves as a gatekeeping procedure (hierarchical or fixed-sequence test procedure)[74, 75]. Therefore, for any given LEfSe-defined bacterial taxa, the probability of a false rejection of the null hypothesis and the false discovery rate can only decrease when follow-on adjusted analyses are applied to the LEfSe-defined bacterial taxa. By definition, the original biomarker cannot be a true discovery if it is an artifact of confounding. As these follow-on adjusted ANCOVA analyses neither increase the familywise type I error rate nor the FDR, additional multiple testing corrections were not applied. In the first set of ANCOVA analyses, month of sample collection (with June as the base category) was evaluated as a significant predictor of the Box-Cox-transformed relative abundance of all sampling month-associated bacterial taxa identified by LEfSe, adjusting for the effects of sex (male as the base category), year of sample collection (2010 as the base category), latitude of collection site, and longitude of collection site. In the second set of ANCOVA analyses, season of sample collection (determined by average total rainfall in the month of sample collection; wet season 2 as the base category) was evaluated as a significant predictor of the Box-Cox-transformed relative abundance of all seasonal-associated bacterial taxa identified by LEfSe, adjusting for the effects of sex (male as the base category), year of sample collection (2010 as the base category), latitude of collection site, and longitude of collection site. In the third set of ANCOVA analyses, season of sample collection (determined by fruit availability; LF as the base category) was evaluated as a significant predictor of the Box-Cox-transformed relative abundance of all sampling month-associated bacterial taxa identified by LEfSe, adjusting for the effects of sex (male as the base category), year of sample collection (2010 as the base category), latitude of collection site, and longitude of collection site. In the final set of ANCOVA analyses, average daily rainfall over the 30 days prior to sample collection was evaluated as a significant predictor of the Box-Cox-transformed relative abundance of all seasonal-associated bacterial taxa identified by LEfSe (wet vs. dry months), adjusting for the effects of sex (male as the base category), year of sample collection (2010 as the base category), latitude of collection site, and longitude of collection site. All Box-Cox transformations were performed in XLSTAT version 2014.3.07 based on an optimized lambda for each taxon. ANCOVA analyses were performed with SPSS version 21 and XLSTAT version 2014.3.07.

Identification of enterotype: Enterotypes were identified in both the WLG, chimpanzee, baboon, US human, and Mongolian human datasets using the methods described by Arumugam et al.[6] (http://enterotype.embl.de/enterotypes.html). For all datasets, root Jensen-Shannon divergence (rJSD) and Bray–Curtis (BC) dissimilarity were calculated from genus-level relative abundance profiles in R as described by Arumugam et al. and Koren et al.[6, 76]. Only taxa that could be classified at the genus level was included in the clustering analyses. The optimal

number of clusters ($k$) for each dataset was determined for both distance metrics using the Calinski-Harabasz (CH) index and the average silhouette width calculated using the clustersim and cluster packages, respectively, in R. Samples were assigned to $k$ clusters for each dataset and distance metric using the partitioning around medoids (PAM) algorithm implemented in the cluster package in R. Defining genera of each cluster, or enterotype, were identified using both between class analysis (BCA) implemented in the ade4 package[77] in R and LEfSe analysis. Unsupervised PCoA plots were generated using the ade4 package[77] in R. Significance of WLG sample clusters as determined by the PAM algorithm based on rJSD and BC dissimilarity was assessed by comparing within-cluster weighted UniFrac distance to between-cluster weighted UniFrac distance through the Monte Carlo label permutation test as implemented in QIIME with Bonferroni correction for multiple comparisons. Enterotyping was not performed on the red-tailed guenon, black-and-white colobus, or red colobus datasets as each dataset only contained three samples.

Evaluation of seasonal fluctuations in enterotypes: The relationship between season of WLG and chimpanzee sample collection and the enterotype to which each sample was assigned by PAM clustering based on rJSD was evaluated by $\chi^2$- and Fisher's exact tests. $\chi^2$-tests were used to evaluate differences in the proportions of all four WLG enterotypes in wet months/LF seasons vs. the proportions all four enterotypes in dry months/HF seasons, while Fisher's exact test was used to evaluate the differences between the proportion of WLG enterotype 2 in wet months/LF seasons vs. dry months/HF seasons, the proportion of WLG enterotype 3 in wet months/LF seasons vs. dry months/HF seasons, the proportion of chimpanzee enterotype 1 in wet months/LF seasons vs. dry months/HF seasons, and the proportion of chimpanzee enterotype 2 in wet months/LF seasons vs. dry months/HF seasons. As these tests were confirmatory for the findings in the seasonal LEfSe analyses, corrections for multiple comparisons were not applied.

**Phylogenetic analysis.** To examine the phylogenetic relationships between unique WLG and chimpanzee taxa and related taxa from other hosts, representative sequences from the top most abundant *Treponema*, *Sphaerochaeta*, SHD-231, *Solibacillus*, and *Staphylococcus* 97% OTUs in WLG samples (collectively accounting for >90% of all Spirochaete, Chloroflexi, *Solibacillus*, and *Staphylococcus* abundance in WLG samples), as well as the top most abundant *Sphaerochaeta* 97% OTUs in chimpanzee samples (collectively accounting for >90% of all Spirochaete abundance in WLG chimpanzee) were combined with sequences of the V1–V3 region of the 16S rRNA gene from GenBank that represent the top NCBI BLAST hits for each of the sequences, as well as additional sequences from GenBank used to give context to phylogenetic placement. Each of the sequence sets (Spirochaete, Chloroflexi, *Solibacillus*, and *Staphylococcus*) were aligned using MUSCLE[78], and bootstrapped (1000 replicates) maximum likelihood phylogenies using the general time reversible substitution model with a gamma distribution and invariant site correction for rate heterogeneity (GTR + i + Γ) were generated using MEGA7[79]. Phylogenies were annotated using the Interactive Tree of Life[80].

**Data availability.** De-multiplexed, trimmed, and quality-filtered bacterial 16S rRNA gene sequences from 454 pyrosequencing of WLG and chimpanzee samples are available from the MG-RAST database (http://metagenomics.anl.gov) under the project ID 10912. Raw shotgun metagenomic sequences from WLG samples are available from the Sequence Read Archive (SRA) under PRJNA382701. All additional relevant data are available in this article and its Supplementary Information files, or from the corresponding author upon request.

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

## Acknowledgements

The authors would like to thank Diane Doran-Sheehy and Beatrice Hahn for helpful discussion and assistance. This study was made possible by the National Institute of Health Centers of Excellence for Translational Research (grant no. U19AI109761), the United States Fish and the Wildlife Great Ape Conservation Fund (grant nos. 98210-5-G195, 98210-6-G107, 98210-7-G292, 98210-8-G643, 98210-0-G280), the Neu Foundation, and Mr. and Mrs. Bradley L. Goldberg.

## Author contributions

B.L.W., A.L.H., K.J.L., S.H.O., T.A.S., A.S., A.U.O., W.B.K., P.R., K.N.C., and W.I.L. contributed to study design. B.L.W., K.J.L., and M.C.-R. designed the experiments. B.L.W., A.L.H., K.J.L., and M.C.-R. performed the experiments. B.L.W., A.L.H., J.P., R.S., and C.G. designed analyses. B.L.W., A.L.H., K.J.L., M.C.-R., J.P., R.S., C.G., A.S., and W.I.L. analyzed the results. B.L.W., A.L.H., K.J.L., M.C.-R., S.H.O., A.S., T.A.S., A.U.O., W.B.K., P.R., K.N.C., and W.I.L. wrote the paper.

## Additional information

**Competing interests:** The authors declare no competing interests.

