## [Peer Review File · Nature Communications]

Reviewers' comments:

Reviewer #1 (Remarks to the Author):

The manuscript by Williams and colleagues presents some results from analysis of the gorilla microbiome using nearly 100 samples from the wild. The focus is on enterotypes, an assessment of co-occurring taxa.

The author have two main claims. First, they suggest that the enterotypes found in gorillas are remarkably different from those found in chimpanzees and humans. This does not appear consistent with the results of Moeller and colleagues, appearing a few days ago in PNAS, in which analysis of sequences from hundreds of chimpanzees, gorillas, and humans of varied origins suggested that human microbiomes lack diversity, and hence are quite different from those of our closest living relatives. It may be that the study by Williams et al reaches a different conclusion due to the use of a limited sampling of gorillas or an inappropriate comparative dataset or other methodological issues, such as the focus on enterotypes. However, the results are not presented in a phylogenetic framework that would aid in inferring which lineages have undergone particular changes since the common ancestors. Indeed, the authors should reassess their description of gorillas as 'vestiges of early hominids' (in the Introduction, page and line numbers are unfortunately absent) as present day gorillas are no more vestiges of early hominids than we are; all great ape lineages, including ourselves, have been evolving continuously since they shared common ancestors. It is also rather premature to state that 'the relationships between humans and their microbiota are no more remarkable or complex than those found in the rest of the animal kingdom' as it is a supposition without supporting data and a rather odd way to justify a comparative evolutionary analysis.

The second major claim of the paper is that western gorilla enterotypes vary with season; by correlating enterotype shifts with changes in rainfall and fruit availability the authors suggest a dynamic response of individuals to ecology. Unfortunately, this intriguing possibility, although plausible, is suggested by rather tenuous approximate associations- the 87 analyzed gorilla samples were collected over a period of three years, yielding very few samples per month, and rainy season data is averaged over several years. The authors note in the Results that small family groups of western gorillas might be expected to share enterotypes, but that geographical clustering of enterotypes was not observed. So there are apparently shifts in response to fruit availability, but no tendency for nearby individuals to shift enterotypes to resemble one another? One wishes that the sampling of wild gorillas of apparently unknown identity and group membership had been complemented by longitudinal sampling of wild western gorillas habituated to human presence, as microbiome data in combination with the feeding or phenology data typically collected on gorillas would allow evaluation of these posited enterotype shifts. As a minor point, it is not quite right to say that mountain gorillas do not eat fruit, as the Bwindi mountain gorillas eat appreciable amounts of fruit when available.

The authors undertook some microsatellite genotyping analyses to evaluate whether different samples may have come from the same individual. However, little information is provided regarding the degree of repetition of genotyping (necessary because of 'allelic dropout' when using DNA from feces), the number of differences observed between the most similar genotypes deemed from different individuals, or the probability of identity significance thresholds employed.

Reviewer #2 (Remarks to the Author):

Overall I found the article a fascinating study of gorilla microbiome. It provides an overall description of the microbiome, but is particularly original in trying to determine the functional explanations (rainfall and diet) which may account for seasonal changes in the microbiome. It is somewhat difficult to follow at times, but this could be greatly improved by using the introduction to provide a better summary of what was learned in the previous studies of human and chimpanzee gut microbiome. For example, how much variation is there in human or ape microbiota across sites. Is there evidence for temporal variation in other studies. How is diet thought to influence it, etc..). How are enterotypes defined? What is known about their function (linked to what specific diet) How have these been shown to vary in within and between human and ape populations?

Why is this study based on comparison at phylum and family level given that the chimpanzee/ human study cited was based on genus level comparisons?

What effect, if any, would be predicted if apes at one site had considerable contact with human (as in the chimpanzees at Gombe) and none as in the present gorilla study

Rainfall data used in the current study do not match exactly with data from the sites where gorilla diet has been studied. The major difference is that in these other areas there is a single dry season (Dec-Feb) rather than the two dry seasons identified here. The coding of the fruiting season is not completely accurate based on currently available feeding data on gorillas in the area. Major fruit feeding season includes September (Masi et al 2009; Doran-Sheehy et al). Doran-Sheehy et al do not indicate a peak in fruit feeding in Feb as is suggested here. Rather, the months of December - March are always very low in fruit feeding time. The authors should consider how recoding their data to reflect this would influence the results. The rationale for the 30 day time lag in analyses should be explained more clearly. Is it necessary in this case?

Small points

1. Gorillas are classified by IUCN and others as two species, western gorillas (*Gorilla gorilla*) and eastern gorillas (*Gorilla beringei*), not one as indicated here.
2. p.1 2nd sentence: Do not cite unpublished research report. WCS has conducted many censuses of ape in Central Africa but does not make these reports accessible to researchers upon request. Therefore these results should not be cited.

Reviewer #3 (Remarks to the Author):

Williams and colleagues survey the fecal microbial communities in Western Lowland Gorillas (WLG) across a number of years. They show that the gut microbial community structures of the WLG change significantly over the course of seasons, corresponding to rainfall and fruit availability. They propose 4 enterotypes for WLG that differ from previously proposed human enterotypes, and suggest that there may be a functional basis for the community restructuring across seasons. Two important findings are the change in microbiota over seasonal dietary variation, and the demonstration that enterotype does not define an individual. The paper presents an important dataset in understanding how diet and microbiota interact, and as the first study for the third closest living evolutionary relative to our own species, it has implications for human evolution. It is well-written and has a thoughtful discussion, although the interest to a broad readership is in question in the absence of some key data and

analyses described below.

Major points

For this to be a high impact study with significance to a broad readership, the authors need to do a better job placing this data into context of other human data (Tanzania, Malawi, Venezuela, US, Europe, etc) and non-human primates. Fig. S2 and S3 are not sufficient in this respect. It actually is now fairly trivial to resequence distinct regions of 16S (if DNA is already purified, which appears to be the case here) so that proper comparisons can be made. Obtaining V4 (Caporaso et al., 2011, PNAS), which would enable direct comparison to traditionally-living human populations AND V6-V9, for NHP, would enable comparisons required for important insight to make this a high impact study.

The paper is lacking diversity measures, which need to be included in comparison to human and NHP datasets.

The authors are only able to explore potential function restructuring with *Treponema*.

"The remaining two enterotypes were defined by under-characterized bacterial genera (SHD-231 and *Solibacillus*). As such, their functional roles in the WLG microbiota remain enigmatic."

Just a few metagenomic datasets for these currently enigmatic types would provide great insight and significantly strengthen the paper.

The authors miss the opportunity to re-frame the term enterotype. The vague definition of "long-term dietary patterns" currently cited with enterotype in this field gains some concreteness in this study, albeit in another primate, but the authors should consider additional discussion of this point. There are additional questions about the validity of data manipulation required to create these groupings (not specific to this study) and how the number of clusters is determined (specific to this study). Just looking at Fig. 2c, it appears that three of the four clusters overlap significantly with many points localized more closely to another enterotype than the one to which the line is drawn. Designation of the 4 clusters presented in Fig. 2c as "enterotypes" requires more support. A major concern is - as presented - the ambiguous determination of 4 clusters (CH and SI scores are not easily distinguishable from pure chance). Furthermore, the differentiation of the 4 clusters on a PCoA plot appears to depend more on PC2 than PC1.

Assignment of significance to Fig 3a and other figures that present LDA scores requires more support. Authors did not adjust for multiple testing in their ANOVA analysis and reference LEfSe's minimization of false positives as a basis for not performing multiple hypothesis correction. Please elaborate and demonstrate that results from LEfSe's do not statistically suffer from false positives.

Minor points

There should be a better attempt to depict/quantify how other variables (age, gender, location) do or do not contribute to interindividual variability.

- Does the study or sampling process account for gorilla age variability? Are the microbial communities of young gorillas relatively unstable?
- Authors differentiated between the genders of the WLG, yet have not presented data. Findings with respect to non-/differentiating factors may be of general interest for those reading this study.
- Claims of no sample-site bias in HF/LF are made based on the geographical map in Fig 8f. Are authors able to provide additional information - this reviewer is unfamiliar with the region and it

seems plausible that these areas may differ in availability of amounts and types of fruits and so potentially introduce sampling bias. This may be difficult, but any information about different regions (e.g. are different fruits available in different areas?) could help to strengthen this case.

Abstract reads that the authors performed "metagenomic sequencing". More precisely, it should read "16S rRNA sequencing".

Fig 1c appears to be ordered by sample ids that don't appear to be in any order relevant to the body of the paper. Please re-order these to something better than random - perhaps by month, or using clustering?

Fig. 1d should be eliminated or moved to supplemental. The figure is fairly useless due to the inability to clearly differentiate colors and it takes up a lot of space to make the point of variability, which is evident enough from other panels/figs. Perhaps a PCoA plot comparing all of these gorilla samples would be a good replacement. Are there clusters (that correspond to seasons) when a method distinct from that used in 2C to illustrate enterotypes is used?

Fig 2c and Fig 2d should indicate % of variance explained by PC1 and PC2.

Authors should be more precise in their usage of the terms "relative abundance" and "abundance". There are many cases where it appears it would be more appropriate and potentially less ambiguous for them to use "relative abundance".

For instance: pg5 last paragraph, pg 6, pg 7 last paragraph, etc.

Y-axis of 5b needs better labeling.

Figures were not labeled/numbered making review a bit more challenging than necessary.

Reviewer #4 (Remarks to the Author):

General Comments

I enjoyed reading your manuscript on the dietary fluctuations within the intestinal microbiota of western lowland gorillas (WLG) residing in the Republic of the Congo. Overall, the manuscript is well written and organized. The results clearly demonstrate that WLG and humans/chimpanzees have distinct intestinal microbiota, and that the prevalence of certain WLG enterotypes change with diet/season. I have some concerns regarding the persistence of enterotypes 1 across dietary/seasonal shifts, especially given that this is the most prevalent enterotypes within the WLG.

Major Comments

1. My major concern is the prevalence of enterotypes 1 across dietary/seasonal shifts. It is unclear from the provided analyses if this enterotype is robust to dietary/seasonal shifts or if changes in diet/season impact the intestinal microbiota of WLG with this enterotype (though, to an extent insufficient to cause a change to a different enterotype). For example, while the relative abundance of samples assigned to enterotype 1 does not change between dry and wet months (as shown by Fig. 4e), there may still be important changes in the bacterial community within this enterotype. A more detailed examination between enterotype 1 samples across different factors (dry vs. wet, HF vs. LF)

would substantial increase the impact of this paper.

2. While some of the PCoA plots are explicitly specified as being generated under the weighted UniFrac distance, it is not always clear which distance measure was used. In particular, are the PCoA plots in Figure 1c and 1d generated using the rJSD measure? I understand that the enterotypes are defined using rJSD with PAM clustering, but am uncertain if this measure or some other measure was used during the PCoA inference. These PCoA plots, along with a few others, are also missing the amount of variation captured by the displayed components. Can you also please specify which program was used to generate the PCoA plots and verify that these are "unsupervised PCoA" plots, as opposed to "PCoA with an instrumental variable" plots as used in the Arumugam et al. paper. Related to the above point, do samples within enterotype 1 separate out into distinct dry vs. wet (or HF vs. LF) clusters within the PCoA plots.

Minor Comments requiring Attention

The following points are minor in nature, but important for clarifying the paper:

1. Given the wide use of LEfSe in the manuscript, a sentence or two on how to interpret the log-transformed LDA scores would be instructive.
2. I strongly recommend that the community profiles obtained from QIIME for the 87 WLG be included as supplemental data. This would substantially aid in further exploring the results of this study.
3. Based on Extended Data Figure 1d, it does not appear that *Treponema* and *Fibrobacter* are positively correlated except at extremely low abundances (i.e., *Treponema* < 0.1). I believe the large number of samples where *Treponema* < 0.1 and *Fibrobacter* < 0.002 are strongly influencing the correlation.
4. The whiskers in a box plot can represent many different statistics (2nd percentile, 9th percentile, measure of standard deviation from mean, etc.) so the statistic shown needs to be explicitly stated.
5. USEARCH should be cited in addition to QIIME.
6. The Box-Cox-transformation represents a family of transformation that is typically parameterized on lambda. Which value of lambda was used?
7. In at least one instance, Fig. 4e is referred to as Fig. 4d in the manuscript.

Minor Suggestions

Given the high quality of your manuscript, I would like to draw your attention to a few additional minor points:

1. Page numbers and line numbers should be added to aid in reviewing the manuscript.
2. The introduction indicates that human enterotypes are "dominated" by a *Bacteroides*, *Prevotella*, and *Ruminococcus*. Dominated may be a bit strong given that *Ruminococcus* is present < 1% and acts more as a biomarker.

3. Gorillas are sometimes referred to as our closest relatives after chimpanzees, while other times bonobos are explicitly mentioned.

4. Figure 1c and 1d would be far more striking and easier to follow if samples were organized by decreasing abundance of Firmicutes. This approach was taken in Extended Data Figure 1b.

5. Extended Data Figure 10a lists 'wet season 1' twice in the legend.

6. The last sentence in the second paragraph of the "Bioinformatic analysis" section is awkward. It also mentions UPGMA clustering, but this does not appear in any figures.

Authors' Response Summary – We are grateful for all of the reviewers' comments and suggestions. We have addressed all of the reviewers concerns. Indeed, the manuscript is greatly improved as a result. In order to place our findings in Western lowland gorillas in context, we have now conducted additional experimentation to include a dataset of sympatric chimpanzees, as well as included previously published microbiota datasets for Old World monkeys (baboons, red-tailed guenons, black-and-white colobi, and red colobi) and two human microbiota datasets (U.S. humans and a longitudinal dataset from Mongolian humans). In addition to our initial findings that demonstrated seasonal fluctuation of the microbiota and enterotype distribution in Western lowland gorillas, we now provide additional evidence for seasonal fluctuation of the microbiota and enterotypes of sympatric chimpanzees. We have also now conducted shotgun metagenomic sequencing for representative samples from the four Western lowland gorilla enterotypes. Results from metagenomic comparisons have revealed important differences in functional metabolic pathways, archaeal communities and dietary plant taxa found in gorilla fecal samples that were not appreciated in our prior submission. We believe that these additional findings, as well as additional analytical analyses requested by the reviewers and additional interpretation and discussion, elevates our manuscript to warrant publication in Nature Communications. Furthermore, our findings of seasonal fluctuation in our closest great ape relatives is timely given the recent publication by Smits et al, that demonstrated seasonal fluctuation in the microbiome of the human Hadza hunter-gatherers from Tanzania¹.

Reviewer #1 (Remarks to the Authors and Authors' Responses)

The manuscript by Williams and colleagues presents some results from analysis of the gorilla microbiome using nearly 100 samples from the wild. The focus is on enterotypes, an assessment of co-occurring taxa.

Reviewer 1, comment 1: *The authors have two main claims. First, they suggest that the enterotypes found in gorillas are remarkably different from those found in chimpanzees and humans. This does not appear consistent with the results of Moeller and colleagues, appearing a few days ago in PNAS, in which analysis of sequences from hundreds of chimpanzees, gorillas, and humans of varied origins suggested that human microbiomes lack diversity, and hence are quite different from those of our closest living relatives. It may be that the study by Williams et al reaches a different conclusion due to the use of a limited sampling of gorillas or an inappropriate comparative dataset or other methodological issues, such as the focus on enterotypes.*

Authors' Response: We are grateful for the opportunity to address these concerns, as there are several aspects of Moeller and colleagues' publications that are not readily apparent without careful examination. At the time of our initial manuscript submission, the Moeller *et al.* 2014 paper had not been published. It is first worth pointing out that while Moeller *et al.*, 2014 indicated that their analyses were based on 160 chimpanzees, 70 bonobos, and 186 gorillas², this was in fact not the case. A

large number of these samples were excluded from their analyses due to “sequencing depth” issues as we only learned from direct correspondence with Moeller. In fact, visual evaluation of Figure 2B and Figure S3 from Moeller’s paper reveals that only approximately 31-43 (about half the number evaluated in our current study) of their initial 186 gorillas were analyzed, and only approximately 25 of their initial 70 bonobos were included for analyses. As sample numbers were not included in their study for individual analyses and these exclusions were not made apparent in their manuscript, it is quite difficult to reconcile the claimed number of animals studied with those that were actually evaluated. Thus, our paper including 87 WLGs, surpasses the number that was actually evaluated by Moeller.

Moeller’s earlier work evaluating the microbiome of chimpanzees and humans suggested that humans and chimpanzees have compositionally similar enterotypes³. Thus, if it is the conclusion of Reviewer 1 that based on the recent Moeller *et al.* study in PNAS, chimpanzees and gorilla microbiomes are quite different from human microbiomes, then it is Moeller’s publications that are in conflict. In fact, we have now included in our study the microbiomes and enterotypes of sympatric chimpanzees. We do not find that these chimpanzees have similar enterotypes to human groups. The conflict between our findings and those of Moeller *et al.*³ may be due to the fact that our samples were collected from a different geographic location than those in Moeller *et al.*; that our samples were collected across different seasons, while those from Moeller *et al.* were only collected during October, November, and December; and/or because of other methodological variations. Indeed, it is difficult to ascertain the precise methodology employed by Moeller *et al.* to identify human and chimpanzee enterotypes, as they reported using the same methods as Arumugam *et al.* but failed to re-capitulate the human enterotypes identified by Arumugam *et al.* (a *Bacteroides* enterotype, a *Prevotella* enterotype, and a *Ruminococcus/Blautia/Lachnospiraceae* enterotype) using the same dataset.

Further, our inclusion of sympatric chimpanzees for comparison to gorilla populations is far more telling of differences in the microbiomes of these great apes that are independent of environment. When considering the microbiome from an evolutionary perspective, it is important to limit the influence of potentially confounding factors, such as environment. Moeller’s recent study compares gorillas and chimpanzees from entirely different countries that could introduce extreme environment-dependent biases. In fact, a recent paper by Gomez *et al.*, showed that geographical range may be an important modulator of the gut microbiomes of Western lowland gorillas even within the same country⁴. Further, in an earlier publication, Moeller *et al.*, showed that sympatric chimpanzees and gorillas harbored more similar gut microbiomes than allopatric populations⁵. Thus, we believe our new analysis of sympatric chimpanzees and gorillas represents a more appropriate comparative dataset.

In our original submission, we did not limit our focus to enterotypes, but also examined variation in taxon abundance between WLGs and humans, as well as within WLGs from different months and seasons, using LEfSe analyses. As this may not have been clear in our original submission, we have now clarified this at the end of the introduction: (**Page 5, lines 102-109**). Further, we have now also presented

analyses of alpha diversity (**Fig. 1d and Supplementary Fig. 3**) and Bray-Curtis dissimilarity among the different human and NHP datasets (**Fig. 1e and Supplementary Fig. 4a,c**), as well as Bray-Curtis dissimilarity among and between WLG and chimpanzee samples collected during different seasons (**Fig. 5g and Supplementary Fig. 11a-d**).

Reviewer 1, comment 2: *However, the results are not presented in a phylogenetic framework that would aid in inferring which lineages have undergone particular changes since the common ancestors. Indeed, the authors should reassess their description of gorillas as 'vestiges of early hominids' (in the Introduction, page and line numbers are unfortunately absent) as present day gorillas are no more vestiges of early hominids than we are; all great ape lineages, including ourselves, have been evolving continuously since they shared common ancestors. It is also rather premature to state that 'the relationships between humans and their microbiota are no more remarkable or complex than those found in the rest of the animal kingdom' as it is a supposition without supporting data and a rather odd way to justify a comparative evolutionary analysis.*

Authors' Response: We appreciate these recommendations, and we believe the addition of several new datasets has greatly strengthened our study. In addition to our new analyses of sympatric chimpanzees, we have included datasets for Old World monkeys (baboons, red-tailed guenons, black-and-white colobi, and red colobi). Inclusion of sympatric chimpanzees and Old World monkeys expands the phylogenetic framework substantially. We also provide direct comparison of these non-human primate datasets with human populations in the U.S. and Mongolia. Thus, we have now addressed Reviewer 1's additional concerns about presenting our data in a phylogenetic framework to infer which lineages have undergone particular changes since the common ancestors.

We have removed the line describing gorillas as vestiges of early hominids, as well as the statement that the relationships between humans and their microbiota are no more remarkable or complex than those found in the rest of the animal kingdom, from the introduction. We have also added page and line numbers.

Reviewer 1, comment 3: *The second major claim of the paper is that western gorilla enterotypes vary with season; by correlating enterotype shifts with changes in rainfall and fruit availability the authors suggest a dynamic response of individuals to ecology. Unfortunately, this intriguing possibility, although plausible, is suggested by rather tenuous approximate associations- the 87 analyzed gorilla samples were collected over a period of three years, yielding very few samples per month, and rainy season data is averaged over several years. The authors note in the Results that small family groups of western gorillas might be expected to share enterotypes, but that geographical clustering of enterotypes was not observed. So there are apparently shifts in response to fruit availability, but no tendency for nearby individuals to shift enterotypes to resemble one another? One wishes that the sampling of wild gorillas of apparently unknown identity and group membership had been complemented by longitudinal sampling of wild western gorillas habituated to human presence, as*

microbiome data in combination with the feeding or phenology data typically collected on gorillas would allow evaluation of these posited enterotype shifts. As a minor point, it is not quite right to say that mountain gorillas do not eat fruit, as the Bwindi mountain gorillas eat appreciable amounts of fruit when available.

Authors' Response: We do not agree with Reviewer 1's assessment that our findings of seasonal enterotype shifts are suggested by "tenuous approximate associations." It is true there were a limited number of samples from some months over the three-year study period. Despite this limitation, when stratified by month, we found statistically significant differences in bacterial taxa. We did not use monthly stratification to assess enterotype distribution. Rather, we assessed enterotype distribution based on broader seasonal patterns (with larger numbers of samples per season, e.g., **Figure 4e** and **Supplementary Figure 10c,d** and **Figure 5d**). While one of our seasonal designations (wet season 1, dry season 1, wet season 2, dry season 2; **Supplementary Figure 10d**) is based on long-term rainfall patterns determined by averaging rainfall data over several years, our primary seasonal analyses are based on average daily rainfall over the 30 days prior (previously presented as total rainfall in the 30 days prior) to each individual sample collection date to account for any rainfall anomalies (**Figure 4d,e**). A similar analysis based on rainfall in the 30 days prior to sample collection and its relationship with the microbiota of baboons has recently been reported⁶. Further, in our monthly (**Figure 3a**) and seasonal (**Supplementary Figure 10a,b,e**) LEfSe analyses, the metagenomic biomarkers presented were only those that were still significantly associated with month or season of sample collection after adjusting for the effects of potentially confounding variables, including year of sample collection, in ANCOVA analyses. Overall, this dataset and these analyses provide a substantially more comprehensive perspective on seasonal shifts in the microbiota of WLGs than could have been achieved in any previous publication, where seasonality was either not considered at all or was analyzed based on samples collected from only a few different months.

We did not claim that WLG family members are expected to share enterotypes. Rather we suggest that based on what is known in humans, cohabitation can lead to shared microbiota (a section that has been removed in this resubmission). However, even in humans it has not been demonstrated that enterotypes are linked to cohabitation. Rather, enterotypes in humans are associated with diet. Thus, the widespread geographical distribution of WLG enterotypes, lack of association of enterotypes with geographical distribution based on our new analyses (**Supplementary Figure 8**) and their association with seasonality and frugivory strongly suggest that gorilla proximity is less of a determining factor for enterotypes than seasonality and frugivory. While longitudinal sampling of habituated gorillas might have complemented our findings, there is ample evidence to suggest that habituation of wild animals to humans can cause stress to the animals, influencing the behavior of the animals (including dietary patterns), and can result in transfer of microbiota between humans and the habituated animals^{7,8}. Furthermore, the large number of gorillas that we were able to investigate using non-invasive sampling methods, serving to represent a broader

population-level study rather than an investigation of small family groups, could not have been achieved through habituation. To address concerns over diet, we have now applied shotgun metagenomic analyses of plants found in WLG feces to look for differences in plant taxa consumed by representative WLGs from the different enterotypes (**Fig. 6c,d,e** and **Supplementary Table 8**). We have removed any suggestion that mountain gorillas do not eat fruit.

Reviewer 1, comment 4: *The authors undertook some microsatellite genotyping analyses to evaluate whether different samples may have come from the same individual. However, little information is provided regarding the degree of repetition of genotyping (necessary because of 'allelic dropout' when using DNA from feces), the number of differences observed between the most similar genotypes deemed from different individuals, or the probability of identity significance thresholds employed.*

Authors' Response: We employed microsatellite genotyping analysis using 8 different microsatellite markers. Each marker was evaluated in duplicate, and those markers with apparent homozygosity were repeated 7 times to exclude allelic dropout as recommended in the literature⁹⁻¹¹. In response to the reviewer's request, we have now also assessed the probability of identity for the gorilla and chimpanzee samples included in our analyses based on these 8 loci ($P=1.955 \times 10^{-18}$). This has now been clarified in the Materials and Methods section (**Page 49, lines 1117-1120**), and a table of individual gorilla and chimpanzee microsatellite results has been included (**Supplementary Table 1**).

Reviewer #2 (Remarks to the Authors and Authors' Responses)

Reviewer 2, comment 1: *Overall I found the article a fascinating study of gorilla microbiome. It provides an overall description of the microbiome, but is particularly original in trying to determine the functional explanations (rainfall and diet), which may account for seasonal changes in the microbiome. It is somewhat difficult to follow at times, but this could be greatly improved by using the introduction to provide a better summary of what was learned in the previous studies of human and chimpanzee gut microbiome. For example, how much variation is there in human or ape microbiota across sites. Is there evidence for temporal variation in other studies. How is diet thought to influence it, etc..). How are enterotypes defined? What is known about their function (linked to what specific diet) How have these been shown to vary in within and between human and ape populations?*

Authors' Response: We have added information about the effects of environment and resource availability on the ape microbiota in the third paragraph of the introduction: (**Page 4-5, lines 86-89 and lines 92-96**). We have also added datasets from humans from the U.S. and Mongolia to our analyses to demonstrate that there is substantial variation in the microbiota of human populations from different geographic locations as well as within human individuals (Mongolians)

(Fig 1d,e; Supplementary Fig. 1f,g; Supplementary Fig. 3a-h; Supplementary Fig. 4b; Supplementary Fig. 16)

We have added information and references about temporal variation in human and animal enterotypes in the second paragraph of the introduction: **(Page 3, lines 60-64)**.

We have added information and references about human enterotypes, their consistency across different human populations, and their link to dietary patterns to the first paragraph of the introduction: **(Page 3, lines 49-59)**.

We have added information about variability in enterotypes between human populations to the introduction: **(Page 3, lines 56-59)**. Within human populations, our current, rudimentary understanding of enterotype variability is that variability is associated with long-term dietary patterns as indicated in the introduction: **(Page 3, lines 49-53)**. We have also now demonstrated with our new analyses that enterotype switch frequently in longitudinal samples collected from Mongolian humans **(Supplementary Fig. 16)**. There is little information on enterotype variability within and between ape populations. Our new submission provides the first such comparison, as we have now assessed enterotype groups in nonhuman primates and humans, including the Western lowland gorilla population and their association with seasonality, as well as, new analyses from sympatric chimpanzees and their association with seasonality, Old World monkeys (baboons), and two human populations (from the U.S. and Mongolia) (see **Fig. 2; Fig. 4d-g; Fig. 5d,f; Supplementary Fig. 4b; Supplementary Fig. 5a-r; Supplementary Fig. 10c,d; Supplementary Fig. 16)**.

Finally, we have carried out functional analyses on representative samples from the 4 Western lowland gorilla enterotypes using shotgun metagenomics to understand the functional metabolic pathways that distinguish these enterotypes and have included analyses of archaeal populations in relation to bacterial enterotypes, as well as dietary plant taxa associated with enterotypes **(Fig. 6, Supplementary Tables 7 and 8, and Supplementary Figs. 12-15)**.

***Reviewer 2, comment 2:** Why is this study based on comparison at phylum and family level given that the chimpanzee/ human study cited was based on genus level comparisons?*

Authors' Response: Our analyses of enterotypes are based on comparison at the genus level as in the original enterotype study¹², as well as the chimpanzee-human study³. Some confusion as to the phylogenetic levels at which different analyses were performed may have arisen from Figure 1C and Figure 1D in our original submission. We have now changed these figures in the new manuscript to reflect the distribution of bacteria in the microbiota of Western lowland gorillas at both the phylum and genus levels. All of our beta-diversity measures including Bray-Curtis dissimilarity and the enterotyping method are carried out at the genus level. It is frequently necessary in microbiome studies to explore different levels of taxonomy to appreciate differences between sampled groups. Thus, to appreciate differences in the relative abundance of bacteria between groups that may arise at different taxonomic levels, all of our LEfSe analyses are performed at all taxonomic levels,

including genus, and alpha diversity measures that have been added for this resubmission are presented at the phylum, genus and OTU (97% OTUs approximating species level) levels.

Reviewer 2, comment 3: *What effect, if any, would be predicted if apes at one site had considerable contact with human (as in the chimpanzees at Gombe) and none as in the present gorilla study*

Authors' Response: We are grateful for the opportunity to address this question. There is ample evidence to suggest that habituation of wild animals to humans can cause stress to the animals, influencing the behavior of the animals (including dietary patterns), and can result in transfer of microbiota between humans and the habituated animals^{4,7,8}, thus compromising the integrity of assessment of untouched wild ape microbiotas and their relationship to dietary patterns. We have alluded to this impact and provided references in our manuscript (**Pages 3-4, Lines 69-71**), but without a direct comparison between chimpanzees at Gombe and chimpanzees from our no-contact study, we do not feel it is appropriate to overstate the impact of human contact. The degree to which human contact may influence primate microbiota may be dependent on the degree of human contact. With extreme contact, as occurs when primates are kept in semi-captivity (kept in sanctuaries) or complete captivity (zoos), the microbiota of non-human primates becomes humanized¹³. While, some previous studies have examined the impact of human contact on the microbiota of gorillas, the impetus of our study was to examine gorilla populations that are more likely to have intact, uncontaminated microbial populations.

Reviewer 2, comment 4: *Rainfall data used in the current study do not match exactly with data from the sites where gorilla diet has been studied. The major difference is that in these other areas there is a single dry season (Dec-Feb) rather than the two dry seasons identified here. The coding of the fruiting season is not completely accurate based on currently available feeding data on gorillas in the area. Major fruit feeding season includes September (Masi et al 2009; Doran-Sheehy et al). Doran-Sheehy et al do not indicate a peak in fruit feeding in Feb as is suggested here. Rather, the months of December - March are always very low in fruit feeding time. The authors should consider how recoding their data to reflect this would influence the results.*

Authors' Response: Information from study sites near ours indicate that July and August are consistently high fruiting periods¹⁴⁻²⁰. While some studies further indicate that June as well as September are also high fruiting months^{14,16,17,19}, there is less consistency in the literature. This may relate to yearly variation in rainfall that may impact the start of the fruiting season (June) and the end of the fruiting season (September). While there is variation in rainfall patterns among sites, Poulsen *et al.* reported a similar bimodal rainfall pattern to ours (highest rainfall in May and October, lowest rainfall in July) that corresponded to the highest fruit availability in July, followed by August²¹. However, because of the variation among different studies regarding whether or not June and/or September are high fruiting

months, we have adopted a conservative approach by only including the less variable fruiting months of July and August in our analyses. For our analyses, we have excluded samples collected in June and September, as fruiting in these transition months may be less consistent (we did not code these as low frugivory months). Thus we have only included in our statistical analyses, the consistent high frugivory months of July and August in comparison with the consistent low frugivory months of October to May. Furthermore, inclusion of September as a high frugivory month (rather than excluding this month from analyses as we have done) does not impact our findings that enterotype distribution is significantly associated with frugivory. The figure below shows this analysis with the inclusion of September as a high frugivory month and can be compared with **Figure 5d** in our manuscript in which September was excluded from analyses.

We agree with this reviewer’s suggestion that no peak in fruit feeding in February has been clearly documented. However, in the referenced article by Doran-Sheehy et al.¹⁶, there was a modest peak in frugivory (especially evident in male WLGs, see Figure 2a in the referenced article) in the month of February, despite an apparent lack of increased fruit availability. We raise this point, not to suggest that there is a major peak in fruit feeding in February, but rather to highlight that we observe a minor peak in frugivory month-associated bacteria in the month of February. We have clarified this statement in the frugivory section of the results to read as follows: “While no prominent peak in fruit feeding in February has been clearly documented, one study indicated a modest increase in frugivory in February compared to December in WLGs¹⁶, but detailed diet studies in habituated WLGs in this region will be needed to evaluate whether any changes in dietary patterns occur at the end of dry season 1 to account for this.”

Reviewer 2, comment 5: *The rationale for the 30 day time lag in analyses should be explained more clearly. Is it necessary in this case?*

Authors’ Response: Two papers suggest that the lag time between changes in rainfall and changes in vegetation could be anywhere from 1 week to 1 month^{22,23}. Further, while it is not well-defined how quickly feeding habits would change in response to vegetation or how quickly the microbiota would respond to changes in

diet (though the latter has been shown to occur very quickly in humans), the average rainfall in the 30 days prior to sample collection is intended to account for the inter-annual variations in rainfall, as well as the actual date on which the sample was collected (early in the month as opposed to later in the month, which is not accounted for in the decadal rainfall pattern analyses), while also accounting for the lag period between changes in rainfall and changes in diet. A recent study in baboons also examined changes in the microbiota associated with seasonal rainfall using total rainfall in the 30 days prior to sample collection for analyses²⁴. We have further investigated whether there were significant differences in total rainfall ranging from 7 to 30 days prior to sample collection between samples assigned to gorilla enterotype 2 (the rainfall-associated enterotype, see **Figure 4d** in the manuscript) and samples assigned to other gorilla enterotypes, as well as between samples assigned to the two newly reported chimpanzee enterotypes (we did not include this analysis in the manuscript). We found that there was a significant difference in total rainfall between samples assigned to gorilla enterotype 2 and gorilla enterotype 3 (the two seasonally-responsive enterotypes) as early as 7 days prior to sample collection, but significance was greatest between 3 weeks and 1 month. Similar results were obtained for seasonal fluctuation in chimpanzee enterotypes. For these reasons, we believe that analysis of the average rainfall in the 30 days prior to sample collection is well justified.

Small points

Reviewer 2, comment 6: *Gorillas are classified by IUCN and others as two species, western gorillas (*Gorilla gorilla*) and eastern gorillas (*Gorilla beringei*), not one as indicated here.*

Authors' Response: We are grateful to this reviewer for pointing this out. We recognize that gorillas are classified as two species, and our sentence about all gorilla subspecies being classified by IUCN as endangered was meant to imply that all subspecies of the two gorilla species are classified as endangered. We have removed this portion of the introduction.

Reviewer 2, comment 7: *p.1 2nd sentence: Do not cite unpublished research report. WCS has conducted many censuses of ape in Central Africa but does not make these reports accessible to researchers upon request. Therefore these results should not be cited.*

Authors' Response: We are grateful for this recommendation. We have removed this portion of the introduction.

Reviewer #3 (Remarks to the Authors and Authors' Responses)

Williams and colleagues survey the fecal microbial communities in Western Lowland Gorillas (WLG) across a number of years. They show that the gut microbial community

structures of the WLG change significantly over the course of seasons, corresponding to rainfall and fruit availability. They propose 4 enterotypes for WLG that differ from previously proposed human enterotypes, and suggest that there may be a functional basis for the community restructuring across seasons. Two important findings are the change in microbiota over seasonal dietary variation, and the demonstration that enterotype does not define an individual. The paper presents an important dataset in understanding how diet and microbiota interact, and as the first study for the third closest living evolutionary relative to our own species, it has implications for human evolution. It is well-written and has a thoughtful discussion, although the interest to a broad readership is in question in the absence of some key data and analyses described below.

Major points

Reviewer 3, comment 1: *For this to be a high impact study with significance to a broad readership, the authors need to do a better job placing this data into context of other human data (Tanzania, Malawi, Venezuela, US, Europe, etc) and non-human primates. Fig. S2 and S3 are not sufficient in this respect. It actually is now fairly trivial to resequence distinct regions of 16S (if DNA is already purified, which appears to be the case here) so that proper comparisons can be made. Obtaining V4 (Caporaso et al., 2011, PNAS), which would enable direct comparison to traditionally-living human populations AND V6-V9, for NHP, would enable comparisons required for important insight to make this a high impact study.*

Authors' Response: We agree that Fig. S2 and S3 in the original submission were not sufficient in placing our data in the context of human and other NHP data. We have now added data from sympatric chimpanzee samples (collected for this study and sequenced using V1-V3 primers). In addition, we compare our findings from sympatric great apes with V1-V3 sequences from humans from urban and rural Mongolia, as well as U.S. humans, and from four different Old World monkey species, including a recently sequenced large dataset of baboons. The addition of Old World monkey samples, along with the sympatric chimpanzee samples, allowed us to expand upon the phylogenetic framework beyond that reported by Moeller *et al.* and allowed us to investigate seasonal impacts on the microbiota in two sympatric great ape populations. Further, the addition of 320 longitudinal samples from 64 individual humans in urban and rural Mongolia (each sampled at 5 different time points over 11 months) has afforded us additional human comparator groups from distinct environments and has also allowed us to examine enterotype stability over relatively short time spans in a human population. We further understand the need to have V4 sequences available for other researchers and for future comparisons. Indeed, all of our current sequencing has moved to V4 sequencing with MiSeq. We are in the process of resequencing all sympatric Western lowland gorillas and chimpanzees in this study for a subsequent publication that will be available to other researchers.

Reviewer 3, comment 2: The paper is lacking diversity measures, which need to be included in comparison to human and NHP datasets.

Authors' Response: Thank you for this useful suggestion. We have added alpha diversity measures at the phylum, genus, and 97% OTU levels (**Figure 1d** and **Supplementary Fig. 3a-h**) for all NHP and human populations now included in our study. We have also added principal coordinate analyses based on Bray-Curtis dissimilarity (**Fig. 1e**), and square root Jensen-Shannon divergence (**Supplementary Fig. 4b**) among samples from the different datasets. These datasets now include two human datasets from Mongolia and the U.S., as well as from new sympatric chimpanzees and four Old World monkey species.

Reviewer 3, comment 3: The authors are only able to explore potential function restructuring with Treponema. "The remaining two enterotypes were defined by under-characterized bacterial genera (SHD-231 and Solibacillus). As such, their functional roles in the WLG microbiota remain enigmatic." Just a few metagenomic datasets for these currently enigmatic types would provide great insight and significantly strengthen the paper.

Authors' Response: We have now explored phylogenetic relationships of these taxa (**Supplementary Fig. 6**) and performed shotgun metagenomic analyses of functional metabolic pathways and microbial genes associated with each of the four Western lowland gorilla enterotype groups. The most abundant *Treponema*, *SHD-231*, and *Solibacillus/Staphylococcus* OTUs all grouped most closely to ruminant-derived sequences (**Supplementary Fig. 6**). We are grateful for the recommendation to include metagenomic analyses, as we believe these analyses and their implications have greatly improved our manuscript. Not only have these analyses provided important insights into the functional metabolic pathways associated with Western lowland gorilla enterotypes, but have also afforded us the ability to assess differences in archaeal communities in relation to bacterial enterotypes and differences in dietary plant taxa found in fecal samples from these enterotypes. The results of shotgun metagenomics are now reported on **Pages 25-33, Lines 552-743, Fig. 6a-e, Supplementary Figs. 12-15, and Supplementary Tables 5-8**. Interpretation and discussion of these results are added to the discussion of the main manuscript and **Supplementary Notes 5, 6**. Methods are provided on **Pages 51-52, Lines 1167-1179 and Pages 55-56, Lines 1244-1270**.

Reviewer 3, comment 4: The authors miss the opportunity to re-frame the term enterotype. The vague definition of "long-term dietary patterns" currently cited with enterotype in this field gains some concreteness in this study, albeit in another primate, but the authors should consider additional discussion of this point. There are additional questions about the validity of data manipulation required to create these groupings (not specific to this study) and how the number of clusters is determined (specific to this study). Just looking at Fig. 2c, it appears that three of the four clusters overlap significantly with many points localized more closely to another enterotype than the one to which the line is drawn. Designation of the 4 clusters presented in Fig.

2c as "enterotypes" requires more support. A major concern is - as presented - the ambiguous determination of 4 clusters (CH and SI scores are not easily distinguishable from pure chance). Furthermore, the differentiation of the 4 clusters on a PCoA plot appears to depend more on PC2 than PC1.

Authors' Response: Thank you for this suggestion. We recognize based on this reviewer's comments that we missed an opportunity to redefine enterotypes. In this resubmission, we have added additional datasets, performed additional analyses, and re-structured the introduction and discussion to provide evidence in both humans and NHPs that enterotypes are not discrete, static states that define an individual and rather are generally gradients of dominant bacteria that can fluctuate rapidly in response to environmental changes. In fact, we have made this a more central theme of the paper, as indicated by the new title of the manuscript. We now show that enterotype distribution as well as the relative abundance of enterotype drivers within populations of both wild Western lowland gorillas and sympatric chimpanzees is influenced by seasonality. We further demonstrate that enterotype switching occurs frequently and over short time frames (months) in Mongolian humans (**Pages 33-35, Lines 745-785; Supplementary Fig. 16**), suggesting that long-term dietary patterns are not the sole determinant of enterotypes. For additional discussion see **Pages 35-36, Lines 802-823**.

Regarding the support of the enterotype clustering performed in this study, while the 2D PCoA plot showing the WLG enterotypes (now **Fig. 2b**) does show overlap among clusters 1, 3, and 4 based on PC1 and PC2, the two seasonally fluctuant enterotypes 2 and 3 separate along PC1 based on both rJSD (**Fig. 2b**) and BC (**Supplementary Fig. 7a**). In addition, PC2 explains 19% of the variance based on rJSD and 11% of the variance based on BC compared with 21% and 15%, respectively, along PC1. Thus, PC2 explains nearly as much variance as PC1. The separation of enterotype 1 samples along PC2 and its position between the seasonally fluctuant enterotypes suggests that enterotype 1 likely represents a gradient/transitional state between enterotypes 2 and 3 which is made evident in our new submission (see **Supplementary Fig. 11**). We demonstrate that the ratio of *Treponema/Prevotella* relative abundance forms a gradient along PC1, while *Chloroflexi* forms a gradient along PC2. In addition, we now show that samples from enterotype 1 (the *Chloroflexi* enterotype) that were collected during high frugivory months have higher relative abundance of bacterial taxa associated with frugivory and samples collected during low frugivory months have higher relative abundance of low frugivory associated taxa (**Supplementary Fig. 11**), supporting our conclusion that enterotype 1 represents a transitional state between the seasonally fluctuant enterotypes. Finally, enterotype 4 separates widely along PC3 from each of the other 3 enterotypes (**Supplementary Fig. 7a**). We recognize the controversy regarding the validity of the enterotyping method (not specific to this study), and we have tried to place this in context in the introduction and in our discussion (**Pages 3-4, Lines 65-74; Pages 35-36, Lines 802-823**). Finally, we have clarified that our intention is not to claim that Arumugam's method defines statistically supported discrete clusters: **Page 14, Lines 301-308**.

Reviewer 3, comment 5: *Assignment of significance to Fig 3a and other figures that present LDA scores requires more support. Authors did not adjust for multiple testing in their ANOVA analysis and reference LEfSe's minimization of false positives as a basis for not performing multiple hypothesis correction. Please elaborate and demonstrate that results from LEfSe's do not statistically suffer from false positives.*

Authors' Response: The LEfSe method, which uses a Kruskal-Wallis test followed by a bootstrapped linear discriminant analysis, was published in 2011 and has been employed for metagenomic biomarker discovery in hundreds of published studies with no additional correction for multiple comparisons. In the paper describing the LEfSe method²⁵, it was demonstrated through use of synthetic data that LEfSe is conservative in identifying differential features among classes, with a substantially lower false positive rate than standard Kruskal-Wallis tests, as well as the Metastats method. However, one limitation of LEfSe that is not considered in any published studies that we have encountered is its inability to account for potentially confounding variables. Therefore, all taxa identified as monthly or seasonal metagenomic biomarkers by LEfSe in this study were further analyzed as dependent variables in ANOVA or ANCOVA analyses, with both the factor of interest (month, season, or total monthly rainfall), as well as other potentially confounding factors (sex, year of collection, latitude of collection site, and longitude of collection site), as independent variables. As these follow-up ANOVA/ANCOVA analyses (the parametric equivalent of the Kruskal-Wallis test used in LEfSe) were planned comparisons made solely to ensure that taxa already identified as metagenomic biomarkers by LEfSe were still significantly associated with the variable of interest after adjusting for the effects of potentially confounding variables, corrections for multiple comparisons were not applied. These analyses increase neither the familywise type I error rate nor the FDR. In fact, by conducting the adjusted analyses only on the biomarkers identified by LEfSe, we are only discarding biomarkers that no longer have a significant effect on the dependent variable after covariate adjustment. These ANOVA/ANCOVA analyses were not used to resurrect a biomarker previously discarded by LEfSe. Therefore, for any given LEfSe-selected biomarker, the probability of a false rejection of the null hypothesis and the FDR can only decrease from what it was with LEfSe alone. To illustrate that, if we remove a biomarker because it's not significant after adjustment, then the original FDR, $(\#false)/(n \text{ discoveries})$, is greater than the final FDR, $(\#false-1)/(n-1)$. By definition, the original biomarker can't be a true discovery if it's an artifact of confounding. Thus, LEfSe plays the role of a gatekeeping procedure (hierarchical or fixed-sequence test procedure)^{26,27} and our ANOVA/ANCOVA analyses strengthen our results from LEfSe by further reducing the FDR. We have clarified this in the methods section of our manuscript (**Pages 59-60, Lines: 1346-1357**).

Minor points

There should be a better attempt to depict/quantify how other variables (age, gender, location) do or do not contribute to interindividual variability.

Authors' Response: (see responses below)

Reviewer 3, comment 6: *Does the study or sampling process account for gorilla age variability? Are the microbial communities of young gorillas relatively unstable?*

Authors' Response: Unfortunately, as these samples were collected from wild, non-habituated gorillas, we cannot account for age variability. We have now indicated this in the Materials and Methods section of the manuscript (**Page 50, Lines 1133-1135**). Most of the feces collected were from substantial deposits found in nests and likely represent adult/adolescent samples. It is unlikely that any of our samples represent infant samples as infants typically defecate on their mothers, a behavior that, perhaps luckily for our own mothers, has been circumvented by the invention of diapers. It is further unlikely that the rare *Solibacillus/Staphylococcus*-driven enterotype samples could represent infants, as infant WLGs nurse until they are 2-3 years old and we detected common plants consumed by post-weaning adolescent and adult WLGs in these samples. However, we cannot exclude the possibility that some samples could have been collected from preadolescent, post-weaned WLGs.

Reviewer 3, comment 7: *Authors differentiated between the genders of the WLG, yet have not presented data. Findings with respect to non-/differentiating factors may be of general interest for those reading this study.*

Authors' Response: We agree that this is an important analysis. We undertook gender analyses but found that no taxa significantly distinguished male vs. female samples. We have now indicated this in the manuscript (**Page 14, Lines 308-309**).

Reviewer 3, comment 8: *Claims of no sample-site bias in HF/LF are made based on the geographical map in Fig 8f. Are authors able to provide additional information - this reviewer is unfamiliar with the region and it seems plausible that these areas may differ in availability of amounts and types of fruits and so potentially introduce sampling bias. This may be difficult, but any information about different regions (e.g. are different fruits available in different areas?) could help to strengthen this case.*

Authors' Response: We agree that this would be very useful, and we have consulted with ecologists that have studied frugivory in neighboring areas. Unfortunately, there are no available data on the fruit availability for our specific study site (in terms of fine scale geographical seasonal fluctuations in overall fruit availability or in terms of individual fruit availability). However, to quantitatively adjust for the effects of location of sample collection, we have added latitude and longitude of sample collection sites to our confirmatory ANCOVA analyses for this resubmission. In addition to the widespread distribution of enterotypes indicated in our geographical map (now **Supplementary Fig. 8a**), we have also now shown that BC dissimilarity within enterotype groups and within high and low frugivory seasons is not associated with distance between sample collection sites, suggesting that the microbiota of samples collected from nearby sites were no more similar to each other than samples that were separated over wide geographical distances

(**Supplementary Fig. 8b,c**). Similarly, the difference between samples in the relative abundance of the two seasonally fluctuant taxa, *Treponema* and *Prevotella*, were not associated with geographical distance within high and low frugivory (**Supplementary Fig. 8d,e**). Finally, to further address this question, we have investigated dietary plant taxa found in fecal samples, based on shotgun metagenomic data, from each of the 4 WLG enterotypes. Our new findings indicate that *Prevotella* dominant (enterotype 3; associated with high frugivory) samples have significantly lower relative abundance of high fiber fallback/staple foods (*Zingiberaceae*, *Moraceae*, and *Marantaceae*) and higher cumulative relative abundance of *Annonaceae* (a fruiting plant) and *Urticaceae* (a liana) plant DNA (**Fig. 6c-e** and **Supplementary Table 8**). In contrast, the rare *Solibacillus/Staphylococcus* dominant (enterotype 4) samples had relatively high levels of *Moraceae* compared to the other enterotypes (**Fig. 6c**). The implications of these findings and their association with differences in functional metabolic pathways identified through shotgun sequencing are discussed (**Pages 31-33, Lines 694-743; Page 38, Lines 861-863; Pages 42-44, Lines 956-989; and Supplementary Note 6**).

Reviewer 3, comment 9: *Abstract reads that the authors performed "metagenomic sequencing". More precisely, it should read "16S rRNA sequencing".*

Authors' Response: We have made this clarification and have also now included shotgun metagenomic analyses on a subset of WLG fecal samples.

Reviewer 3, comment 10: *Fig 1c appears to be ordered by sample ids that don't appear to be in any order relevant to the body of the paper. Please re-order these to something better than random - perhaps by month, or using clustering?*

Authors' Response: We have eliminated **Figure 1c** and replaced it with a box-plot displaying all of the phyla and the most abundant genera in the WLG microbiota.

Reviewer 3, comment 11: *Fig. 1d should be eliminated or moved to supplemental. The figure is fairly useless due to the inability to clearly differentiate colors and it takes up a lot of space to make the point of variability, which is evident enough from other panels/figs. Perhaps a PCoA plot comparing all of these gorilla samples would be a good replacement. Are there clusters (that correspond to seasons) when a method distinct from that used in 2C to illustrate enterotypes is used?*

Authors' Response: We have eliminated **Figure 1d** and replaced it with an alpha rarefaction plot and a PCoA plot based on Bray-Curtis dissimilarity for the human and NHP samples included in this study. **Figure 2c** in the previous submission now corresponds to **Figure 2b** (upper left panel) in the current submission. The clustering observed in **Figure 2b** was largely similar when PCoA based on BC dissimilarity was evaluated with enterotype 2 and 3 samples separating along PC1, enterotype 1 samples separating along PC2 and falling between enterotypes 2 and 3 along PC1, and PC4 separating along PC3 (**Supplementary Fig. 7a**). Similar results

were obtained based on UniFrac (data not shown). Based on PCoA, our results suggest the distribution (prevalence or density) of high frugivory samples changes along PC1 which is defined by the ratio in relative abundance of the two seasonally fluctuant enterotype drivers, *Treponema/Prevotella* ratio. We have now included data that demonstrates this phenomenon in **Supplementary Figure 11**. It is evident, in the newly added **Supplementary Fig. 11c**, that the vast majority of high frugivory samples fall within the *Prevotella*-driven enterotype 3 or are *SHD-231*-driven enterotype 1 samples that cluster more closely to enterotype 3 along PC1. Even within the transitional enterotype 1 samples, those collected during the high frugivory period have a lower *Treponema/Prevotella* ratio, and the overall relative abundance of high frugivory and low frugivory biomarkers differ significantly between the seasons. Thus, enterotype 1 is not robust to dietary/seasonal shifts, which further supports our assertion that enterotype 1 represents a transitional enterotype where rapid changes in dietary patterns have occurred, but the microbiome has not yet responded to a degree that results in enterotype switching.

Reviewer 3, comment 12: *Fig 2c and Fig 2d should indicate % of variance explained by PC1 and PC2.*

Authors' Response: We have made this change.

Reviewer 3, comment 13: *Authors should be more precise in their usage of the terms "relative abundance" and "abundance". There are many cases where it appears it would be more appropriate and potentially less ambiguous for them to use "relative abundance". For instance: pg5 last paragraph, pg 6, pg 7 last paragraph, etc.*

Authors' Response: We have clarified throughout that manuscript and in the figure legends that we are referring to relative abundance.

Reviewer 3, comment 14: *Y-axis of 5b needs better labeling.*

Authors' Response: We have made this change.

Reviewer 3, comment 15: *Figures were not labeled/numbered making review a bit more challenging than necessary.*

Authors' Response: Typically, for manuscript submission, the main figures are numbered/labeled by the file name as they are uploaded, rather than having the figure number incorporated into the figure. All supplementary figures are labeled and numbered directly in the legends beneath them.

Reviewer #4 (Remarks to the Authors and Authors' Responses)

General Comments I enjoyed reading your manuscript on the dietary fluctuations within the intestinal microbiota of western lowland gorillas (WLG) residing in the

Republic of the Congo. Overall, the manuscript is well written and organized. The results clearly demonstrate that WLG and humans/chimpanzees have distinct intestinal microbiota, and that the prevalence of certain WLG enterotypes change with diet/season. I have some concerns regarding the persistence of enterotypes 1 across dietary/seasonal shifts, especially given that this is the most prevalent enterotypes within the WLG.

Major Comments

Reviewer 4, comment 1: *My major concern is the prevalence of enterotypes 1 across dietary/seasonal shifts. It is unclear from the provided analyses if this enterotype is robust to dietary/seasonal shifts or if changes in diet/season impact the intestinal microbiota of WLG with this enterotype (though, to an extent insufficient to cause a change to a different enterotype). For example, while the relative abundance of samples assigned to enterotype 1 does not change between dry and wet months (as shown by Fig. 4e), there may still be important changes in the bacterial community within this enterotype. A more detailed examination between enterotype 1 samples across different factors (dry vs. wet, HF vs. LF) would substantially increase the impact of this paper.*

Authors' Response: Thank you for this suggestion. We have now included **Supplementary Fig. 11** to address this astute concern. Based on PCoA, our results suggest the distribution (prevalence or density) of high frugivory samples changes along PC1 which is defined by the ratio in relative abundance of the two seasonally fluctuant enterotype drivers, *Treponema/Prevotella* ratio. We have now included data that demonstrates this phenomenon in **Supplementary Figure 11**. It is evident, in the newly added **Supplementary Fig. 11c**, that the vast majority of high frugivory samples fall within the *Prevotella*-driven enterotype 3 or are *SHD-231*-driven enterotype 1 samples that cluster more closely to enterotype 3 along PC1. Even within the transitional enterotype 1 samples, those collected during the high frugivory period have a lower *Treponema/Prevotella* ratio, and the overall relative abundance of high frugivory and low frugivory biomarkers differ significantly between the seasons. Thus, enterotype 1 is not robust to dietary/seasonal shifts, which further supports our assertion that enterotype 1 represents a transitional enterotype where rapid changes in dietary patterns have occurred, but the microbiome has not yet responded to a degree that results in enterotype switching.

Reviewer 4, comment 2: *While some of the PCoA plots are explicitly specified as being generated under the weighted UniFrac distance, it is not always clear which distance measure was used. In particular, are the PCoA plots in Figure 1c and 1d generated using the rJSD measure? I understand that the enterotypes are defined using rJSD with PAM clustering, but am uncertain if this measure or some other measure was used during the PCoA inference. These PCoA plots, along with a few others, are also missing the amount of variation captured by the displayed components. Can you also please specify which program was used to generate the PCoA plots and verify that these are "unsupervised PCoA" plots, as opposed to "PCoA with an instrumental variable" plots as used in the Arumugam et al. paper. Related to the above point, do samples within*

enterotype 1 separate out into distinct dry vs. wet (or HF vs. LF) clusters within the PCoA plots.

Authors' Response: We have added better clarification to the text and figure legends regarding the distance metric on which each PCoA plot is based. We have also added the percent variance for each principal component in the PCoA plots and clarified in the methods that all PCoA plots were generated using either QIIME or the ade4 R package and are based on unsupervised PCoA. We did not see clear seasonal separation of WLG samples by PCoA. As indicated in our response to your prior question, yes, enterotype 1 samples separate along PC1 defined by the Treponema/Prevotella relative abundance ratio which is significantly lower in enterotype 1 samples collected during the high frugivory period compared to those collected in the low frugivory period (see **Supplementary Figure 11**).

Minor Comments requiring Attention

The following points are minor in nature, but important for clarifying the paper:

Reviewer 4, comment 3: *Given the wide use of LEfSe in the manuscript, a sentence or two on how to interpret the log-transformed LDA scores would be instructive.*

Authors' Response: We have added another sentence to the LEfSe portion of the methods to clarify the role and interpretation of the linear discriminant analysis (**Page 57, lines 1298-1305**).

Reviewer 4, comment 4: *I strongly recommend that the community profiles obtained from QIIME for the 87 WLG be included as supplemental data. This would substantially aid in further exploring the results of this study.*

Authors' Response: This is now included as **Supplementary Table 2**.

Reviewer 4, comment 5: *Based on Extended Data Figure 1d, it does not appear that Treponema and Fibrobacter are positively correlated except at extremely low abundances (i.e., Treponema < 0.1). I believe the large number of samples where Treponema < 0.1 and Fibrobacter < 0.002 are strongly influencing the correlation.*

Authors' Response: Thank you for this insight. We have removed this figure.

Reviewer 4, comment 6: *The whiskers in a box plot can represent many different statistics (2nd percentile, 9th percentile, measure of standard deviation from mean, etc.) so the statistic shown needs to be explicitly stated.*

Authors' Response: We have now defined the whiskers of the box-and-whiskers plots in the statistics portion of the methods section (**Page 57, lines 1291-1297**).

Reviewer 4, comment 7: *USEARCH should be cited in addition to QIIME.*

Authors' Response: We have included this citation.

Reviewer 4, comment 8: *The Box-Cox-transformation represents a family of transformation that is typically parameterized on lambda. Which value of lambda was used?*

Authors' Response: The Box-Cox transformations were performed in XLSTAT, which has an algorithm implemented to optimize the lambda for each variable. Therefore, each dependent variable (each taxon) had a different optimized lambda. We have now clarified this at the end of the *ANCOVA analysis* subsection of the statistics portion of the methods (**Page 61, lines 1379-1381**).

Reviewer 4, comment 9: *In at least one instance, Fig. 4e is referred to as Fig. 4d in the manuscript.*

Authors' Response: We have made this correction.

Minor Suggestions

Given the high quality of your manuscript, I would like to draw you attention to a few additional minor points:

Reviewer 4, comment 10: *Page numbers and line numbers should be added to aid in reviewing the manuscript.*

Authors' Response: Our apologies for this oversight. We have added page and line numbers.

Reviewer 4, comment 11: *The introduction indicates that human enterotypes are "dominated" by a Bacteroides, Prevotella, and Ruminococcus. Dominated may be a bit strong given that Ruminococcus is present < 1% and acts more as a biomarker.*

Authors' Response: We have changed the wording of this sentence so that it now indicates that these genera "define" the human enterotypes.

Reviewer 4, comment 12: *Gorillas are sometimes referred to as our closest relatives after chimpanzees, while other times bonobos are explicitly mentioned.*

Authors' Response: We have changed much of the phrasing in the introduction and discussion as we have added chimpanzees to our analyses, but when we referred to gorillas as our closest living relative after chimpanzees, we were referring to chimpanzees as the genus *Pan*, which includes the common chimpanzee and bonobos.

Reviewer 4, comment 13: *Figure 1c and 1d would be far more striking and easier to follow if samples were organized by decreasing abundance of Firmicutes. This approach was taken in Extended Data Figure 1b.*

Authors' Response: We have removed **Figure 1c-d** for this re-submission.

Reviewer 4, comment 14: *Extended Data Figure 10a lists 'wet season 1' twice in the legend.*

Authors' Response: We have made this correction.

Reviewer 4, comment 15: *The last sentence in the second paragraph of the "Bioinformatic analysis" section is awkward. It also mentions UPGMA clustering, but this does not appear in any figures.*

Authors' Response: We have made this correction and removed reference to UPGMA clustering.

Authors' References

- 1 Smits, S. A. *et al.* Seasonal cycling in the gut microbiome of the Hadza hunter-gatherers of Tanzania. *Science* **357**, 802-806, doi:10.1126/science.aan4834 (2017).
- 2 Moeller, A. H. *et al.* Rapid changes in the gut microbiome during human evolution. *Proceedings of the National Academy of Sciences of the United States of America* **111**, 16431-16435, doi:10.1073/pnas.1419136111 (2014).
- 3 Moeller, A. H. *et al.* Chimpanzees and humans harbour compositionally similar gut enterotypes. *Nature communications* **3**, 1179, doi:10.1038/ncomms2159 (2012).
- 4 Gomez, A. *et al.* Gut microbiome composition and metabolomic profiles of wild western lowland gorillas (*Gorilla gorilla gorilla*) reflect host ecology. *Molecular ecology* **24**, 2551-2565, doi:10.1111/mec.13181 (2015).
- 5 Moeller, A. H. *et al.* Sympatric chimpanzees and gorillas harbor convergent gut microbial communities. *Genome research* **23**, 1715-1720, doi:10.1101/gr.154773.113 (2013).
- 6 Ren, T., Grieneisen, L. E., Alberts, S. C., Archie, E. A. & Wu, M. Development, diet and dynamism: longitudinal and cross-sectional predictors of gut microbial communities in wild baboons. *Environmental microbiology* **18**, 1312-1325, doi:10.1111/1462-2920.12852 (2016).
- 7 Sak, B. *et al.* Long-Term Monitoring of Microsporidia, Cryptosporidium and Giardia Infections in Western Lowland Gorillas (*Gorilla gorilla gorilla*) at Different Stages of Habituation in Dzanga Sangha Protected Areas, Central African Republic. *PloS one* **8**, doi:ARTN e71840 10.1371/journal.pone.0071840 (2013).
- 8 Shutt, K. *et al.* Effects of habituation, research and ecotourism on faecal glucocorticoid metabolites in wild western lowland gorillas: Implications for conservation management. *Biol Conserv* **172**, 72-79, doi:10.1016/j.biocon.2014.02.014 (2014).
- 9 Santiago, M. L. *et al.* Simian immunodeficiency virus infection in free-ranging sooty mangabeys (*Cercocebus atys atys*) from the Tai Forest, Cote d'Ivoire: implications for the origin of epidemic human immunodeficiency virus type 2. *Journal of virology* **79**, 12515-12527, doi:10.1128/JVI.79.19.12515-12527.2005 (2005).
- 10 Van Heuverswyn, F. *et al.* Genetic diversity and phylogeographic clustering of SIVcpzPtt in wild chimpanzees in Cameroon. *Virology* **368**, 155-171, doi:10.1016/j.virol.2007.06.018 (2007).
- 11 Neel, C. *et al.* Molecular epidemiology of simian immunodeficiency virus infection in wild-living gorillas. *Journal of virology* **84**, 1464-1476, doi:10.1128/JVI.02129-09 (2010).
- 12 Arumugam, M. *et al.* Enterotypes of the human gut microbiome. *Nature* **473**, 174-180, doi:10.1038/nature09944 (2011).

- 13 Clayton, J. B. *et al.* Captivity humanizes the primate microbiome. *Proceedings of the National Academy of Sciences of the United States of America* **113**, 10376-10381, doi:10.1073/pnas.1521835113 (2016).
- 14 Nishihara, T. Feeding Ecology of Western Lowland Gorillas in the Nouabale-Ndoki National-Park, Congo. *Primates* **36**, 151-168, doi:Doi 10.1007/Bf02381342 (1995).
- 15 Remis, M. J., Dierenfeld, E. S., Mowry, C. B. & Carroll, R. W. Nutritional aspects of western lowland gorilla (*Gorilla gorilla gorilla*) diet during seasons of fruit scarcity at Bai Hokou, Central African Republic. *Int J Primatol* **22**, 807-836, doi:Doi 10.1023/A:1012021617737 (2001).
- 16 Doran-Sheehy, D., Mongo, P., Lodwick, J. & Conklin-Brittain, N. L. Male and female western gorilla diet: preferred foods, use of fallback resources, and implications for ape versus old world monkey foraging strategies. *American journal of physical anthropology* **140**, 727-738, doi:10.1002/ajpa.21118 (2009).
- 17 Masi, S., Cipolletta, C. & Robbins, M. M. Western lowland gorillas (*Gorilla gorilla gorilla*) change their activity patterns in response to frugivory. *American journal of primatology* **71**, 91-100, doi:10.1002/ajp.20629 (2009).
- 18 Cipolletta, C. Ranging patterns of a western gorilla group during habituation to humans in the Dzanga-Ndoki National Park, Central African Republic. *Int J Primatol* **24**, 1207-1226, doi:Doi 10.1023/B:Ijop.0000005988.52177.45 (2003).
- 19 Poulsen, J. R., Clark, C. J. & Smith, T. B. Seasonal variation in the feeding ecology of the grey-cheeked mangabey (*Lophocebus albigena*) in Cameroon. *American journal of primatology* **54**, 91-105, doi:Doi 10.1002/Ajp.1015 (2001).
- 20 Doran-Sheehy, D. M., Shah, N. F. & Heimbaurer, L. A. in *Feeding ecology in apes and other primates* (eds G. Hohmann, M. M. Robbins, & C. Boesch) 49-72 (Cambridge University Press, 2006).
- 21 Poulsen, J. R., Clark, C. J. & Smith, T. B. Seasonal variation in the feeding ecology of the grey-cheeked mangabey (*Lophocebus albigena*) in Cameroon. *American journal of primatology* **54**, 91-105, doi:10.1002/ajp.1015 (2001).
- 22 Rousvel, S. *et al.* Comparison between Vegetation and Rainfall of Bioclimatic Ecoregions in Central Africa. *Atmosphere-Basel* **4**, 411-427, doi:Doi 10.3390/Atmos4040411 (2013).
- 23 Shinoda, M. Seasonal Phase-Lag between Rainfall and Vegetation Activity in Tropical Africa as Revealed by Noaa Satellite Data. *Int J Climatol* **15**, 639-656, doi:Doi 10.1002/Joc.3370150605 (1995).
- 24 Ren, T., Grieneisen, L. E., Alberts, S. C., Archie, E. A. & Wu, M. Development, diet and dynamism: longitudinal and cross-sectional predictors of gut microbial communities in wild baboons. *Environmental microbiology*, doi:10.1111/1462-2920.12852 (2015).
- 25 Segata, N. *et al.* Metagenomic biomarker discovery and explanation. *Genome Biol* **12**, R60, doi:10.1186/gb-2011-12-6-r60 (2011).

- 26 Westfall, P. H. & Krishen, A. Optimally weighted, fixed sequence and gatekeeper multiple testing procedures. *J Stat Plan Infer* **99**, 25-40, doi:Doi 10.1016/S0378-3758(01)00077-5 (2001).
- 27 Dmitrienko, A., Offen, W. W. & Westfall, P. H. Gatekeeping strategies for clinical trials that do not require all primary effects to be significant. *Statistics in Medicine* **22**, 2387-2400 (2003).

Reviewers' comments:

Reviewer #1 (Remarks to the Author):

The manuscript "Seasonally Fluctuant Microbiota of Primates and Evidence Against the Enterotype Paradigm" by Williams and colleagues is a vastly revised revision of a manuscript submitted to Nature Communications some years ago. A notable alteration is the inclusion of new microbiome data from sympatric chimpanzees, which along with published data from humans and other primates allows for interesting comparisons. The authors have done a better job of describing the pattern of seasonal fluctuations, and have also included consideration of the probable functional differences of the enterotypes.

1. Please define enterotypes more thoroughly in the Introduction. In the Discussion it would also be possible to discuss whether this is a useful construct, or does it need revision? It is notable that lines 241-244 raise the possibility that continuous gradients may be a more accurate description, and analyses are structured accordingly, but the manuscript seems to vacillate and a clear statement would be welcome.

2. Lines 108-109. I do not find (discrete clustering) and the other parenthetical comment to be illuminating, please explain more clearly. Also in this paragraph, please make it clear that this is sampling from a population across time, not particular individuals repeatedly across time.

3. Lines 147-179. Here both genus level and species level patterns are described. Comparisons relative to humans differ at these two levels. Is this section interesting at all? If so, I would expect it to be addressed in the Discussion, but I think it is not mentioned, so I would suggest shortening this section.

4. line 221-222, rephrase first sentence, as one is not investigating relationships among those primate populations.

5. Lines 694-743, it is not clear to me why it is interesting to report the percentage of sequences that could be classified to whatever level, it obscures the message that here one can find the dominant taxa of the WLG diet.

6. Line 831, and elsewhere. Avoid statements that appear to attribute values to the different enterotypes. Here replace 'disintegration' with, for example, scarcity in humans in industrialized cultures

7. Line 845 replace 'salvage'

8. Line 883-884 replace 'drives' and driver as one cannot attribute agency to a bacterium.

9. Line 883-903. 956- onwards. So 4.6% of gorillas have the S-St enterotype, and are considered potentially 'atypical', while much is made of the much lower, or sometimes high, prevalence of *Treponema* in humans. Is it possible to make strong conclusions about prevalence or absence of particular enterotypes in different species/populations, or would it be more reasonable to admit that sample sizes are small and thus providing preliminary insights?

10. The last three paragraphs of the Discussion could be compressed as these are somewhat repetitive and wordy.

11. The title leads one to expect a mention of an 'enterotype paradigm' and how this is contradicted, but it is left a bit opaque in the manuscript. Is one not accepting the utility of classification of variation via enterotypes? Or what other characteristic of enterotypes is in question- presumably their immutability?

Reviewer #5 (Remarks to the Author):

This manuscript aims to characterize the fecal microbiota of wild western lowland gorillas (*Gorilla gorilla gorilla*) and sympatric central chimpanzees (*Pan troglodytes troglodytes*) using 16S rRNA sequencing. A major goal of their analysis is to demonstrate divergence of gorilla and chimpanzee enterotypes from the enterotypes previously reported in humans. This study included the following sample sets: 87 wild WLGs and 18 sympatric central chimpanzees from the Republic of the Congo, U.S. humans, Mongolian humans from urban and rural areas, and other nonhuman primate species (baboons, black and white colobus, red colobus, and red-tailed guenons). The authors showed that WLG microbiomes are seasonally dependent, and are primarily influenced by rainfall and fruit consumption. Using shotgun metagenomics sequencing on the gorilla samples, the authors showed distinctions in functional metabolic pathways and plant taxa included in the diet between gorilla enterotypes. A major conclusion of the manuscript is that enterotypes are seasonally fluctuant, change over short time scales, and do not define an individual. Overall, this manuscript presents a novel study focused on an important topic (i.e., seasonal differences in microbiome composition related to dietary intake), and therefore is consistent with articles published in the Nature Communications.

Assessment of the Author's Responses to Reviewer #3's comments:

Comment #1: For this to be a high impact study with significance to a broad readership, the authors need to do a better job placing this data into context of other human data (Tanzania, Malawi, Venezuela, US, Europe, etc) and non-human primates. Fig. S2 and S3 are not sufficient in this respect. It actually is now fairly trivial to resequence distinct regions of 16S (if DNA is already purified, which appears to be the case here) so that proper comparisons can be made. Obtaining V4 (Caporaso et al., 2011, PNAS), which would enable direct comparison to traditionally-living human populations AND V6-V9, for NHP, would enable comparisons required for important insight to make this a high impact study.

In my opinion, the addition of the sympatric chimpanzee samples, as well as the inclusion of the data from U.S. humans and the additional 4 nonhuman primate species addresses Reviewer #3's concerns. Also, I would argue that re-sequencing distinct regions of the 16S gene is not trivial. Sure, it is trivial from the standpoint that the primers for those regions are readily available. However, there is still a cost involved. Further, the time from sample submission to data generation, and ultimately data analysis can be lengthy. With that being said, it is important to compare the same V region when conducting meta-analyses. Currently, the most used V region is the V4 region. This is the V region that is used by the Earth Microbiome Project, and is the region that appears to offer the greatest resolution of bacterial taxa. The authors should have considered the best datasets to compare their data prior to selecting a V region for sequencing, however this is not relevant at this point. With all that being said, the authors selected datasets that were generated using V1-V3 primers. Given that the same V region was used for all the data analyzed, I feel that they have used sound methodology for both data generation and data analysis. My only reservation/question would be why they chose to compare the great ape populations with humans from Mongolia. Regardless, the addition of the U.S. humans, and other datasets, is sufficient to elevate this manuscript for publication in Nature Communications.

Comment #2: The paper is lacking diversity measures, which need to be included in comparison to human and NHP datasets.

While I did not see the original version of the manuscript, I would have made the same comment as Reviewer #3. For any publication focused on microbiome, diversity measures are standard. In other words, any paper that is focused on microbiome where data was generated using 16S rRNA sequencing should report diversity measures, including alpha diversity and beta diversity. As the authors state in their rebuttal, they added alpha diversity measures at phylum-level, genus-level, and 97% OTU-level. They also added beta diversity measures in the form of Bray-Curtis dissimilarity and square root Jensen-Shannon divergence. In my opinion, in order for the manuscript to be published in Nature Communications, the authors should also include beta diversity plots based on Weighted UniFrac and Unweighted UniFrac. If these are included, it is my opinion that the authors will have appropriately addressed Comment #2 by Reviewer 3.

Comment #3: The authors are only able to explore potential function restructuring with *Treponema*. "The remaining two enterotypes were defined by under-characterized bacterial genera (SHD-231 and *Solibacillus*). As such, their functional roles in the WLG microbiota remain enigmatic." Just a few metagenomic datasets for these currently enigmatic types would provide great insight and significantly strengthen the paper.

In my opinion, the addition of the shotgun metagenomics sequencing data (functional data) addresses this comment. That is, if I fully understand what the reviewer is asking for. Regardless, the addition of the shotgun sequencing data adds a substantial amount of invaluable information to their analysis. I know from personal experience addressing reviewers comments that the generation of shotgun sequencing data is substantially more expensive than 16S sequencing. Thus, the authors financial commitment to strengthening the manuscript through additional data generation (compared to the original submission) is noteworthy. Additionally, given that the authors are trying to establish a link between dietary patterns and microbiome composition, the inclusion of functional data becomes critical. Of the methods available to generate functional data (microbial function), shotgun sequencing offers that highest level of resolution, and is thus the most trusted. Additionally, the authors used Illumina HiSeq 2500, and generated 12-42 million reads per sample. This range (12-42 million reads per sample) is considered deep shotgun sequencing, and therefore I would trust this data in terms of providing accurate functional information. Overall, the addition of this data greatly strengthens the authors results, as it offers them the ability to test claims that they otherwise would not have been able to via the inclusion of information regarding microbial function.

Comment #4: The authors miss the opportunity to re-frame the term enterotype. The vague definition of "long-term dietary patterns" currently cited with enterotype in this field gains some concreteness in this study, albeit in another primate, but the authors should consider additional discussion of this point. There are additional questions about the validity of data manipulation required to create these groupings (not specific to this study) and how the number of clusters is determined (specific to this study). Just looking at Fig. 2c, it appears that three of the four clusters overlap significantly with many points localized more closely to another enterotype than the one to which the line is drawn. Designation of the 4 clusters presented in Fig. 2c as "enterotypes" requires more support. A major concern is - as presented - the ambiguous determination of 4 clusters (CH and SI scores are not easily distinguishable from pure chance). Furthermore, the differentiation of the 4 clusters on a PCoA plot appears to depend more on PC2 than PC1.

In my opinion, evidence for the concept of “enterotype” is lacking, and therefore the use of the term itself is highly controversial among research groups who readily undertake studies focused on the microbiome. The authors state the following: “In fact, there is much debate about whether discrete clustering into enterotypes is a valid method for describing variation between individuals, as human microbial communities appear to vary as continuous gradients rather than discrete clusters.” I like the fact that the authors have acknowledged that the characterization of discrete enterotypes is a much debated topic. Looking at Figure 2b, I would agree that for the WLJ that three of the four clusters overlap. In the authors rebuttal, they say “generally gradients of dominant bacteria that can fluctuate rapidly in response to environmental change”. I’m glad they made this statement. Again, as Reviewer 3 mentioned, I do not see the need for the use of enterotype at all here. The data is very interesting, as there are clearly seasonal differences, therefore why not just discuss this rather than giving this a term (i.e., enterotype). Despite my opinion here, at least they have defined what they mean by enterotype. However, I would caution the use of this term, as it may be confusing to readers who have read previous manuscripts where the term was used in a different context. The fact that they state the following in their rebuttal is helpful: “Finally, we have clarified that our intention is not to claim that Arumugam’s method defines statistically supported discrete clusters”.

Comment #5: Assignment of significance to Fig 3a and other figures that present LDA scores requires more support. Authors did not adjust for multiple testing in their ANOVA analysis and reference LEfSe's minimization of false positives as a basis for not performing multiple hypothesis correction. Please elaborate and demonstrate that results from LEfSe's do not statistically suffer from false positives.

Reviewer 3 raised some issues with the use of the LEfSe method, which the authors attempted to address in their rebuttal. In my opinion, the authors addressed Reviewer 3's concerns with their explanation of the method and the extensiveness for which it has been used in other studies. Additionally, the authors performed ANOVA/ANCOVA analyses on all bacterial taxa identified as monthly or seasonal metagenomics biomarkers by the LEfSe method. The combination of their thorough explanation and inclusion of additional statistical methods appropriately addresses Reviewer 3's concerns. On another note, regarding the use of the LEfSe method, Figure 2a is confusing to me. Specifically, it appears that they are using multiple taxonomic levels in combination to describe bacterial biomarkers for each study population. I disagree with this approach. If the authors want to describe bacterial biomarkers, the same taxonomic level should be used for all study populations. And, I would not use any taxonomic level higher than genus. For the red colobus they include Firmicutes as a bacterial biomarker, however Firmicutes is a phylum. Phylum-level does not offer enough resolution (i.e., not specific enough) to serve as a bacterial biomarker. Due to this, the analysis shown in Figure 2a should be redone with the aforementioned comments in mind.

Reviewer #6 (Remarks to the Author):

This is a really interesting study about the temporal fluctuations of the fecal microbiota of Gorillas and how they relate to season and feeding behavior. Some of the findings obtained are highly relevant, especially as equivalent observations have been made in a recent study on the seasonal fluctuations of the fecal microbiota in the Hadza, suggesting conserved patterns among hominoids. That being said, the manuscript has some serious weaknesses.

Major points:

1) I am a bit puzzled that a substantial portion of this manuscript is about findings obtained with comparisons of data obtained during this study (Gorillas and Chimpanzees) with those obtained in other studies (humans from the US and Mongolia, monkeys). First, I do not think these comparisons are very important (and an objective is not provided in the introduction). Second, and more importantly, I would argue that such comparisons are hugely confounded as the approaches used to generate such data were not standardized. It has been clearly established that, for example, DNA extraction methods impact sequencing outcomes, and even sending the same DNA to different sequencing centers has revealed substantial differences. In fact, some of the findings are questionable, for example that the WLG had a lower diversity based on OTUs compared to even humans from the US. In addition, there are a lot of studies showing high Firmicutes proportions in US individuals, not supporting the findings reported here (likely because differences in DNA extraction methods). I would argue that all the comparisons of sequence sets that originated from different studies should be removed as confounded.

2) I do not understand the excessive focus on enterotypes in the manuscript. As the authors note themselves, the whole concept is hugely controversial (I would say it is refuted), and the data doesn't support it. I do agree that enterotypes should be discussed in this manuscript, but I think it makes little sense to devote most of the manuscript on it if there is in fact little evidence that they exist. For example, in lines 802-803 and in line 821, the authors conclude that their findings challenge the notion of discrete enterotypes and that the mere term "enterotype" is misleading, just to then continue in line 824 to discuss the findings in the context of "enterotypes". I think there should be a concise section on enterotypes in the results and discussion, but overall, the findings should be presented as they make the most sense biologically. For example, in the recent paper on the Hadza by Smit et al, the term is not even mentioned, although the findings would in theory be relevant in this context. Clearly, the authors should leave enterotypes in the manuscript, but I suggest focusing the presentation and discussion of the findings on aspects that are biologically meaningful.

3) The manuscript is exhaustingly long and should be shortened, which could be achieved by removing comparisons of data generated in different studies (point 1 above) and content on enterotypes (point 2 above).

Specific points:

Line 29: First, I do not understand what 'divergence of enterotypes' means, and second, I disagree with the focus on enterotypes in the abstract given the fact that the authors consider themselves as 'misleading' in the text.

Line 44: I think this is an overstatement.

Line 59: 'Rural' refers to the countryside rather than the town. To my knowledge, Prevotella is overrepresented in non-industrialized populations and in the countryside.

Line 64-69: Please make consistent arguments here. First, the authors state that the factors that cause 'enterotype switching' are not understood, just to then state that 'in fact, there is much debate if discrete clusters are a valid method for describing variation between individuals'. These are two different things!

Line 96-100: I struggle to follow the logic here.

Line 103-105: Apart from the fact that such comparisons should not be made (see above), it is not clear that the objective was.

Line 117-118: Again, I think this is an overstatement. I do not see how that study showed anything about primate evolution.

Line 132-136: As written above, such comparisons should not be made if the sequences were not generated together. In fact, there are many studies that showed that humans are not dominated by Bacteroidetes.

Line 160" Shannon index is not a measure of evenness. It is a diversity measure that considers both abundance and evenness.

Line 167-168: I think this is highly unlikely and probably a bias due to the comparisons of data obtained with non-standardized methods (e.g. DNA extraction).

Line 307-308: Did Arumugam et al. not propose that these are in fact discrete clusters?

Line 752: There were earlier reports suggesting that enterotypes switch in US individuals. See: <https://www.ncbi.nlm.nih.gov/pmc/articles/PMC3744518/>

Lines 926-932: I do not find this discussion very convincing. Fruits contain quite a bit of fiber. Also, if Prevotella would in fact be a colonic mucolytic microbe dominating when fiber intake is minimized, would we not expect this genus to have a feast in industrialized humans? However, it is underrepresented in westerners.

Lines 1024-1045: I think these last two paragraphs are a bit far-fetched.

Line 1240-1243: As I write above, I don't think this should be done, but if included, there has to be a lot more information here on DNA extractions, cycle number in PCR, primers, etc. etc. Also, it should be mentioned that this approach has limitations and that even small differences in methods, handling, and sequencing chemistry can lead to artifacts.

Reviewers' comments

Reviewer #1 (Remarks to the Author):

The manuscript “Seasonally Fluctuant Microbiota of Primates and Evidence Against the Enterotype Paradigm” by Williams and colleagues is a vastly revised revision of a manuscript submitted to Nature Communications some years ago. A notable alteration is the inclusion of new microbiome data from sympatric chimpanzees, which along with published data from humans and other primates allows for interesting comparisons. The authors have done a better job of describing the pattern of seasonal fluctuations, and have also included consideration of the probable functional differences of the enterotypes.

Reviewer 1, comment 1: *Please define enterotypes more thoroughly in the Introduction. In the Discussion it would also be possible to discuss whether this is a useful construct, or does it need revision? It is notable that lines 241-244 raise the possibility that continuous gradients may be a more accurate description, and analyses are structured accordingly, but the manuscript seems to vacillate and a clear statement would be welcome.*

Authors' Response: Thank you for these questions. We have more thoroughly defined enterotypes in the introduction (**Lines 86-87** and **90-92**). In terms of whether enterotyping is a useful construct, our opinions and findings are in line with a perspective authored by many of the luminaries in the field of microbiome research published this month in *Nature Microbiology*¹. While many of these authors have previously questioned the validity of enterotypes, they appear to have now reached a consensus. They conclude that “enterotypes can be a useful tool for studying the human microbial community landscape” and that enterotypes “represent a way of capturing preferred microbial compositions in the human gut and thus appear to be useful stratifiers in many settings.” They do however suggest that, “Relying solely on enterotype classifications can obscure potentially important microbial variation, and therefore should not replace direct clinical associations and expert statistical analysis with microbial species and functions where possible.” Our study is in line with the opinions expressed by these leading experts: to supplement analysis of differential relative abundance of individual taxa based on factors of interest (e.g., seasons), we agree that enterotyping may provide some insights, independent of a factor of interest, by identifying high abundance taxa with substantial variation across samples and determining which samples fall around the extremes of these abundance distributions. Therefore, assessment of relative abundance of individual taxa combined with ordination analyses, dissimilarity metrics, enterotyping, and functional shotgun metagenomics, as we have done in this study, ultimately provides the most thorough assessment of variation in microbiota composition across samples.

Our findings, however, further suggest that, at least in our closest living relatives, enterotypes are malleable in response seasonality and diet. Thus, they are not static states that define an individual. We therefore suggest that caution should be applied when interpreting cluster membership, as individuals can change enterotypes.

We have more explicitly defined our position in the Discussion section (**Lines 775-789**).

Reviewer 1, comment 2: *Lines 108-109. I do not find (discrete clustering) and the other parenthetical comment to be illuminating, please explain more clearly. Also in this paragraph, please make it clear that this is sampling from a population across time, not particular individuals repeatedly across time.*

Authors' Response: We are grateful for the recommendation to clarify. We have removed parenthetical comments and more clearly defined the methods used (**Lines 120-125**). We have further made clear that this is a population-based study and not repeated sampling of individual great apes (**Lines 108-115**).

Reviewer 1, comment 3: *Lines 147-179. Here both genus level and species level patterns are described. Comparisons relative to humans differ at these two levels. Is this section interesting at all? If so, I would expect it to be addressed in*

the Discussion, but I think it is not mentioned, so I would suggest shortening this section.

Authors' Response: Thank you for highlighting this point. While we do think that it is interesting that there are some distinctions in alpha diversity between genus and species level comparisons with humans, the majority of our analyses throughout the manuscript are focused on genus-level and higher classification. As this point was raised by this reviewer and another, we have removed the species-level comparison, which has greatly shortened this section of the manuscript.

Reviewer 1, comment 4: *line 221-222, rephrase first sentence, as one is not investigating relationships among those primate populations.*

Authors' Response: We have rephrased this sentence (**Line 216**).

Reviewer 1, comment 5: *Lines 694-743, it is not clear to me why it is interesting to report the percentage of sequences that could be classified to whatever level, it obscures the message that here one can find the dominant taxa of the WLG diet.*

Authors' Response: Thank you for this recommendation. We agree with your assessment that this information, while accurate, obscures the broader message. These statements have been removed from the manuscript and supplementary information, but the information is retained in Supplementary Table 8.

Reviewer 1, comment 6: *Line 831, and elsewhere. Avoid statements that appear to attribute values to the different enterotypes. Here replace 'disintegration' with, for example, scarcity in humans in industrialized cultures*

Authors' Response: Thank you for this recommendation. We have made this change (**Lines 799, 814-815**).

Reviewer 1, comment 7: *Line 845 replace 'salvage'*

Authors' Response: Thank you for this suggestion. We have replaced 'salvage.' (**Lines 814-815**).

Reviewer 1, comment 8: *Line 883-884 replace 'drives' and driver as one cannot attribute agency to a bacterium.*

Authors' Response: We are grateful for this recommendation. We have replaced 'drives' and 'driver' when referring to a bacterium or its association with an enterotype throughout the manuscript. While "driver" was used to describe taxa that are associated with enterotypes in the original publication by Arumugam et al., it is more accurate to say that the relative abundance of a given bacterium

defines an enterotype, rather than that a bacterium drives or is a driver of the enterotype.

Reviewer 1, comment 9: *Line 883-903. 956- onwards. So 4.6% of gorillas have the S-St enterotype, and are considered potentially 'atypical', while much is made of the much lower, or sometimes high, prevalence of Treponema in humans. Is it possible to make strong conclusions about prevalence or absence of particular enterotypes in different species/populations, or would it be more reasonable to admit that sample sizes are small and thus providing preliminary insights?*

Authors' Response: We are grateful for the opportunity to clarify. The *Solibacillus/Staphylococcus*-dominated enterotype 4 is clearly extremely divergent in terms of the distributions of both bacterial genus and function abundance. Further, these samples are also divergent based on the abundance of fecal plant DNA, indicating that this microbial profile may be associated with a unique diet. As it is extremely divergent and present in only 4 out of 87 WLGs, we do consider it to be atypical. With our chimpanzee analysis limited to only 18 different chimpanzees, it is more difficult to make strong conclusions about the prevalence of particular enterotypes, though it is clear that the chimpanzees differ from the WLGs in terms of the dominant Spirochete genus in each ape species. We have generally refrained from making such strong statements, but we have added a note to the discussion stating that it's possible that with additional sampling of this chimpanzee population, additional shared or unique enterotypes might be revealed (**Lines 791-793**). It is clear that a few human populations have been identified that harbor appreciable levels of *Treponema*. We believe this is an important point to make, as *Treponema* (and other Spirochaetes genera) does not appear as a prevalent or abundant constituent of the microbiome of heavily studied industrialized societies, suggesting that Spirochaetes are largely missing from contemporary industrialized human populations (**Lines 872-885**).

Reviewer 1, comment 10: *The last three paragraphs of the Discussion could be compressed as these are somewhat repetitive and wordy.*

Authors' Response: Thank you for this recommendation, we have shortened and merged the 3rd and 4th to the last paragraphs of the discussion and also shortened the last two paragraphs.

Reviewer 1, comment 11: *The title leads one to expect a mention of an 'enterotype paradigm' and how this is contradicted, but it is left a bit opaque in the manuscript. Is one not accepting the utility of classification of variation via enterotypes? Or what other characteristic of enterotypes is in question- presumably their immutability?*

Authors' Response: At the request of the editors and other reviewers, we have changed the title of our manuscript and clarified our discussion of enterotypes. It

was our intention to contrast the utility of classification of enterotypes with the limitations of enterotyping that others and our study provide. Indeed, the utility of enterotyping has been questioned by many in the field. However, many of those experts in microbiome research who have questioned the utility of enterotypes have very recently acknowledged their utility along with their limitations¹ (see response to comment 1 above). We have more carefully defined our stance in the context of our findings in the discussion section (**Lines 775-789**).

Reviewer #5 (Remarks to the Author):

*This manuscript aims to characterize the fecal microbiota of wild western lowland gorillas (*Gorilla gorilla gorilla*) and sympatric central chimpanzees (*Pan troglodytes troglodytes*) using 16S rRNA sequencing. A major goal of their analysis is to demonstrate divergence of gorilla and chimpanzee enterotypes from the enterotypes previously reported in humans. This study included the following sample sets: 87 wild WLGs and 18 sympatric central chimpanzees from the Republic of the Congo, U.S. humans, Mongolian humans from urban and rural areas, and other nonhuman primate species (baboons, black and white colobus, red colobus, and red-tailed guenons). The authors showed that WLG microbiomes are seasonally dependent, and are primarily influenced by rainfall and fruit consumption. Using shotgun metagenomics sequencing on the gorilla samples, the authors showed distinctions in functional metabolic pathways and plant taxa included in the diet between gorilla enterotypes. A major conclusion of the manuscript is that enterotypes are seasonally fluctuant, change over short time scales, and do not define an individual. Overall, this manuscript presents a novel study focused on an important topic (i.e., seasonal differences in microbiome composition related to dietary intake), and therefore is consistent with articles published in the Nature Communications.*

Assessment of the Author's Responses to Reviewer #3's comments:

Comment #1: *For this to be a high impact study with significance to a broad readership, the authors need to do a better job placing this data into context of other human data (Tanzania, Malawi, Venezuela, US, Europe, etc) and non-human primates. Fig. S2 and S3 are not sufficient in this respect. It actually is now fairly trivial to resequence distinct regions of 16S (if DNA is already purified, which appears to be the case here) so that proper comparisons can be made. Obtaining V4 (Caporaso et al., 2011, PNAS), which would enable direct comparison to traditionally-living human populations AND V6-V9, for NHP, would enable comparisons required for important insight to make this a high impact study.*

Reviewer 5, Assessment of Comment 1: *In my opinion, the addition of the*

sympatric chimpanzee samples, as well as the inclusion of the data from U.S. humans and the additional 4 nonhuman primate species addresses Reviewer #3's concerns. Also, I would argue that re-sequencing distinct regions of the 16S gene is not trivial. Sure, it is trivial from the standpoint that the primers for those regions are readily available. However, there is still a cost involved. Further, the time from sample submission to data generation, and ultimately data analysis can be lengthy. With that being said, it is important to compare the same V region when conducting meta-analyses. Currently, the most used V region is the V4 region. This is the V region that is used by the Earth Microbiome Project, and is the region that appears to offer the greatest resolution of bacterial taxa. The authors should have considered the best datasets to compare their data prior to selecting a V region for sequencing, however this is not relevant at this point. With all that being said, the authors selected datasets that were generated using V1-V3 primers. Given that the same V region was used for all the data analyzed, I feel that they have used sound methodology for both data generation and data analysis. My only reservation/question would be why they chose to compare the great ape populations with humans from Mongolia. Regardless, the addition of the U.S. humans, and other datasets, is sufficient to elevate this manuscript for publication in Nature Communications.

Authors' Response: Thank you for concluding that we have met Reviewer #3's concerns on these points. At the time of selecting our V region for sequencing, nearly all non-human primate datasets (and still the majority of non-human primate datasets) were sequenced with the V1-V3 primers. Our inclusion of the Mongolian humans was because this was one of the few longitudinal datasets from humans, was sequenced with the same V-region as our study, and provided an additional comparator group to U.S. humans to demonstrate the differences in *Prevotella* vs. *Bacteroides* relative abundance as well as enterotype distribution in human populations.

Comment #2: *The paper is lacking diversity measures, which need to be included in comparison to human and NHP datasets.*

Reviewer 5, Assessment of Comment 2: *While I did not see the original version of the manuscript, I would have made the same comment as Reviewer #3. For any publication focused on microbiome, diversity measures are standard. In other words, any paper that is focused on microbiome where data was generated using 16S rRNA sequencing should report diversity measures, including alpha diversity and beta diversity. As the authors state in their rebuttal, they added alpha diversity measures at phylum-level, genus-level, and 97% OTU-level. They also added beta diversity measures in the form of Bray-Curtis dissimilarity and square root Jensen-Shannon divergence. In my opinion, in order for the manuscript to be published in Nature Communications, the authors should also include beta diversity plots based on Weighted UniFrac and Unweighted UniFrac. If these are included, it is my opinion that the authors will have appropriately addressed Comment #2 by Reviewer 3.*

Authors' Response: Thank you for this suggestion. We have now included PCoA based on unweighted UniFrac distance in the manuscript (**Supplementary Fig. 4d** and **Lines 223-226**). We agree that it is important to examine results based on different diversity metrics, as we have done with alpha diversity and with beta diversity. We have chosen to present plots and analyses based on BC, as this is commonly used in similar nonhuman primate microbiome studies (e.g.,^{2,3}) and is consistent with the use of abundance-weighted, non-phylogenetic-based metrics in enterotyping analyses. However, we had thoroughly assessed sample clustering based on weighted and unweighted UniFrac distance (below, left and right, respectively) and have now included PCoA based on unweighted UniFrac distance in the manuscript, but we don't believe that abundance-weighted phylogeny-based diversity metrics are well-suited for analysis of samples from different hosts. While there are some differences between clustering of samples based on host group by BC (**Fig. 1e**) and unweighted UniFrac (below, right and now **Supplementary Fig. 4d**) (*i.e.*, the two human populations separated from each other along PC1 and separated from the NHPs along PC2 by BC, while the opposite was the case by unweighted UniFrac), the sample clustering was overall largely similar between the two metrics. However, the clustering by weighted UniFrac (below, left) was quite different and provides little insight: in this case, position along PC1 is almost exclusively driven by *Bacteroides* and *Bifidobacterium* abundance, which are at high relative abundance in humans and baboons, respectively. The differential abundance of these two highly abundant taxa between humans and baboons is really the only significant interpretation that is revealed by weighted UniFrac. The large effect of these two taxa on sample clustering is likely due to the fact that *Bacteroides* and *Bifidobacterium* are so phylogenetically divergent, which, while noteworthy, obscures many important differences, such as the ratio of *Bacteroides* to *Prevotella* abundance, which is clearly a feature that differentiates the US humans from the majority of the Mongolian humans, as well as the great apes. While *Bacteroides* and *Prevotella* are more closely related phylogenetically, it is evident that their relative abundance in the gut microbiome has vastly different implications. Furthermore, due to the strong effect of *Bacteroides* and *Bifidobacterium* on the clustering, differences between the great apes and other Old World monkeys are also obscured (no separation). Thus, using a weighted phylogeny-based diversity metric in this context presents confusing results by obscuring such differences.

Comment #3: *The authors are only able to explore potential function restructuring with *Treponema*. "The remaining two enterotypes were defined by under-characterized bacterial genera (*SHD-231* and *Solibacillus*). As such, their functional roles in the WLG microbiota remain enigmatic." Just a few metagenomic datasets for these currently enigmatic types would provide great insight and significantly strengthen the paper.*

Reviewer 5, Assessment of Comment 3: *In my opinion, the addition of the shotgun metagenomics sequencing data (functional data) addresses this comment. That is, if I fully understand what the reviewer is asking for. Regardless, the addition of the shotgun sequencing data adds a substantial amount of invaluable information to their analysis. I know from personal experience addressing reviewers comments that the generation of shotgun sequencing data is substantially more expensive than 16S sequencing. Thus, the authors financial commitment to strengthening the manuscript through additional data generation (compared to the original submission) is noteworthy. Additionally, given that the authors are trying to establish a link between dietary patterns and microbiome composition, the inclusion of functional data becomes critical. Of the methods available to generate functional data (microbial function), shotgun sequencing offers that highest level of resolution, and is thus the most trusted. Additionally, the authors used Illumina HiSeq 2500, and generated 12-42 million reads per sample. This range (12-42 million reads per sample) is considered deep shotgun sequencing, and therefore I would trust this data in terms of providing accurate functional information. Overall, the addition of this data greatly strengthens the authors results, as it offers them the ability to test claims that they otherwise would not have been able to via the inclusion of information regarding microbial function.*

Authors' Response: Thank you for this positive assessment of our efforts.

Comment #4: *The authors miss the opportunity to re-frame the term enterotype. The vague definition of "long-term dietary patterns" currently cited with*

enterotype in this field gains some concreteness in this study, albeit in another primate, but the authors should consider additional discussion of this point. There are additional questions about the validity of data manipulation required to create these groupings (not specific to this study) and how the number of clusters is determined (specific to this study). Just looking at Fig. 2c, it appears that three of the four clusters overlap significantly with many points localized more closely to another enterotype than the one to which the line is drawn. Designation of the 4 clusters presented in Fig. 2c as "enterotypes" requires more support. A major concern is - as presented - the ambiguous determination of 4 clusters (CH and SI scores are not easily distinguishable from pure chance). Furthermore, the differentiation of the 4 clusters on a PCoA plot appears to depend more on PC2 than PC1.

Reviewer 5, Assessment of Comment 4: *In my opinion, evidence for the concept of "enterotype" is lacking, and therefore the use of the term itself is highly controversial among research groups who readily undertake studies focused on the microbiome. The authors state the following: "In fact, there is much debate about whether discrete clustering into enterotypes is a valid method for describing variation between individuals, as human microbial communities appear to vary as continuous gradients rather than discrete clusters." I like the fact that the authors have acknowledged that the characterization of discrete enterotypes is a much debated topic. Looking at Figure 2b, I would agree that for the WLG that three of the four clusters overlap. In the authors rebuttal, they say "generally gradients of dominant bacteria that can fluctuate rapidly in response to environmental change". I'm glad they made this statement. Again, as Reviewer 3 mentioned, I do not see the need for the use of enterotype at all here. The data is very interesting, as there are clearly seasonal differences, therefore why not just discuss this rather than giving this a term (i.e., enterotype). Despite my opinion here, at least they have defined what they mean by enterotype. However, I would caution the use of this term, as it may be confusing to readers who have read previous manuscripts where the term was used in a different context. The fact that they state the following in their rebuttal is helpful: "Finally, we have clarified that our intention is not to claim that Arumugam's method defines statistically supported discrete clusters".*

Authors' Response: We are grateful for this assessment. We have removed the term enterotype from the manuscript title and removed our analyses of enterotypes in the longitudinal samples from Mongolians at the request of the editor. However, we would like to further draw this reviewers attention to a recent perspective published in Nature Microbiology authored by many of the luminaries in the field of microbiome research¹. While many of these authors have previously questioned the validity of enterotypes, they appear to have now reached a consensus. They conclude that "enterotypes can be a useful tool for studying the human microbial community landscape" and that enterotypes "represent a way of capturing preferred microbial compositions in the human gut and thus appear to be useful stratifiers in many settings." They also suggest that,

“Relying solely on enterotype classifications can obscure potentially important microbial variation, and therefore should not replace direct clinical associations and expert statistical analysis with microbial species and functions where possible.” Our study is in line with the opinions expressed by these leading experts and therefore, we have now cited this recent article. To supplement analysis of differential relative abundance of individual taxa based on factors of interest (e.g., seasons), enterotyping may provide some insights, independent of a factor of interest, by identifying high abundance taxa with substantial variation across samples and determining which samples fall around the extremes of these abundance distributions. Therefore, assessment of relative abundance of individual taxa combined with ordination analysis, dissimilarity metrics, enterotyping, and functional shotgun metagenomics, as we have done here, ultimately provides the most thorough assessment of variation in microbiota composition across samples.

Comment #5: *Assignment of significance to Fig 3a and other figures that present LDA scores requires more support. Authors did not adjust for multiple testing in their ANOVA analysis and reference LEfSe's minimization of false positives as a basis for not performing multiple hypothesis correction. Please elaborate and demonstrate that results from LEfSe's do not statistically suffer from false positives.*

Reviewer 5, Assessment of Comment 5: *Reviewer 3 raised some issues with the use of the LEfSe method, which the authors attempted to address in their rebuttal. In my opinion, the authors addressed Reviewer 3's concerns with their explanation of the method and the extensiveness for which it has been used in other studies. Additionally, the authors performed ANOVA/ANCOVA analyses on all bacterial taxa identified as monthly or seasonal metagenomics biomarkers by the LEfSe method. The combination of their thorough explanation and inclusion of additional statistical methods appropriately addresses Reviewer 3's concerns. On another note, regarding the use of the LEfSe method, Figure 2a is confusing to me. Specifically, it appears that they are using multiple taxonomic levels in combination to describe bacterial biomarkers for each study population. I disagree with this approach. If the authors want to describe bacterial biomarkers, the same taxonomic level should be used for all study populations. And, I would not use any taxonomic level higher than genus. For the red colobus they include Firmicutes as a bacterial biomarker, however Firmicutes is a phylum. Phylum-level does not offer enough resolution (i.e., not specific enough) to serve as a bacterial biomarker. Due to this, the analysis shown in Figure 2a should be redone with the aforementioned comments in mind.*

Authors' Response: Thank you for recognizing that we have met the concerns of the previous reviewer regarding our use of LEfSe. We have included all taxonomic levels phylum to species-level for each of the non-human and human primate datasets. Thus, we are using the same taxonomic levels for all study populations. Our intentions here were two-fold. First, LEfSe provides

comparisons of relative abundance across taxa, which is informative regardless of taxonomic level. While LEfSe is billed as a biomarker discovery method, we don't particularly like the use of the term "biomarker" in this context, especially at high taxonomic levels. Therefore, in the text we now only refer to genus-level taxa that were identified by LEfSe, as well as metagenomic pathways, as "biomarkers". Higher taxa are described as having higher relative abundance in relation to other groups. Taking the example given by the reviewer in **Figure 2a**, we do indicate that the relative abundance of Firmicutes is significantly higher in red colobi, but it should be clear that the higher relative abundance of Firmicutes is largely attributable to the higher relative abundance of the lower-level Firmicutes taxa that are also indicated (i.e. Clostridia and unclassified *Ruminococcaceae*). However, another example wherein distinctions at higher taxonomic levels are informative comes from comparisons of WLGs and chimpanzees in this study. Both WLGs and chimpanzees have high relative abundance of Spirochaete genera that distinguish them, *Treponema* in WLGs and *Sphaerochaeta* in chimpanzees. However the relative abundance of Spirochaete phylum is higher in gorillas than chimpanzees, indicating that gorillas have higher overall relative abundance of Spirochaetes than chimpanzees. Another important distinction is the phylum Chloroflexi, which is quite unique in its high relative abundance in WLGs and is attributable to high relative abundance of SHD-231. However, if our analyses are restricted to genus-level and lower, it is not made clear that this entire phylum is unique in its high relative abundance in WLGs compared to humans and other NHPs. Thus, we believe that including all phylogenetic levels in our analyses places our findings in context for the reader. Secondly, most of the genera and species in non-human primates have not been characterized. This becomes readily apparent with our genus-level analyses for gorillas and other NHPs (**Fig. 1c** and **Supplementary Figure 1a-e**). Even the most abundant genera within NHPs remain unclassified at higher taxonomic levels (i.e. uBacteroidales in WLGs, u*Lachnospiraceae* in Chimpanzees, u*Erysipelotrichaceae* and u*Ruminococcaceae* in black-and-white colobi, u*Ruminococcaceae* and u*Lachnospiraceae* in red colobi and red-tailed guenons, to name a few).

Reviewer #6 (Remarks to the Author):

This is a really interesting study about the temporal fluctuations of the fecal microbiota of Gorillas and how they relate to season and feeding behavior. Some of the findings obtained are highly relevant, especially as equivalent observations have been made in a recent study on the seasonal fluctuations of the fecal microbiota in the Hadza, suggesting conserved patterns among hominoids. That being said, the manuscript has some serious weaknesses.

Major points:

Reviewer 6, comment 1: *I am a bit puzzled that a substantial portion of this*

manuscript is about findings obtained with comparisons of data obtained during this study (Gorillas and Chimpanzees) with those obtained in other studies (humans from the US and Mongolia, monkeys). First, I do not think these comparisons are very important (and an objective is not provided in the introduction). Second, and more importantly, I would argue that such comparisons are hugely confounded as the approaches used to generate such data were not standardized. It has been clearly established that, for example, DNA extraction methods impact sequencing outcomes, and even sending the same DNA to different sequencing centers has revealed substantial differences. In fact, some of the findings are questionable, for example that the WLG had a lower diversity based on OTUs compared to even humans from the US. In addition, there are a lot of studies showing high Firmicutes proportions in US individuals, not supporting the findings reported here (likely because differences in DNA extraction methods). I would argue that all the comparisons of sequence sets that originated from different studies should be removed as confounded.

Authors' Response: We appreciate the reviewers concerns and comments on this topic. Our inclusion of human and other non-human primate datasets in comparison to our own analyses of WLGs and chimpanzees was in direct response to two previous reviewers' critiques from our original submission to Nature Communications. One reviewer requested (and the editor agreed) that we should present our results "in a phylogenetic framework that would aid in inferring which lineages have undergone particular changes since the common ancestors." Another reviewer, similarly, asked that we "do a better job placing this data into context of other human data and non-human primates." In addition to our WLG population, we included sympatric chimpanzees (humans and WLGs closest phylogenetic relative) from our population survey, two human datasets generated by others (U.S. and Mongolian humans), as well as 4 Old World monkeys from datasets that were sequenced with the same primers (V1-V3) and on the same platform but by different laboratories. Thus, in **Figure 1c-e, Figure 2, Supplementary Figures 1, 3, and 4**, we contrast the specific bacterial lineages, alpha and beta diversity, and enterotypes that differentiate these species and populations and further elaborate on these differences in the discussion section. The remainder of our paper is focused on our WLG and chimpanzee analyses.

We fully agree that differences in methodology can have some influence on sequencing outcomes as the literature supports this assertion. However, the literature does not support the intensity of the reviewer's statement (i.e. "hugely confounded" and "substantial differences"). We would draw this reviewer's attention to the publication of Catherine Lozupone, Jeffrey Gordon, Rob Knight and others wherein they have shown that technical variation (including choice of PCR primers, sequencing platform used, and DNA extraction technique) is outweighed by compositional differences observed between Westernized and agrarian cultures. These authors conclude: "cross-study comparisons of human microbiota are valuable when the studied parameter has a large effect size."⁴. The effect size between different geographic regions (i.e. Mongolians vs. U.S.

humans) and between different animal species (Homo sapiens vs. WLGs vs. chimpanzees vs. different Old World monkeys) represents a study parameter that has a large effect size. In fact, there is precedent for cross-study comparison of great apes and humans that cite this reference², as well as cross-study comparisons between different NHPs and other mammals⁵ (to name only a few examples). Further, the study referenced by the reviewer below on seasonality of the Hadza microbiome compared human sequencing data from 18 populations, across 16 countries, derived from 26 cohorts in cross-study analyses that included a wide range of methodological differences including the variable region of the 16S rRNA gene targeted⁶. We have added a statement about methodological differences and their potential impact in our study and referenced the appropriate literature to support our comparisons (**Lines 1226-1236**).

Similar to this reviewer, we were perplexed by our results showing that WLGs had lower diversity at the OTU-level compared to humans, while WLGs had higher diversity at all other taxonomic levels. However, we do not agree that this is an artifact of extraction method as this reviewer suggests. We would like to direct this reviewer to two publications that assessed variation in the microbiota due to DNA extraction methods^{7,8}. In these studies, each of the methods used to generate datasets investigated in our study were compared. Only the ZR Fecal DNA MiniPrep method, which was not used for any of the datasets in our study, had a substantial impact. None of the extraction methods used for datasets in our study were found to have a significant impact on diversity metrics (the Qiagen DNA stool Mini-Kit with bead-beating used by ourselves and for the Mongolian humans, phenol-chloroform extraction used by Ren et al. in the baboons and by Yildirim et al. for the other old world monkeys, and the MO BIO PowerSoil DNA Isolation method used by the HMP). Furthermore, one would expect that differences in extraction methods would largely impact the diversity of Gram-positive to Gram-negative bacteria, but this effect should be seen at the genus level as well, yet all NHPs have higher genus-level diversity than U.S. humans. Thus, we can only postulate two interpretations: 1) WLGs have more genera but fewer species within each genera overall, which is what we suggested in the manuscript, or 2) there is some other methodological factor at the level of bioinformatics that artificially reduces the number of OTUs in WLGs and perhaps chimpanzees when OTUs are picked en masse for the combined dataset. Because we cannot definitively validate our finding at the OTU-level, most of our analyses are conducted at the genus-level, OTUs are only units that arbitrarily define species diversity and frequently over-inflate species diversity measures, and another reviewer has also suggested that our results section comparing genus-level and OTU-level alpha diversity is too long, we have decided to remove the OTU-level alpha diversity analyses from the manuscript.

Reviewer 6, comment 2: *I do not understand the excessive focus on enterotypes in the manuscript. As the authors note themselves, the whole concept is hugely controversial (I would say it is refuted), and the data doesn't support it. I do agree that enterotypes should be discussed in this manuscript, but I think it makes little sense to devote most of the manuscript on it if there is in fact*

little evidence that they exist. For example, in lines 802-803 and in line 821, the authors conclude that their findings challenge the notion of discrete enterotypes and that the mere term “enterotype” is misleading, just too then continue in line 824 to discuss the findings in the context of “enterotypes”. I think there should be a concise section on enterotypes in the results and discussion, but overall, the findings should be presented as they make the most sense biologically. For example, in the recent paper on the Hadza by Smit et al, the term is not even mentioned, although the findings would in theory be relevant in this context. Clearly, the authors should leave enterotypes in the manuscript, but I suggest focusing the presentation and discussion of the findings on aspects that are biologically meaningful.

Authors’ Response: Thank you for these comments and suggestions. To say that the concept of enterotypes “is refuted” and to suggest that enterotypes are not “biologically meaningful” is not in line with the perspectives of many of the luminaries in the field of microbiome research. We would like to draw the reviewer’s attention to a recent article in Nature Microbiology authored by many experts in the field of microbiome research¹. While many of these researchers have previously questioned the validity of enterotypes, they appear to have now reached a consensus. They conclude that “enterotypes can be a useful tool for studying the human microbial community landscape” and that enterotypes “represent a way of capturing preferred microbial compositions in the human gut and thus appear to be useful stratifiers in many settings.” They also suggest that, “Relying solely on enterotype classifications can obscure potentially important microbial variation, and therefore should not replace direct clinical associations and expert statistical analysis with microbial species and functions where possible.” Furthermore, as these experts point out and reference, enterotypes do seem to discriminate biological functions in humans, suggesting that enterotypes are biologically meaningful^{9,10}. Our study is in line with the opinions expressed by these leading experts: to supplement analysis of differential relative abundance of individual taxa based on factors of interest (e.g., seasons), we agree that enterotyping may provide some insights, independent of a factor of interest, by identifying high abundance taxa with substantial variation across samples and determining which samples fall around the extremes of these abundance distributions. Therefore, assessment of relative abundance of individual taxa combined with ordination analyses, dissimilarity metrics, enterotyping, and functional shotgun metagenomics, as we have done here, ultimately provides the most thorough assessment of variation in microbiota composition across samples.

Our original manuscript submitted to Nature Communications did not contain the word “Enterotypes” in the title and enterotyping was less represented in the manuscript. It was the consensus of multiple reviewers from our prior submission that we should add more information and analyses relating to enterotypes to the manuscript. Some examples: Initial Reviewer #2- “How are enterotypes defined?” What is known about their function (linked to what specific diet)? How have these been shown to vary in within and between human and

ape populations?” Initial Reviewer #3- *“Two important findings are the change in microbiota over seasonal dietary variation, and the demonstration that enterotype does not define an individual. The authors miss the opportunity to re-frame the term enterotype. The vague definition of “long-term dietary patterns” currently cited with enterotype in this field gains some concreteness in this study, albeit in another primate, but the authors should consider additional discussion of this point.”* Initial Reviewer #4- *“My major concern is the prevalence of enterotypes 1 across dietary/seasonal shifts. It is unclear from the provided analyses if this enterotype is robust to dietary/seasonal shifts or if changes in diet/season impact the intestinal microbiota of WLG with this enterotype (though, to an extent insufficient to cause a change to a different enterotype). For example, while the relative abundance of samples assigned to enterotype 1 does not change between dry and wet months (as shown by Fig. 4e), there may still be important changes in the bacterial community within this enterotype. A more detailed examination between enterotype 1 samples across different factors (dry vs. wet, HF vs. LF) would substantially increase the impact of this paper.”*

We agree with the assessment of this reviewer that, in some aspects, we may have overshot our attempts to appease previous reviewers' comments. We have therefore made several revisions here. We have removed mention of “enterotypes” from the title of the manuscript and reduced its mention in the abstract. We have also restructured the introduction to better highlight our assessment of the impact of seasonality and diet on the microbiome. We have also removed the assessment of longitudinal Mongolian human enterotype switching from the results and discussion at the request of the editor.

Reviewer 6, comment 3: *The manuscript is exhaustingly long and should be shortened, which could be achieved by removing comparisons of data generated in different studies (point 1 above) and content on enterotypes (point 2 above).*

Authors' Response: At the request of the editor, we have removed our entire section on enterotype switching in Mongolians. We have removed our OTU-level alpha diversity analyses and results, which were particularly long. See our response to point 1 and 2 above.

Specific points:

Reviewer 6, comment 4: *Line 29: First, I do not understand what ‘divergence of enterotypes’ means, and second, I disagree with the focus on enterotypes in the abstract given the fact that the authors consider themselves as ‘misleading’ in the text.*

Authors' Response: Thank you for noting that the term “divergence of enterotypes” is confusing to the reader. We have changed this statement to reflect that we have “demonstrated compositional divergence” (**Lines 32-33**). Further, we now only mention enterotypes once in the abstract (**Line 35**).

Reviewer 6, comment 5: *Line 44: I think this is an overstatement.*

Authors' Response: We have tempered this statement to suggest that insights may be gained “into the mechanisms by which diet has driven the evolution of modern human gut microbial communities” rather than human evolution (**Line 47**), although both diet-driven evolution and the hologenome theory of evolution are relevant to our findings.

Reviewer 6, comment 6: *Line 59: 'Rural' refers to the countryside rather than the town. To my knowledge, Prevotella is overrepresented in non-industrialized populations and in the countryside.*

Authors' Response: Thank you for this suggestion. We have changed our wording to reflect differences between industrialized communities and non-industrialized and rural populations.

Reviewer 6, comment 7: *Line 64-69: Please make consistent arguments here. First, the authors state that the factors that cause 'enterotype switching' are not understood, just to then state that 'in fact, there is much debate if discrete clusters are a valid method for describing variation between individuals'. These are two different things!*

Authors' Response: Thank you for this suggestion. We have now made our arguments consistent. In light of the recently published perspective regarding the enterotyping method in Nature Microbiology¹, we have removed the second statement (**Lines 100-103**).

Reviewer 6, comment 8: *Line 96-100: I struggle to follow the logic here.*

Authors' Response: We have now removed discussion of humans here and only focus on seasonal fluctuation in WLGs. (**Lines 79-91**).

Reviewer 6, comment 9: *Line 103-105: Apart from the fact that such comparisons should not be made (see above), it is not clear that the objective was.*

Authors' Response: See response to comment 1 above. In response to our first submission, one reviewer requested (and the editor agreed) that we should present our results “in a phylogenetic framework that would aid in inferring which lineages have undergone particular changes since the common ancestors.” Another reviewer, similarly asked that we “do a better job placing this data into context of other human data and non-human primates.” We have now clarified this objective in the manuscript (**Lines 114-115**).

Reviewer 6, comment 10: *Line 117-118: Again, I think this is an overstatement. I do not see how that study showed anything about primate evolution.*

Authors' Response: There have been many recent studies suggesting that primates and their microbiomes have co-evolved and that diet, seasonality and the microbiome played an important role in shaping primate evolution. We have provided a thorough discussion, replete with relevant references, placing our findings in the context of these evolutionary concepts (e.g., **Lines 973-1002**).

Reviewer 6, comment 11: *Line 132-136: As written above, such comparisons should not be made if the sequences were not generated together. In fact, there are many studies that showed that humans are not dominated by Bacteroidetes.*

Authors' Response: See our response to comment 1 above. We recognize that there is a large degree of inconsistency in the literature regarding the dominance of Bacteroidetes over Firmicutes in humans (especially in the U.S.), but many studies using different extraction methods indicate a dominance of Bacteroidetes. This is, in fact, a question that we are addressing in another study, wherein we have used the same extraction method with bead-beating on U.S. human stool samples (from 5 states) and compared unbiased shotgun metagenomic sequencing, V1-V3 sequencing (454), V1-V2 sequencing (MiSeq), and V4 sequencing (MiSeq) on the same samples. Our currently unpublished findings indicate a dominance of Bacteroidetes using unbiased shotgun metagenomics, V1-V3 and V1-V2. In contrast, V4 sequencing on the same extracted samples showed a very high relative abundance of Firmicutes compared to Bacteroidetes. Thus, based on our findings, it appears that V4 primers may have some inherent bias toward Firmicutes that may explain why more recent studies are reporting higher Firmicutes in humans. While we are still evaluating the reason for this apparent bias, it is clear from the literature that there is no consensus on this topic and addressing this here is beyond the scope of this study.

Reviewer 6, comment 12: *Line 160" Shannon index is not a measure of evenness. It is a diversity measure that considers both abundance and evenness.*

Authors' Response: Thank you for pointing this out, we have made this correction. (**Lines 171-173**).

Reviewer 6, comment 13: *Line 167-168: I think this is highly unlikely and probably a bias due to the comparisons of data obtained with non-standardized methods (e.g. DNA extraction).*

Authors' Response: See response to comment 1. We have removed this section of the manuscript.

Reviewer 6, comment 14: *Line 307-308: Did Arumugam et al. not propose that these are in fact discrete clusters?*

Authors' Response: In fact, Arumugam did not propose that enterotypes are discrete in their original definition, which is highlighted in a recent perspective:

“the original definition [of enterotypes] had made clear that they are not discrete, and that clustering is just one way to define them and stratify samples to reduce complexity.”¹ As we have referenced, others have challenged the notion that these clusters are statistically supported. Therefore, we have included this statement to indicate that we recognize this limitation of enterotyping. Regardless of whether Arumugam et al. originally proposed that enterotypes are discrete clusters, the utility of enterotyping has recently been supported by many of those researchers who have challenged the method¹. In fact, they go so far as to say: “We find that the gut microbial composition is structured and that clustering can provide useful insights into some microbiome datasets, even when not strongly supported statistically.”¹

Reviewer 6, comment 15: Line 752: There were earlier reports suggesting that enterotypes switch in US individuals. See: <https://www.ncbi.nlm.nih.gov/pmc/articles/PMC3744518/>

Authors’ Response: Thank you for providing this reference, we have now included this reference in **Line 101**.

Reviewer 6, comment 16: Lines 926-932: I do not find this discussion very convincing. Fruits contain quite a bit of fiber. Also, if *Prevotella* would in fact be a colonic mucolytic microbe dominating when fiber intake is minimized, would we not expect this genus to have a feast in industrialized humans? However, it is underrepresented in westerners.

Authors’ Response: We have now provided eight references (although there are many more) from leading experts on the diets of gorilla populations and seasonality in their diets (**Line 916**). Studies that have assessed the macro- and micro-nutrient composition of gorilla diets during the low frugivory period compared to the high frugivory period consistently indicate that succulent fruits consumed by gorillas during the frugivory period are lower in fiber content and higher in easily digestible sugars than staple/fallback foods consumed during the low frugivory period. What we as humans consider high in fiber (given that humans in the U.S. consume an approximate daily average of 15 grams of fiber) is not necessarily high for gorillas that are consuming 1200-1700 grams of fiber per day.

Thank you for this question about *Prevotella*. It would be incorrect to assume that all species within a genus have the same metabolic functions. We have indicated that our metagenomic findings relating to *Prevotella* in WLGS is in opposition to our understanding of *Prevotella* in humans and offered explanations for why this might be the case (**Lines 841-847**). Indeed, the dominant *Prevotella* OTUs in humans (accounting cumulatively for 89% of all *Prevotella* sequences in U.S. and Mongolians humans) were classified to the species level (*Prevotella copri* and *P. stercorea*). In contrast, 92% of all *Prevotella* sequences from WLGS remained unclassified at the species level, suggesting that they are different species. We have now clarified this in **Lines 847-853**. One cannot make a

generalized assumption that all species within a genus encode similar functions. Different species within a genus can have very different metabolic functions, require different substrates for their growth, or behave as specialists or generalists with respect to their ability to utilize glycans. Even within the same host “genus-level metabolic consistency is not guaranteed.”⁹ To further make this distinction, we indicated that some strains of *Prevotella*, such as *Prevotella strain RS2*, encode a mucin desulfating sulfatase (see **Lines 616-618**).

For these reasons, we would not expect human *Prevotella copri* or *Prevotella stercorea* to have a feast in westernized humans as they have a distinct set of requirements for growth on glycans compared to the species of *Prevotella* found in WLGs.

Reviewer 6, comment 17: *Lines 1024-1045: I think these last two paragraphs are a bit far-fetched.*

Authors’ Response: Without specific information as to what this reviewer deems as far-fetched, we must respectfully disagree. In our second to last paragraph, our intention is to draw attention to conservation, which we believe to be extremely important for these endangered species. There is already substantial evidence that we have cited in the second to last paragraph to indicate that external forces are having an impact on NHP microbiomes and health. Our last paragraph suggests that studying the microbiome of our closest living relatives may provide clues to our evolutionary past and our future health. Some of these evolutionary relationships are already being evaluated. We would draw your attention to the work of Moeller and Ochman on cospeciation of gut microbiota with hominids¹¹, as well as Groussin et al, 2017¹². We do not believe that suggesting that microbes could be transferred from one species to another and confer functional benefits is far-fetched and there are a large number of published studies in mice with humanized microbiomes, as well as studies with human-derived probiotics given to mice, NHPs and other animals which show its potential. We have now shortened these two paragraphs to make them more succinct (**Lines 1003-1024**).

Reviewer 6, comment 18: *Line 1240-1243: As I write above, I don’t think this should be done, but if included, there has to be a lot more information here on DNA extractions, cycle number in PCR, primers, etc. etc. Also, it should be mentioned that this approach has limitations and that even small differences in methods, handling, and sequencing chemistry can lead to artifacts.*

Authors’ Response: See our response to comment 1.

1 Costea, P. I. *et al.* Enterotypes in the landscape of gut microbial community composition. *Nat Microbiol* **3**, 8-16, doi:10.1038/s41564-017-0072-8 (2018).

- 2 Moeller, A. H. *et al.* Rapid changes in the gut microbiome during human
evolution. *Proceedings of the National Academy of Sciences of the United States*
of America **111**, 16431-16435, doi:10.1073/pnas.1419136111 (2014).
- 3 Gomez, A. *et al.* Gut microbiome composition and metabolomic profiles of
wild western lowland gorillas (*Gorilla gorilla gorilla*) reflect host ecology. *Mol*
Ecol **24**, 2551-2565, doi:10.1111/mec.13181 (2015).
- 4 Lozupone, C. A. *et al.* Meta-analyses of studies of the human microbiota.
Genome research **23**, 1704-1714, doi:10.1101/gr.151803.112 (2013).
- 5 Ren, T., Grieneisen, L. E., Alberts, S. C., Archie, E. A. & Wu, M. Development,
diet and dynamism: longitudinal and cross-sectional predictors of gut
microbial communities in wild baboons. *Environmental microbiology* **18**,
1312-1325, doi:10.1111/1462-2920.12852 (2016).
- 6 Smits, S. A. *et al.* Seasonal cycling in the gut microbiome of the Hadza hunter-
gatherers of Tanzania. *Science* **357**, 802-806, doi:10.1126/science.aan4834
(2017).
- 7 Wagner Mackenzie, B., Waite, D. W. & Taylor, M. W. Evaluating variation in
human gut microbiota profiles due to DNA extraction method and inter-
subject differences. *Front Microbiol* **6**, 130, doi:10.3389/fmicb.2015.00130
(2015).
- 8 Rintala, A. *et al.* Gut Microbiota Analysis Results Are Highly Dependent on the
16S rRNA Gene Target Region, Whereas the Impact of DNA Extraction Is
Minor. *Journal of biomolecular techniques : JBT* **28**, 19-30, doi:10.7171/jbt.17-
2801-003 (2017).
- 9 Vieira-Silva, S. *et al.* Species-function relationships shape ecological
properties of the human gut microbiome. *Nat Microbiol* **1**, doi:Artn 16088
10.1038/Nmicrobiol.2016.88 (2016).
- 10 Li, J. *et al.* A metagenomic approach to dissect the genetic composition of
enterotypes in Han Chinese and two Muslim groups. *Systematic and applied*
microbiology, doi:10.1016/j.syapm.2017.09.006 (2017).
- 11 Moeller, A. H. *et al.* Cospeciation of gut microbiota with hominids. *Science*
353, 380-382, doi:10.1126/science.aaf3951 (2016).
- 12 Groussin, M. *et al.* Unraveling the processes shaping mammalian gut
microbiomes over evolutionary time. *Nature communications* **8**, doi:Artn
14319
10.1038/Ncomms14319 (2017).

REVIEWERS' COMMENTS:

Reviewer #5 (Remarks to the Author):

[No further comments for author.]

Reviewer #6 (Remarks to the Author):

I do think that this revised version is vastly improved. I continue to have concerns about the comparisons of data sets that have not been produced together, but agree with the authors that this is often standard in the field. So overall, I think this is an interesting manuscript that makes an important contribution to the field.

Response to Referees

REVIEWERS' COMMENTS:

Reviewer #5 (Remarks to the Author):

[No further comments for author.]

Reviewer #6 (Remarks to the Author):

I do think that this revised version is vastly improved. I continue to have concerns about the comparisons of data sets that have not been produced together, but agree with the authors that this is often standard in the field. So overall, I think this is an interesting manuscript that makes an important contribution to the field.

Authors' Response: We are grateful that reviewer 6 has acknowledged improvement of the manuscript and recognized the important contribution our manuscript will make to the field.